# Protrudin and PDZD8 contribute to neuronal integrity by promoting lipid extraction required for endosome maturation

Michiko Shirane[1✉], Mariko Wada[1], Keiko Morita[1], Nahoki Hayashi[1], Reina Kunimatsu[1], Yuki Matsumoto[1], Fumiko Matsuzaki[2], Hirokazu Nakatsumi[1], Keisuke Ohta [3], Yasushi Tamura[4] & Keiichi I. Nakayama [2✉]

Endosome maturation depends on membrane contact sites (MCSs) formed between endoplasmic reticulum (ER) and endolysosomes (LyLEs). The mechanism underlying lipid supply for this process and its pathophysiological relevance remains unclear, however. Here, we identify PDZD8—the mammalian ortholog of a yeast ERMES subunit—as a protein that interacts with protrudin, which is located at ER-LyLE MCSs. Protrudin and PDZD8 promote the formation of ER-LyLE MCSs, and PDZD8 shows the ability to extract various lipids from the ER. Overexpression of both protrudin and PDZD8 in HeLa cells, as well as their depletion in mouse primary neurons, impairs endosomal homeostasis by inducing the formation of abnormal large vacuoles reminiscent of those apparent in spastin- or REEP1-deficient neurons. The protrudin-PDZD8 system is also essential for the establishment of neuronal polarity. Our results suggest that protrudin and PDZD8 cooperatively promote endosome maturation by mediating ER-LyLE tethering and lipid extraction at MCSs, thereby maintaining neuronal polarity and integrity.

[1] Department of Molecular Biology, Graduate School of Pharmaceutical Sciences, Nagoya City University, Nagoya, Aichi, Japan. [2] Department of Molecular and Cellular Biology, Medical Institute of Bioregulation, Kyushu University, Fukuoka, Fukuoka, Japan. [3] Department of Anatomy, Kurume University School of Medicine, Kurume, Fukuoka, Japan. [4] Department of Material and Biological Chemistry, Faculty of Science, Yamagata University, Yamagata, Yamagata, Japan. ✉email: shiram@phar.nagoya-cu.ac.jp; nakayak1@bioreg.kyushu-u.ac.jp

The endoplasmic reticulum (ER) is the largest intracellular organelle, consisting of an interconnected network of tubules and sheets, and is a major storage site for lipids and ions. It has recently become clear that most organelles do not function independently but rather communicate with the ER network through membrane contact sites (MCSs), microdomains at which the membranes of the ER and other organelles are closely apposed (with a gap of <30 nm) and tethered. MCSs are thought to function as intracellular synapses, at which molecular information is exchanged within a confined space. The identity of tethering factors at MCSs and the mechanisms by which these structures regulate cellular processes—such as lipid transfer, calcium ion homeostasis, and organelle dynamics—have remained largely unclear, however[1–6].

Endosomes are organelles that play an essential role in the reutilization or degradation of membrane components associated with the regulation of fundamental cellular activities. Components endocytosed from the plasma membrane are delivered to early endosomes (EEs), a subset of which undergoes conversion to late endosomes (LEs). Luminal invaginations of the LE membrane can be pinched off as free cargo-containing intraluminal vesicles (ILVs), resulting in the formation of a multivesicular body (MVB). The formation of MVBs is pivotal not only to lysosome-mediated degradation of endocytosed material but also to intercellular communication via the release of extracellular vesicles, or exosomes[7]. Given that the conversion of LEs to lysosomes is a continuous process, it has recently been proposed that the vesicles on this continuum be collectively referred to as endolysosomes (LyLEs)[8]. Endosomal membrane dynamics are largely attributable to a combination of endosome fusion and fission[9,10]. Endosome maturation is associated with expansion of the LyLE membrane[11], with the lipids required for this expansion being thought to be supplied by the ER at MCSs. However, the factors that tether LyLEs to the ER at MCSs and the mechanism of lipid transfer from the ER to LyLEs have been mostly unknown[12–15].

Protrudin is an ER-resident protein that functions as a tethering factor at ER-LyLE MCSs[16–19]. Protrudin promotes neurite formation by tethering endosomes to the ER in cooperation with vesicle-associated membrane protein-associated protein (VAP) and thereby facilitating loading of the endosome membrane with KIF5 (kinesin heavy chain), which is required for movement of the endosomes along microtubules in the plus-end direction[20–23]. Protrudin thus promotes membrane supply to the tip of neurites via MCS-dependent endosome trafficking, resulting in polarized neurite outgrowth that establishes neuronal polarity. Protrudin contains a hairpin (HP) domain that regulates membrane curvature and ER morphology, and mutations of the protrudin gene have been found to be responsible for hereditary spastic paraplegia (HSP), a neurodegenerative disease[24–29]. Neurons with HSP-associated mutations of the genes for spastin or REEP1, both of which also contain an HP domain, were recently shown to manifest grossly enlarged LyLEs and lysosomal dysfunction as a result of defects in ER-LyLE MCSs and impaired endosomal homeostasis[11,30].

To investigate the function of protrudin in endosome maturation, we search for proteins that associate and function cooperatively with protrudin at MCSs between the ER and LyLEs with the use of a proteomics approach. We thereby identify PDZD8 (PDZ domain-containing 8) as a key binding partner of protrudin. PDZD8 contains a synaptotagmin-like mitochondrial lipid-binding protein (SMP) domain and is a mammalian ortholog of yeast Mmm1, a subunit of the ER-mitochondrial encounter structure (ERMES) complex that serves as a physical connection between the ER and mitochondria and facilitates the transfer of lipids[31,32]. Although PDZD8 has been shown to tether the ER and mitochondria and to regulate calcium dynamics in neurons, it remains unclear whether it tethers the ER and endosomes and possesses lipid transfer activity. We now show that PDZD8 does indeed possess such functions and that it contributes to endosome maturation in cooperation with protrudin at MCSs. In addition, we find that the protrudin-PDZD8 system is essential for the establishment of neuronal polarity and the maintenance of neuronal integrity.

## Results

**Identification of PDZD8 as a protrudin-associated protein by a proteomics approach.** Protrudin is an ER membrane protein that tethers the ER and LyLE membranes and promotes directional endosome trafficking. We performed a proteomics analysis to identify proteins that associate with protrudin. With the use of dual affinity purification with antibodies to the FLAG epitope and nickel-nitrilotriacetic acid (Ni-NTA) agarose, we purified His$_6$-FLAG-tagged protrudin from the membrane fraction of Neuro2A cells stably expressing this protein. Protrudin binding proteins in the final eluate were fractionated by SDS–polyacrylamide gel electrophoresis (PAGE) (Fig. 1a) and digested with trypsin, and the generated peptides were subjected to liquid chromatography and tandem mass spectrometry (LC–MS/MS). A MASCOT search revealed that the most frequently identified protein was PDZD8, a mammalian ortholog of yeast Mmm1, a subunit of the ERMES complex. The protrudin-associated proteins also included VAP-A and VAP-B, both of which were previously shown to interact with protrudin, as well as with other key proteins that contribute to lipid transfer at MCSs[20–23]. We also performed a similar proteomics analysis with immunoprecipitates prepared from brain extracts of wild-type (WT) or protrudin-deficient mice with antibodies to protrudin. In two independent experiments, peptides derived from PDZD8 were most frequently identified (Fig. 1b). Co-immunoprecipitation analysis with a membrane fraction of mouse brain confirmed the interaction between endogenous protrudin and endogenous PDZD8 (Fig. 1c), suggesting that protrudin and PDZD8 form a complex under physiological conditions. We also confirmed specificity of the interaction between FLAG-protrudin and Myc epitope-tagged PDZD8 expressed in HEK293T cells (Supplementary Fig. 1).

To dissect the interaction between protrudin and PDZD8, we produced a series of deletion mutants and examined their association by co-immunoprecipitation analysis (Fig. 1d–f). Deletion mutants of protrudin tagged with FLAG at the NH$_2$-terminus were expressed together with PDZD8 tagged with the hemagglutinin (HA) epitope at its COOH-terminus in HEK293T cells, and only protrudin mutants containing the region spanning residues 62 to 206, which includes the two transmembrane (TM) domains and the HP domain, were found to interact with PDZD8 (Fig. 1d and Supplementary Fig. 2). Similar analysis with PDZD8 deletion mutants tagged with Myc or FLAG epitopes at the COOH-terminus and expressed together with FLAG- or HA-tagged protrudin, respectively, in HEK293T cells revealed that the TM domain, as well as the region between the PDZ and C1 domains of PDZD8 were required for interaction with protrudin (Fig. 1e and Supplementary Fig. 3).

**Protrudin and PDZD8 mutually stabilize each other.** We generated mice deficient in PDZD8 (*Pdzd8*$^{-/-}$ mice) and found that the abundance of protrudin was greatly diminished in the brain of these mice (Fig. 1g and Supplementary Fig. 4a). In a reciprocal manner, the amount of PDZD8 was also reduced in the brain of mice deficient in protrudin (Fig. 1h and Supplementary Fig. 4b). We attempted to reproduce these effects in PC12

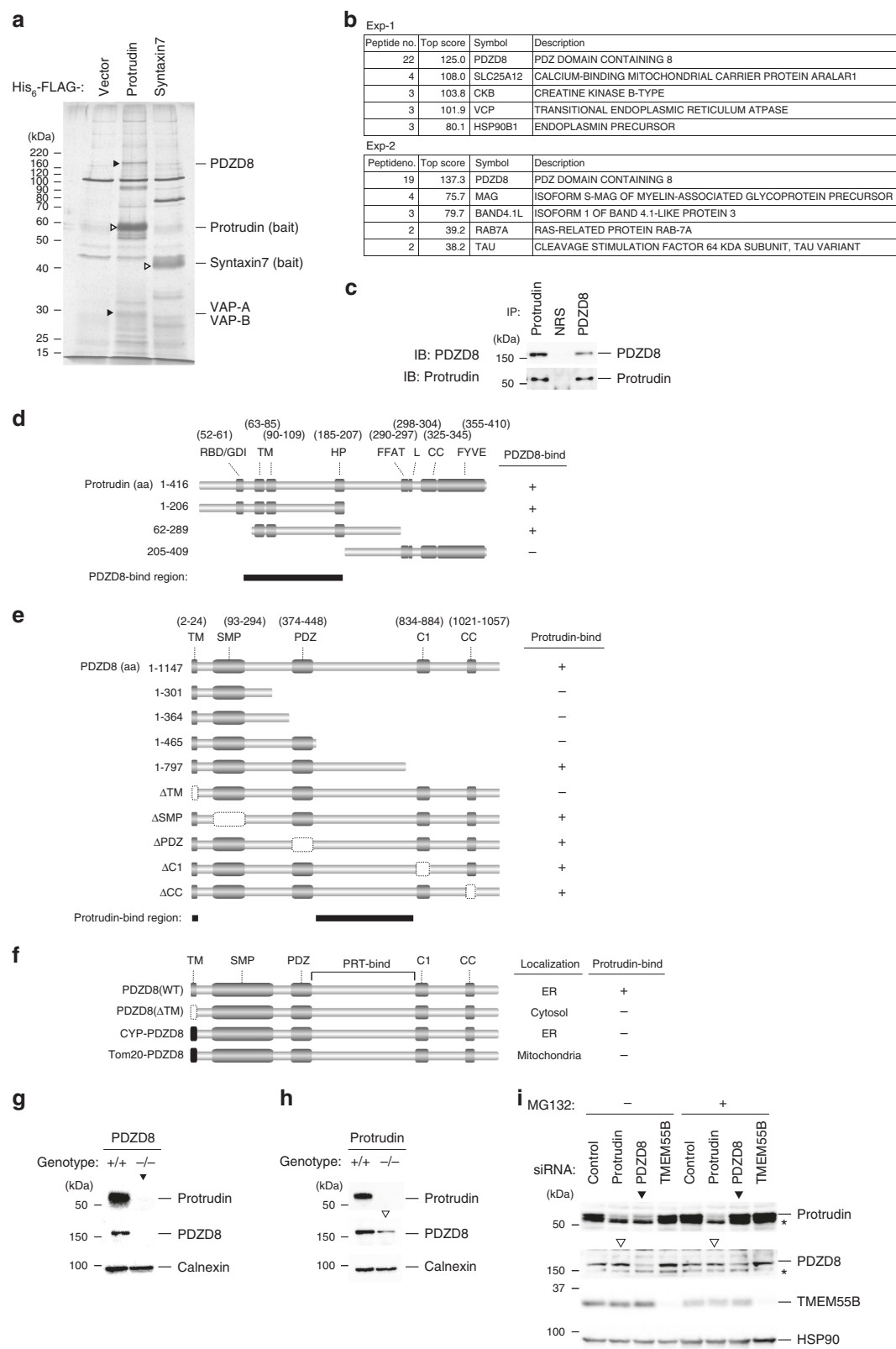

pheochromocytoma cells by small interfering RNA (siRNA)-mediated RNA interference. Knockdown of protrudin mRNA did not affect the abundance of PDZD8 mRNA, and vice versa (Supplementary Fig. 5). On the other hand, depletion of PDZD8 resulted in a reduction in the amount of protrudin at the protein level, with this effect being prevented in the presence of the proteasome inhibitor MG132 (Fig. 1i), suggesting that protrudin was unstable and subjected to proteasomal degradation in the absence of PDZD8. In contrast, the abundance of PDZD8 was not affected by depletion of protrudin in PC12 cells (Fig. 1i). Altogether, these results suggested that protrudin and PDZD8 might be cotranslated and form a stable complex at the ER membrane in the brain, whereas the individual proteins undergo rapid degradation by the proteasome when expressed alone.

**Fig. 1 Identification of protrudin-associated proteins with a proteomics approach. a** Extracts prepared from Neuro2A cells either stably expressing His$_6$-FLAG-tagged mouse protrudin or syntaxin7 (negative control) or infected with the corresponding empty retrovirus (Vector) were subjected to dual affinity purification with antibodies to FLAG and Ni-NTA agarose. The purified proteins were separated by SDS-PAGE and stained with silver. Open and filled arrowheads indicate bands corresponding to bait proteins or protrudin binding proteins, respectively. **b** Extracts prepared from WT mouse brain were subjected to immunoprecipitation with antibodies to protrudin. The identified binding proteins were ranked according to the number of identified peptides from two independent experiments (Exp). **c** Extracts prepared from WT mouse brain were subjected to immunoprecipitation (IP) with antibodies to protrudin or to PDZD8 or with normal rabbit serum (NRS). The resulting precipitates were subjected to immunoblot (IB) analysis with antibodies to protrudin and to PDZD8. **d, e** Domain organization of human protrudin (**d**) and mouse PDZD8 (**e**), and structure of deletion mutants thereof. **f** Domain structure of mouse PDZD8 derivatives in which the TM domain is replaced with the corresponding domain of mouse CYP or Tom20, as well as of WT and ΔTM mutant forms of PDZD8. **g, h** Protein extracts prepared from the brain of WT (+/+) or PDZD8-deficient (−/−) mice (**g**), or from WT (+/+) or protrudin-deficient (−/−) mice (**h**), were subjected to immunoblot analysis with antibodies to protrudin, to PDZD8, and to calnexin (loading control). Filled and open arrowheads indicate bands corresponding to protrudin in PDZD8$^{-/-}$ mice and to PDZD8 in protrudin$^{-/-}$ mice, respectively. **i** PC12 cells transfected with protrudin, PDZD8, TMEM55B (negative control), or scrambled (Control) siRNAs were incubated in the absence or presence of 5 μM MG132 for 6 h. Immunoblot analysis was performed with antibodies to protrudin, to PDZD8, to TMEM55B, and to HSP90 (loading control). Filled and open arrowheads indicate blots corresponding to protrudin in cells transfected with PDZD8 siRNA or to PDZD8 in cells transfected with protrudin siRNA, respectively. The asterisks indicate nonspecific signals. We repeated all experiments at least three times independently with similar results.

**The TM domain of PDZD8 is essential for ER localization and protrudin interaction**. Given that the TM domain of PDZD8 was essential for interaction with protrudin, we examined the specificity of this domain for protrudin. To this end, we replaced the TM domain of PDZD8 with either that of CYP, which localizes to the ER, or that of Tom20, which localizes to mitochondria (Fig. 1f). Immunofluorescence microscopy revealed that, whereas the WT protein was localized to the ER in HeLa cells, a PDZD8 mutant (ΔTM) lacking the TM domain was diffusely distributed throughout the cytosol (Supplementary Fig. 6). The CYP-PDZD8 and Tom20-PDZD8 chimeric proteins localized to the ER and mitochondria, respectively (Supplementary Fig. 7a, b), suggesting that the TM domain primarily determines the localization of these chimeras. Co-immunoprecipitation analysis showed that neither PDZD8(ΔTM) nor the chimeric proteins interacted with protrudin, whereas PDZD8(WT) did so (Supplementary Fig. 7c). Altogether, these results suggested that the TM domain of PDZD8 is specific not only for ER localization but also for association with protrudin. Mere colocalization of protrudin and PDZD8 at the ER was, thus, not sufficient for their interaction (Fig. 1f).

**PDZD8 promotes tethering between the ER and LyLEs**. Protrudin was previously shown to be located at MCSs between the ER and LyLEs[18]. We examined whether PDZD8 might also be localized at ER-LyLE contact sites in neurons, which highly express both protrudin and PDZD8. Immunofluorescence analysis with antibodies to PDZD8, to protein disulfide isomerase (PDI, a marker for the ER), and to lysosome-associated membrane protein 1 (LAMP1, a marker for LyLEs) and with the use of super-resolution microscopy revealed that PDZD8 was indeed located at MCSs between the ER and LyLEs (Fig. 2a, b). PDZD8 colocalized to a markedly greater extent with the ER marker than with the LyLE marker at ER-LyLE MCSs, consistent with the notion that PDZD8 is an ER-resident protein that makes contact with LyLEs (Fig. 2c).

We examined whether PDZD8 also facilitates membrane tethering between the ER and LyLEs, given that it forms a complex with protrudin. HeLa cells stably expressing FLAG-protrudin or PDZD8-Myc were transfected with expression vectors for split-GFP fragments—ERj1-GFP(1–10) and LAMP1-GFP(11)—that contain the retention signals of ERj1 and LAMP1 for the ER and LyLEs, respectively[33]. The fluorescence signal of green fluorescent protein (GFP) is detected only when ERj1-GFP (1–10) and LAMP1-GFP(11) are located in close proximity (Fig. 2d). Similarly, ERj1-GFP(1–10) and Tom70-GFP(11) were applied to detect contact between the ER and mitochondria. We

confirmed that the GFP signal for detection of ER-LyLE or ER-mitochondria MCSs was appropriately localized to the expected organelles (Supplementary Fig. 8a, b), although it is difficult to formally exclude the possibility that these organelle contact sites are artificially formed as a result of the irreversible association of the split-GFP probes. The accuracy of quantification was ensured by normalization of transfection efficiency for each cell on the basis of the fluorescence of mCherry derived from a cotransfected expression vector (Fig. 2d and Supplementary Fig. 8c). Imaging with confocal microscopy revealed that the normalized split-GFP signal for ER-LyLE contacts was greater in cells expressing protrudin or PDZD8 than in control cells, with expression of both molecules having a synergistic effect (Fig. 2e, f). On the other hand, siRNA-mediated depletion of protrudin or PDZD8 in HeLa cells (Supplementary Fig. 9a, b) significantly attenuated MCS formation between the ER and LyLEs (Fig. 2g). We also examined the effect of expression of the PDZD8(ΔC1) mutant, which lacks the C1 domain required for membrane tethering (see below), or the CYP-PDZD8 chimera that is not able to form a complex with protrudin on the formation of MCSs between the ER and LyLEs (Fig. 2h). The PDZD8(ΔC1) mutant was found to have lost the ability to promote the formation of MCSs between the ER and LyLEs, whereas the CYP-PDZD8 chimera fully retained this ability. These results suggested that, although protrudin and PDZD8 are expressed in proximity to each other and cooperate to promote MCS formation between the ER and LyLEs, direct molecular interaction between the two proteins might not be necessary for this effect.

Protrudin and PDZD8 also promoted tethering between the ER and mitochondria (Fig. 2i, j), with this effect of PDZD8 being consistent with previous results[32]. Depletion of protrudin or PDZD8 had a smaller effect on MCS formation between the ER and mitochondria (Fig. 2k) compared with that on ER-LyLE contacts (Fig. 2g). These results suggested that endogenous protrudin and PDZD8 indeed contribute to MCS formation, with the formation of ER-LyLE contacts being more dependent on these proteins than that of ER-mitochondria MCSs.

**PDZD8 possesses lipid extraction activity**. The major functions of MCSs include mediation of lipid transfer, calcium ion homeostasis, and organelle dynamics, and PDZD8 has been shown to play an essential role in calcium ion homeostasis[32]. Given that PDZD8 contains an SMP domain, which is present in the TULIP (tubular lipid-binding protein) superfamily of proteins that possess lipid transfer activity[34–36], and that the yeast PDZD8 ortholog Mmm1 mediates interorganellar lipid transfer between membranes in the ERMES complex, we examined whether

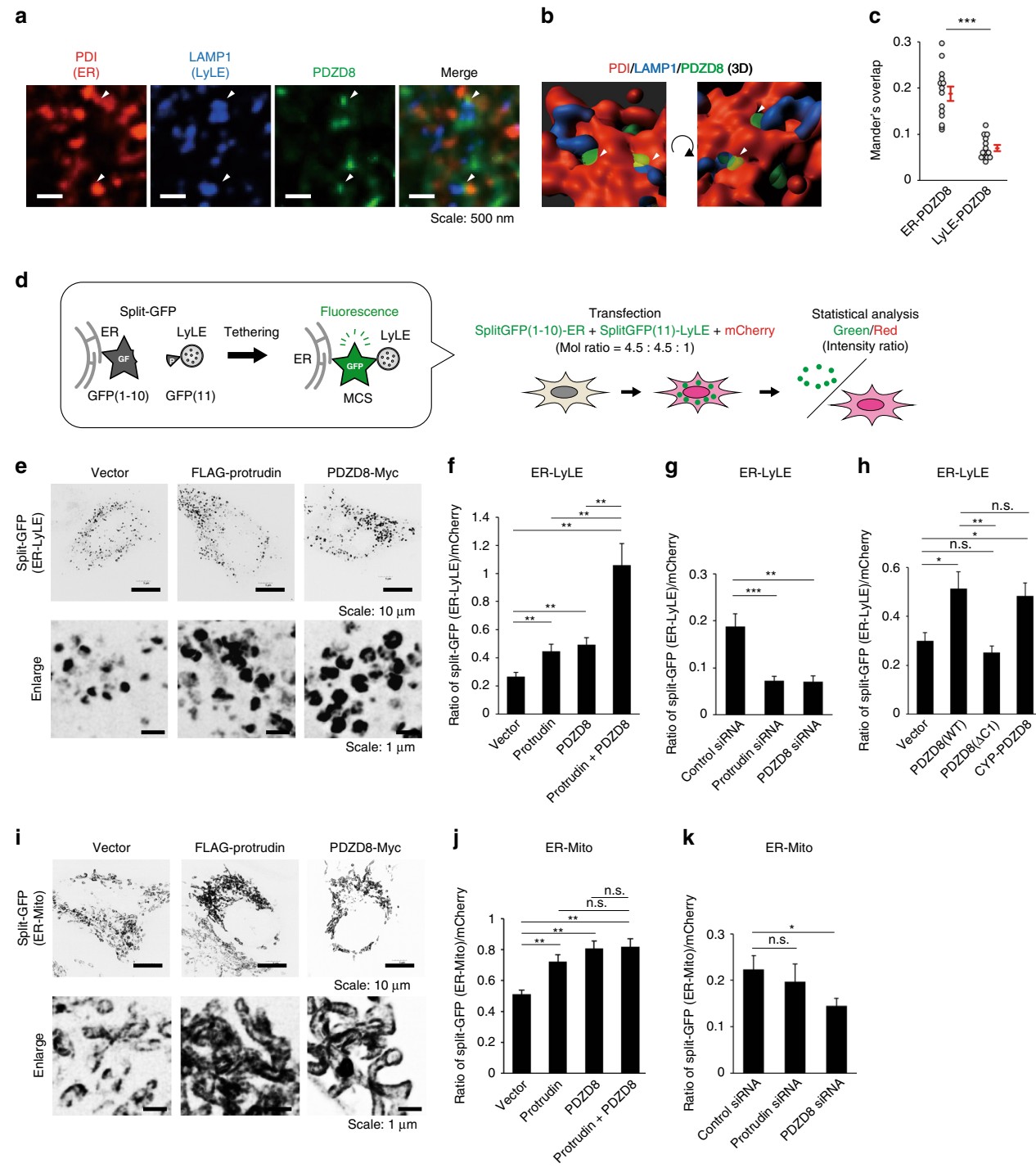

PDZD8 also manifests lipid transfer activity. We prepared a series of recombinant PDZD8 derivatives tagged with glutathione S-transferase (GST) at the $NH_2$-terminus by bacterial expression (Fig. 3a). Given that recombinant PDZD8(WT) was insoluble, we eliminated the TM domain from all the recombinant proteins in order to reduce their hydrophobicity. Lipid transfer activity was measured with the liposome-FRET (fluorescence resonance energy transfer) assay as described previously[37]. The presence of rhodamine-labeled lipid in close proximity to 7- nitrobenz-2-oxa-l,3-diazol-4-yl (NBD)-labeled lipid in the same donor liposome, thus, results in the quenching of NBD fluorescence by FRET, whereas this quenching is abolished if lipids are extracted from the donor liposome and dispersed (Fig. 3b). The lipid extraction

activity of GST-PDZD8 derivatives was determined from the measured fluorescence intensity of NBD after subtraction of that observed in the presence of GST as a control (Supplementary Fig. 10a–c). Phospholipids—including phosphatidic acid (PA), phosphatidylserine (PS), phosphatidylethanolamine (PE), and phosphatidylcholine (PC)—were indeed extracted by PDZD8 (ΔTM) (Fig. 3c), with the initial velocity of lipid extraction being greatest for PS (Fig. 3e and Supplementary Fig. 10d). PDZD8 (ΔTM) also showed high extraction activity for ceramide and low extraction activity for cholesterol in terms of lipid extraction velocity (Fig. 3d, e and Supplementary Fig. 10d).

Lipid transfer comprises two steps, lipid extraction from the donor membrane and its insertion into the acceptor membrane.

**Fig. 2 Effects of protrudin and PDZD8 on tethering between the ER and LyLEs or mitochondria. a** Mouse primary cortical neurons cultured for 5 days in vitro were fixed, stained with antibodies to the ER marker PDI, to the LyLE marker LAMP1, and to PDZD8, and observed by super-resolution microscopy. Scale bars, 500 nm. **b** Rotated three-dimensional (3D) images derived from (**a**). Arrowheads in **a** and **b** indicate the localization of PDZD8 at ER-LyLE MCSs. **c** Manders overlap coefficient for colocalization of PDZD8 with either PDI (ER) or LAMP1 (LyLE) determined from images as in **a**. Data are means + SE for 14 cells. ***$P < 0.001$ (Student's $t$-test). **d** Schematic representation of the mechanism of action of split-GFP (enclosed, left), and strategy for statistical analysis (right). A representative image is shown in Supplementary Fig. 8c. **e, i** HeLa cells stably expressing FLAG-tagged mouse protrudin or Myc-tagged mouse PDZD8, or those infected with the corresponding empty retrovirus (Vector), were transfected with expression vectors for split-GFP fragments—ERj1-GFP(1–10) and either LAMP1-GFP(11) (**e**) or Tom70-GFP(11) (**i**)—as well as for mCherry and were then imaged by confocal fluorescence microscopy. Scale bars, 10 μm. Enlarged images are also shown. Scale bars, 1 μm. **f, j** Split-GFP analysis for the ER and either LyLEs (data are means ± SE for 19 to 22 independent cells) (**f**) or mitochondria (Mito) (data are means ± SE for 44 to 52 independent cells) (**j**) of HeLa cells infected with retroviruses for protrudin, PDZD8, or both proteins. **g, k** Split-GFP analysis for the ER and either LyLEs (data are means ± SE for 18 independent cells) (**g**) or mitochondria (data are means ± SE for 16 to 18 independent cells) (**k**) of HeLa cells transfected with control, protrudin, or PDZD8 siRNAs. **h** Split-GFP analysis for the ER and LyLEs of HeLa cells infected with retroviruses for PDZD8(WT), PDZD8(ΔC1), or CYP-PDZD8 (data are means ± SE for 19 to 28 independent cells). *$P < 0.05$, **$P < 0.01$, ***$P < 0.001$; n.s., not significant (Kruskal–Wallis test followed by the Steel-Dwass multiple comparison test). The mRNA levels for endogenous human and exogenous mouse protrudin and PDZD8 are shown in Supplementary Fig. 9. We repeated all experiments at least three times independently with similar results.

We therefore examined whether PDZD8 contributes to only lipid extraction or to both lipid extraction and insertion by comparing extraction activities measured with PDZD8(ΔTM) in the absence or presence of acceptor liposomes. The measured activities did not differ substantially between these two conditions (Fig. 3f), suggesting that PDZD8 possesses only lipid extraction activity and that the direction of lipid transfer mediated by PDZD8 is one-way from the ER to other organelles. The initial velocity of lipid extraction by PDZD8(ΔTM) increased in a protein concentration-dependent manner (Fig. 3g, h and Supplementary Fig. 10d). We confirmed that lipids extracted by PDZD8 were dispersed in the solution in this assay by showing that NBD fluorescence intensity did not differ substantially between before and after removal of liposomes by ultracentrifugation (Fig. 3i).

**The SMP and PDZ domains of PDZD8 are responsible for lipid extraction.** A series of deletion mutants of PDZD8 was tested for the ability to extract PA, PS, PE, and PC (Fig. 4a). PDZD8(SMP), which consists of only the SMP domain, showed minimal lipid extraction activity. PS extraction activity was decreased to ~50% of that for PDZD8(ΔTM) by deletion of either the SMP (ΔTMΔSMP) or PDZ (ΔTMΔPDZ) domain. The activity of PDZD8(SMP-PDZ) was also about half that of PDZD8 (ΔTM). Two mutants (ΔTMΔPRT and ΔTMΔC1) showed an activity similar to that of PDZD8(ΔTM) (Fig. 4b, c and Supplementary Fig. 10d). These results suggested that the SMP and PDZ domains contribute to the lipid extraction activity of PDZD8 (Fig. 4a).

The observation that the lipid extraction reaction slows down and, thus, manifests a biphasic time course was likely the result of saturation of PDZD8 in the solution with lipid. To examine this possibility, we performed the assay with PDZD8(ΔTM) tagged with hexahistidine and liposomes containing DGS-NTA(Ni), which mediates docking of $His_6$-tagged proteins (Fig. 4d). Under these conditions, the lipid extraction reaction showed a monophasic time course, suggesting that membrane anchoring of PDZD8 by the TM domain influences lipid extraction activity (Fig. 4e, f and Supplementary Fig. 10d). The measured activities also did not differ substantially in the absence or presence of acceptor liposomes, again suggesting that PDZD8 possesses only lipid extraction activity.

**The C1 domain of PDZD8 specifically binds to PS and PtdIns (4)P.** PDZD8 contains a C1 domain, which has been shown to preferentially bind to diacylglycerol (DAG)[38]. We therefore examined the lipid-binding ability of a recombinant protein

consisting of only the C1 domain. Recombinant GST-PDZD8 (C1) was incubated with strips on which a series of lipids had been spotted, and the strips were then probed with antibodies to GST. Among the lipids tested, PDZD8(C1) preferentially interacted with PS and phosphatidylinositol 4-phosphate [PtdIns(4)P] in a lipid concentration-dependent manner (Fig. 5a–c and Supplementary Fig. 11). Unexpectedly, however, PDZD8(C1) did not interact with DAG in this assay. The interaction with PS was verified with a liposome-binding assay (Fig. 5d, e), revealing that GST-PDZD8(C1) was more efficiently moved to the pellet fraction from the supernatant fraction by binding to liposomes containing PS than by that to control liposomes without PS. Given that deletion of the C1 domain of PDZD8(ΔTM) did not affect extraction activity for PS (Fig. 4a–c), binding of the C1 domain to PS is independent of lipid extraction activity. By analogy to extended synaptotagmins (E-Syts)[39–42], the C1 domain of PDZD8 might function to tether the ER and LyLE membranes by interaction with PS and PtdIns(4)P enriched in the LyLE membrane, with this interaction possibly being regulated by an intracellular signal. It is of note that lipid extraction activity was increased by loss of the coiled-coil (CC) domain of PDZD8(ΔTM) (Fig. 5f, g and Supplementary Fig. 10d), suggesting that the CC domain might inhibit the lipid extraction activity of PDZD8. A conformational change induced by an unknown signal may, thus, increase the lipid extraction activity of the SMP domain of PDZD8 by promoting the binding of the C1 domain to LyLE lipids and consequent ER-LyLE tethering (Fig. 5h).

**Coexpression of protrudin and PDZD8 promotes formation of abnormal large vacuoles in HeLa cells.** We found that coexpression of protrudin and PDZD8 by transient transfection induced the formation of abnormal large vacuoles (ALVs) in HeLa cells (Fig. 6a, c), with protrudin and PDZD8 being colocalized at the vacuolar membrane (Fig. 6b). Expression of PDZD8 alone did not induce any marked morphological changes in organelles of HeLa cells, whereas expression of protrudin alone induced ALV formation to a lesser extent than did that of both proteins (Fig. 6a, c). Immunofluorescence analysis revealed that the ALVs were positive for calreticulin, Rab7, Rab9, and LAMP1, but not for Tom20 (Fig. 6d), suggesting that they originated from the ER and LyLEs. HeLa cells stably expressing exogenous protrudin and PDZD8 at a relatively low level compared with that achieved by transient transfection did not manifest ALV formation (Supplementary Fig. 9c–f and Supplementary Fig. 12).

We next examined the effects of expression of a series of PDZD8 mutants together with protrudin on ALV formation in transiently transfected HeLa cells (Fig. 6e, f). The PDZD8(ΔSMP)

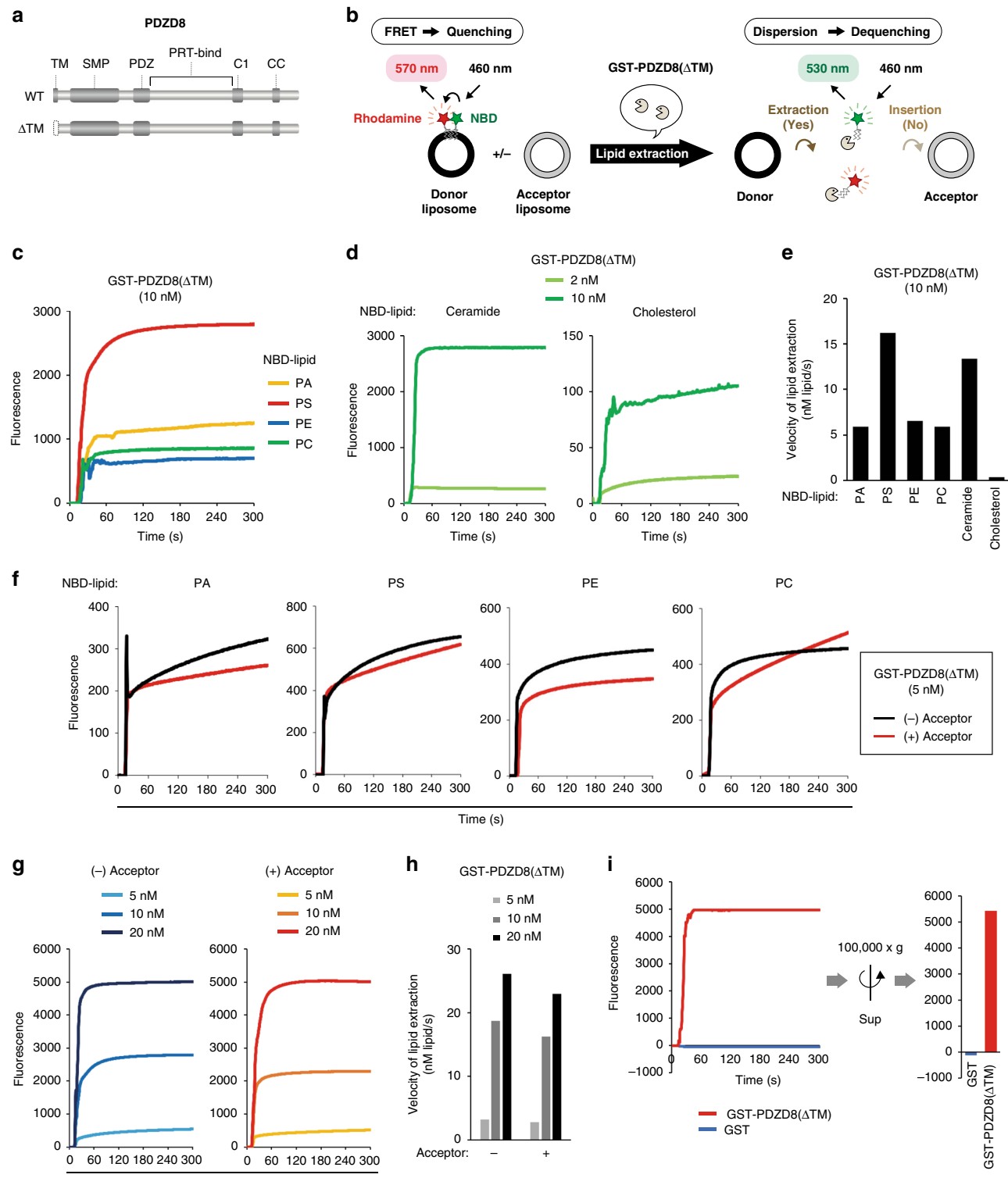

**Fig. 3 Lipid extraction activity of PDZD8. a** Domain structure of mouse PDZD8(WT) and PDZD8(ΔTM). **b** Schematic representation of the liposome-FRET assay. **c** Phospholipid (PA, PS, PE, and PC) extraction activities of GST-PDZD8(ΔTM). **d** Ceramide (left) or cholesterol (right) extraction activity of GST-PDZD8(ΔTM). **e** Initial velocity of lipid extraction by GST-PDZD8(ΔTM) measured in **c** and **d**). **f** Phospholipid extraction activities of GST-PDZD8 (ΔTM) with PA, PS, PE, and PC in the absence (−) or presence (+) of acceptor liposomes. **g** Concentration dependence for lipid extraction activity of GST-PDZD8(ΔTM) with PS in the absence (left) or presence (right) of acceptor liposomes. **h** Initial velocity of lipid extraction by GST-PDZD8(ΔTM) measured in **g**. **i** Detection of lipid (PS) extracted by GST-PDZD8(ΔTM) in the supernatant (Sup) separated from liposomes in the pellet after ultracentrifugation of the reaction mixture at 300 s.

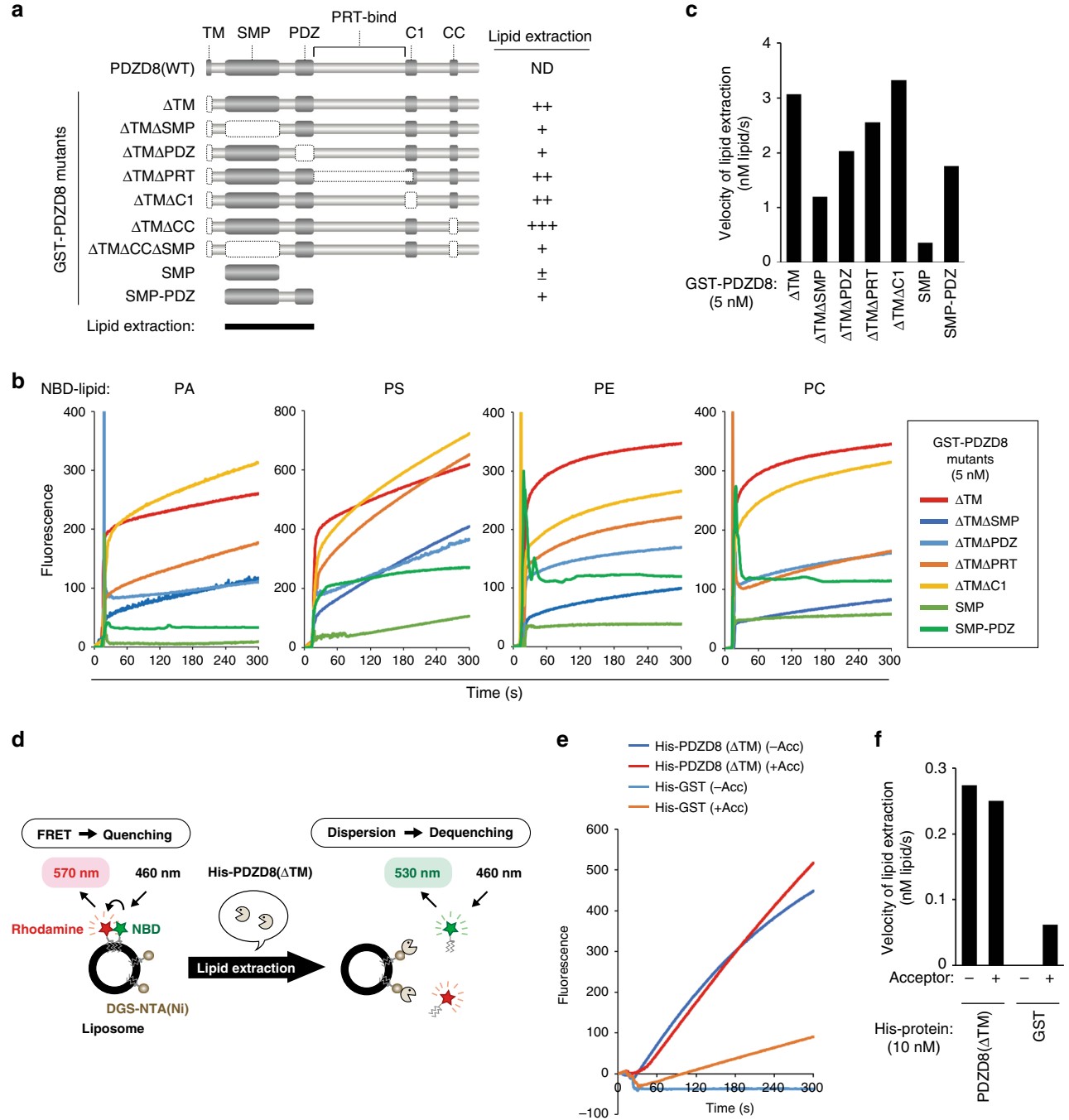

**Fig. 4 Delineation of the region of PDZD8 responsible for lipid extraction. a** Domain structure of mouse PDZD8 deletion mutants. A summary of the lipid extraction activity of the mutants is shown on the right. PRT, protrudin; ND, not determined. **b** Phospholipid extraction activities of PDZD8 deletion mutants with PA, PS, PE, and PC. **c** Initial velocity of lipid extraction by PDZD8 mutants with PS measured in **b**. **d** Schematic representation of the liposome-FRET assay performed with DGS-NTA(Ni) and His₆-tagged proteins. **e** PS extraction activity of His₆-PDZD8(ΔTM) or His₆-GST (negative control) measured with donor liposomes containing DGS-NTA(Ni) and in the absence (−) or presence (+) of acceptor (Acc) liposomes. **f** Initial velocity of lipid extraction by His₆-PDZD8(ΔTM) measured in **e**.

mutant, which has a reduced lipid extraction activity, did not induce the formation of ALVs, suggesting that lipid extraction mediated by PDZD8 is required for this effect. The PDZD8(ΔC1) mutant was partially defective in the ability to induce formation of ALVs, suggestive of the importance of the interaction between the C1 domain and lipids such as PS and PtdIns(4)P for this ability. In contrast, the CYP-PDZD8 chimera induced ALV formation to a similar extent as did PDZD8(WT), consistent with the results of the split-GFP assay for ER-LyLE contact (Fig. 2h).

To investigate the detailed structure of the ALVs, we examined HeLa cells expressing GFP-protrudin and PDZD8-GFP by transmission electron microscopy (TEM). The ALVs contained large, irregular intraluminal structures and a few regular ILVs (Fig. 6g, h), suggesting that they were defective MVBs formed during aberrant endosome maturation. We also detected many abnormal MVBs containing typical ILVs and irregular multi-lammelar structures prior to apparent ALV formation (Fig. 6i). In addition, many ER membrane segments were detected around the

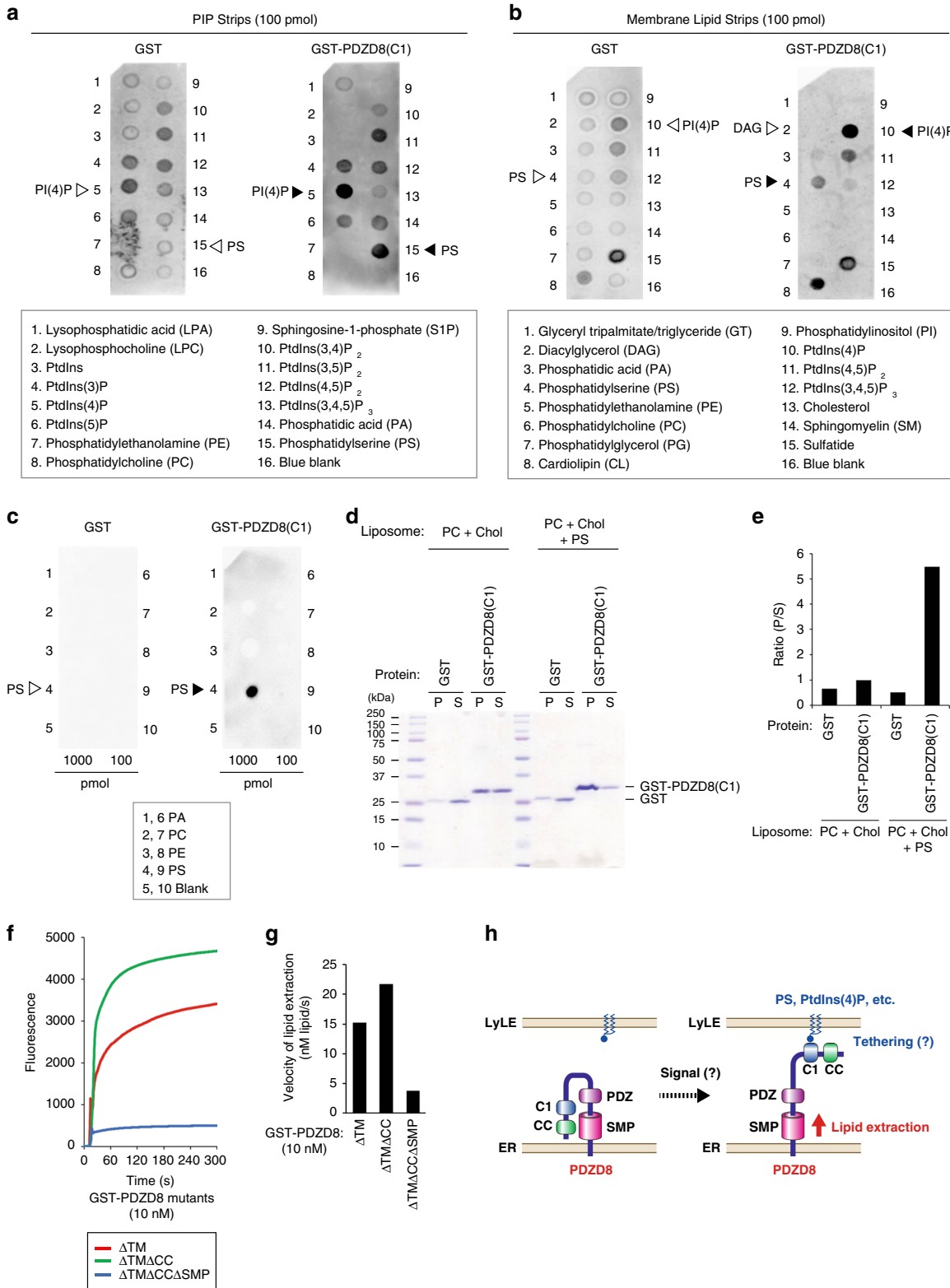

ALVs (Fig. 6i), suggesting that increased ER-LyLE contact induced by expression of protrudin and PDZD8 promoted ER accumulation around LyLEs. The ALVs differed in appearance from autophagosomes containing dense particles (ribosomes) in the lumen. Focused ion beam-scanning electron microscopy (FIB-SEM) also revealed that the ALVs differed in size from LyLEs present in control cells and that they contained large, irregular intraluminal structures (Fig. 6j, k), again suggesting

that they were defective MVBs. In addition, FIB-SEM analysis revealed that the ALVs were connected to each other, consistent with TEM observations indicative of a fission defect of LyLEs during endosome maturation (Fig. 6l). These findings, thus, suggested that excessive amounts of protrudin and PDZD8 cooperatively promote artifactual formation of ALVs as a result of their excess tethering of the ER and LyLEs and subsequent excess lipid extraction from the ER and its transfer to LyLEs. This lipid

**Fig. 5 Lipid-binding activity of PDZD8(C1) and mode of action of PDZD8. a–c** Recombinant GST or GST-PDZD8(C1) was incubated with PIP Strips (**a**), Membrane Lipid Strips (**b**), or custom lipid strips (**c**), after which binding of the proteins to lipid spots on the strips was probed with antibodies to GST and immunoblot reagents. The lipids on the strips are listed below. **d, e** Recombinant GST or GST-PDZD8(C1) was incubated with liposomes consisting of PC and cholesterol (Chol), with or without PS, after which the liposomes were isolated by centrifugation and the resulting pellet (P) and supernatant (S) were subjected to SDS-PAGE followed by staining with Coomassie brilliant blue (**d**) and quantitative analysis of protein amount in each fraction (**e**). **f** Phospholipid extraction activity of the indicated PDZD8 deletion mutants with PS. **g** Initial velocity of lipid extraction by the mutants measured in **f**. **h** Model for the possible mode of action of PDZD8. PDZD8 is integrated into the ER membrane via its TM domain and tethers the LyLE membrane by binding to PS or PtdIns(4)P via its C1 domain. The CC domain inhibits lipid extraction activity of PDZD8. We repeated all experiments at least three times independently with similar results.

extraction likely supplies the membrane components that allow the enlargement of LyLEs, resulting in the development of ALVs instead of normal MVBs.

**The protrudin-PDZD8 axis is essential for endosome maturation.** The amounts of endogenous protrudin and PDZD8 mRNAs are higher in the brain than in HeLa cells (Supplementary Fig. 9g, h), suggesting that both proteins might also play a role in endosomal maturation in neurons. Depletion of protrudin or PDZD8 in mouse primary neurons indeed induced frequent gross enlargement of LAMP1$^+$ LyLEs (Fig. 7a, b). This phenotype of PDZD8 depletion was rescued by additional expression of siRNA-resistant PDZD8(WT) but not by that of the lipid extraction-deficient mutant PDZD8(ΔSMP) (Fig. 7c, d). A similar phenotype was observed in mouse primary neurons expressing PDZD8(ΔSMP) in the presence of endogenous PDZD8, likely as a result of a dominant-negative effect of the mutant protein on the normal fission of LyLEs (Fig. 7e, f). The enlarged LyLEs showed an abnormal, multilamellar ultrastructure with almost no sign of normal ILVs (Fig. 7g) and were highly reminiscent of those apparent in neurons of spastin or REEP1 mutant mice[11]. Forced coexpression of protrudin and PDZD8 in mouse primary neurons did not induce enlargement of LyLEs (Supplementary Fig. 13), likely because the neurons express the endogenous proteins at a much higher level compared with HeLa cells and may be resistant to a further increase in expression level. We next examined whether PDZD8 promotes lipid transport to endosomes in vivo. To this end, mouse primary neurons expressing the PS reporter protein EGFP-Lact-C2 and the LyLE marker protein mCherry-CD63 were subjected to siRNA-mediated depletion of PDZD8. The extent of colocalization of LyLEs and PS was much lower in the neurons depleted of PDZD8 than in control neurons (Fig. 7h, i), suggesting that PDZD8 indeed mediates the transfer of lipid from the ER to endosomes in vivo.

**The protrudin-PDZD8 system is required for neuronal polarity and integrity.** We investigated the potential role of protrudin and PDZD8 in neuronal development. Mouse primary neurons depleted of protrudin or PDZD8 showed a reduced axon length and increased somatodendrite area (Fig. 8), with both of these effects reflecting impairment of cell polarity. These morphological abnormalities became more evident with the progression of neuronal development and were reminiscent of those apparent in neurons derived from protrudin-deficient mice[22]. These results, thus, suggested that MCS formation and subsequent lipid transfer dependent on the protrudin-PDZD8 system are essential for endosomal maturation and the establishment of cell polarity in neurons.

Defects in ER-LyLE contacts induced by HSP-associated mutations of spastin or REEP1 were previously shown to result in failure of lysosomal fission in mouse primary neurons[11]. The facts that mutations of the human protrudin gene (*ZFYVE27*) also cause HSP and that protrudin, spastin, and REEP1 are all

ER-resident proteins that contain an HP domain also suggested that the protrudin-PDZD8 system might contribute to maintenance of neuronal integrity. Mouse primary neurons depleted of protrudin or PDZD8 indeed manifested a morphology consistent with axonal degeneration, including axonal thinning, as well as a reduced level and interrupted pattern of staining for the axonal marker Tau1 (Fig. 9a). We examined colocalization of Tau1 and microtubules visualized by immunofluorescence staining for α-tubulin, given that Tau1 has been shown to dissociate from microtubules during axonal degeneration[43]. The extent of colocalization of Tau1 and α-tubulin was indeed reduced by depletion of protrudin or PDZD8 in mouse primary neurons (Fig. 9b, c), suggesting that the protrudin-PDZD8 system protects neurons from axonal degeneration and is essential for neuronal integrity.

**Discussion**

We initially discovered protrudin as an ER-resident protein that possesses the ability to promote neurite formation through interaction with the GDP-bound form of the small GTPase Rab11[16]. Protrudin was also shown to regulate endosome trafficking through interaction with the GTP-bound form of Rab7, suggesting that it might control the direction of endosome sorting and endosome maturation by changing its partner Rab proteins in complex with GTP or GDP[16,18,44]. We have now identified PDZD8 as a key binding partner of protrudin and shown that the protrudin-PDZD8 system promotes the formation of MCSs between the ER and LyLEs. Importantly, we found that PDZD8 possesses lipid extraction activity that might be fundamental to endosome maturation. Furthermore, endosome maturation governed by the protrudin-PDZD8 system appeared to be essential for maintenance of neuronal integrity and the establishment of neuronal polarity, with its dysfunction giving rise to impaired endosomal homeostasis and neurodegeneration. We thus propose a model for endosome maturation regulated by the protrudin-PDZD8 complex (Fig. 10). Given that protrudin also interacts with VAP, KIF5, and Rab proteins, it appears to serve as a scaffold for membrane tethering, the regulation of endosomal trafficking, and lipid supply from the ER to LyLEs, with the protrudin-PDZD8 system promoting endosomal maturation. The localization of both protrudin and PDZD8 to the ER membrane suggests that they are not likely to affect MCS formation between endosomes and mitochondria. However, given that Rab7 was shown to contribute to anchoring between endosomes and mitochondria[45] and that protrudin binds to Rab7, it is possible that the protrudin-PDZD8 system is indirectly involved in such anchoring.

Forced expression of protrudin and PDZD8 at a high level in HeLa cells induced the formation of grossly expanded LAMP1$^+$ ALVs containing atypical intraluminal structures, with these vesicles being highly similar to those apparent in protrudin- or PDZD8-depleted mouse primary neurons. This phenotype in HeLa cells is thus likely attributable to an artifact of over-expression, which results in defective endosomal fission as a result

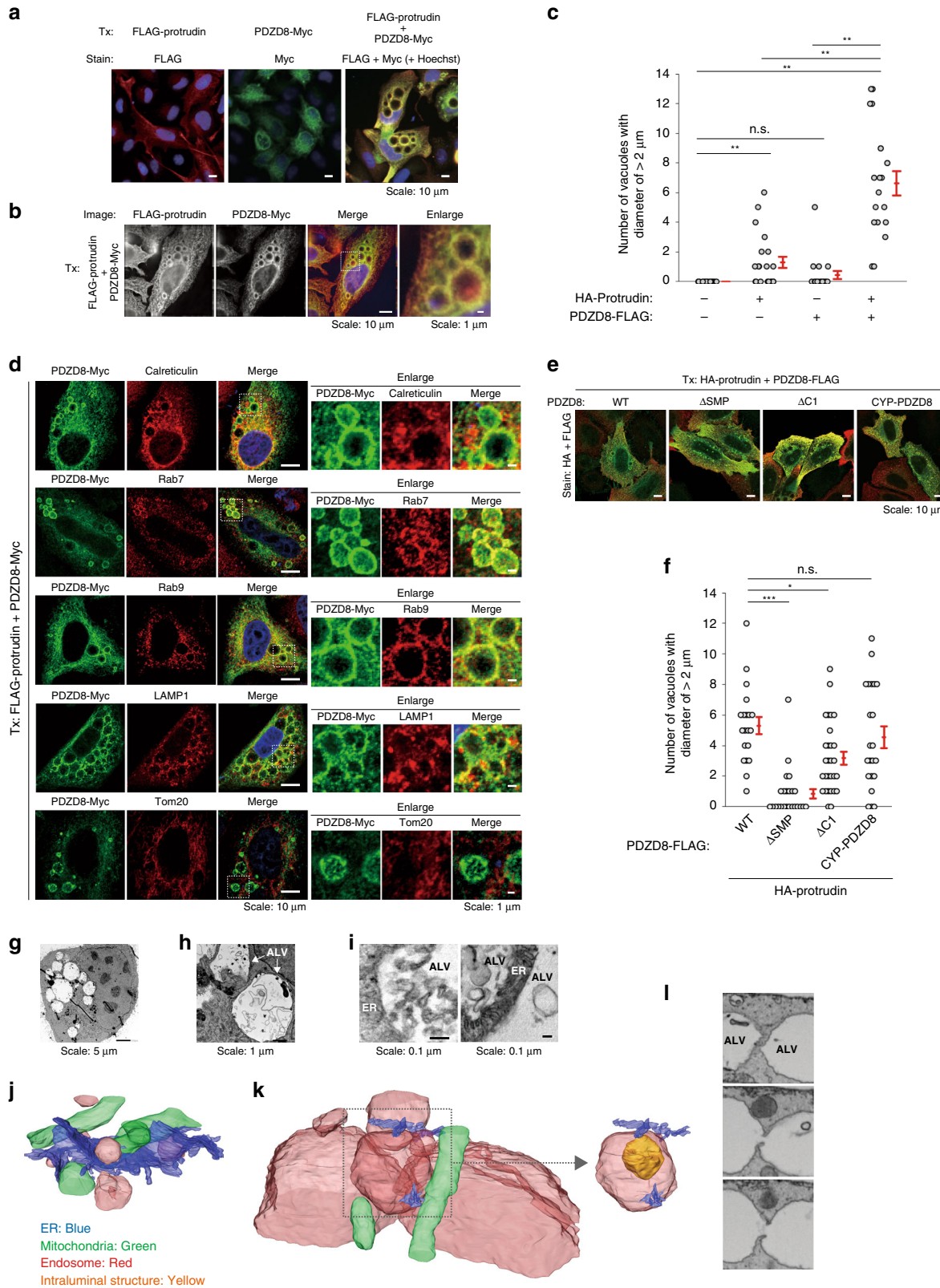

of an impaired balance of endosomal homeostasis (Supplementary Text, Supplementary Figs. 14–17). Indeed, we observed many ALVs linked together, likely as a result of defective LyLE fission. An abnormal increase in MCS formation between the ER and LyLEs and a consequent excessive transfer of lipid at such sites might, thus, impair endosome fission and maturation. HeLa cells stably expressing exogenous protrudin and PDZD8 at a relatively low level did not manifest the grossly enlarged ALVs.

Only a few proteins have been identified in the SMP-like family, including yeast Mmm1, Mdm12, and Mdm34 of the ERMES complex, Tricalbins, and Nvj2, as well as mammalian E-Syts and Tex2[31,39]. Mammalian TMEM24, which also contains a

**Fig. 6 Abnormal large vacuole (ALV) formation induced by coexpression of protrudin and PDZD8. a, b** HeLa cells transiently transfected (Tx) with expression vectors for FLAG-protrudin, PDZD8-Myc, or both proteins were stained with antibodies to FLAG (red) and to Myc (green). Nuclei were also stained with Hoechst 33258 (blue). Scale bars, 10 or 1 μm as indicated. **c** Quantification of the number of vacuoles with a diameter of >2 μm in HeLa cells transiently transfected with expression vectors for HA-protrudin, PDZD8-FLAG, or both proteins. Data are means ± SE for 19 to 21 independent cells. **P < 0.01 (Kruskal–Wallis test followed by the Steel-Dwass multiple comparison test). **d** HeLa cells transiently transfected with expression vectors for FLAG-protrudin and PDZD8-Myc were stained with antibodies to Myc (green) as well as with those to the organelle marker proteins (red) calreticulin (ER), Rab7 or Rab9 (LEs), LAMP1 (LyLEs), or Tom20 (mitochondria). Merged images include nuclear staining with Hoechst 33258 (blue). Scale bars, 10 μm. The boxed regions are shown enlarged on the right. Scale bars, 1 μm. **e, f** HeLa cells transiently transfected with expression vectors for HA-protrudin as well as WT or the indicated mutant forms of PDZD8-FLAG were stained with antibodies to HA (red) and to FLAG (green) (**e**). Scale bar, 10 μm. The number of vacuoles with a diameter of >2 μm was quantitated (**f**). Data are means + SE for 20 to 29 independent cells. *P < 0.05, ***P < 0.001 (Kruskal–Wallis test followed by the Steel-Dwass multiple comparison test). **g–i** HeLa cells transiently transfected with expression vectors for GFP-protrudin and PDZD8-GFP were imaged by TEM. Images of low (**g**), medium (**h**), or high (**i**) magnification are shown (**i**). Scale bars: 5 μm (**g**), 1 μm (**h**), or 0.1 μm (**i**). **j, k** Organelles of control cells (**j**) or of HeLa cells coexpressing FLAG-protrudin and PDZD8-Myc (**k**) were examined by FIB-SEM. The LE with an intraluminal structure in the boxed region of (**k**) is shown on the right. **l** Series of images along the z-axis used for FIB-SEM of HeLa cells coexpressing FLAG-protrudin and PDZD8-Myc. We repeated all experiments at least three times independently with similar results.

SMP domain, was also shown to mediate lipid transport at MCSs between the ER and plasma membrane[46]. SMP domain-containing proteins belong to the TULIP superfamily, members of which are highly hydrophobic and possess the ability to transfer lipids such as phospholipids, ceramide, or cholesterol between membranes[34,36,47]. These proteins commonly consist of a long helix wrapped by antiparallel β sheets that enclose a lipophilic cavity. Two different models of lipid transfer by the SMP domain—the tunnel model and the shuttle model—have been proposed[42]. In both models, the proteins assemble into multimers to form an extended structure that spans two organelles, and some of them have indeed been shown to be located at MCSs and to mediate interorganelle lipid exchange. Despite the structural similarity of mammalian PDZD8 to yeast Mmm1, lipid transfer activity had been identified for the latter but not the former. We have now shown that PDZD8 promotes the formation of MCSs between the ER and LyLEs and also mediates lipid transfer from the ER to LyLEs. In terms of its domain structure, PDZD8 is similar to TMEM24, E-Syts, and Tex2, given that they all contain TM, SMP, and lipid-binding domains such as C1, C2, or PH (pleckstrin homology) that interact with acceptor membranes[41,46]. The $NH_2$-terminal TM domain of E-Syts was shown to be recruited to the donor ER membrane, and the COOH-terminal C2 domain to interact with $PtdIns(4,5)P_2$ in the acceptor plasma membrane in response to a calcium signal. By analogy, PDZD8 is likely recruited to the ER by its TM domain, and its C1 domain likely interacts with PS and PtdIns(4)P in the LyLE membrane in response to an unknown signal. It is of note that PDZD8 was found to possess only lipid extraction activity; it did not show the ability to insert lipids into acceptor liposomes by itself. Other proteins associated with PDZD8 might, thus, contribute to insertion activity, with such a complex containing PDZD8 possessing the ability both to extract lipid from the donor membrane and to insert it into the acceptor membrane. Given that the NBD-labeled lipids used in this study possess bulky hydrophilic groups that might affect the binding of PDZD8 or their solubility, the assay results should be verified with the use of isotopically labeled lipids.

Among ER proteins located at MCSs with other organelles, VAP-A and VAP-B play a key role in MCS function. VAPs possess an MSP (major sperm protein) domain, which interacts with the FFAT (two phenylalanines in an acidic tract) motif, and most FFAT motif-containing proteins are associated with lipid transfer proteins. Protrudin contains an FFAT motif and interacts with VAP proteins as well as with PDZD8, which we have now shown possesses lipid extraction activity. The VAP-protrudin-PDZD8 complex might be essential for maintenance of neural integrity, given that mutations of the genes for each of these proteins are associated with neural diseases. Protrudin is an ER-shaping protein with an HP domain, and its gene (ZFYVE27) is mutated in a subset of individuals with HSP, with protrudin also being referred to as spastic paraplegia (SPG) 33[17,28]. Protrudin interacts with other HSP-related proteins, including myelin proteolipid protein 1 (SPG2), atlastin 1 (SPG3A), REEP1 (SPG31), REEP5 (similar to REEP1), KIF5A (SPG10), KIF5B, KIF5C, and reticulons 1, 3, and 4 (which are similar to reticulon 2, SPG12), most of which also harbor an HP domain[24,26,48,49]. Given that ER-shaping proteins play a role in tethering or lipid exchange between the ER and other organelles at MCSs by shaping the ER membrane, protrudin may promote PDZD8-mediated lipid transfer by increasing ER membrane curvature. It is of interest that spastin and REEP1 contribute to MCS formation in neurons and that neurons of mice with mutations of these proteins manifest a phenotype similar to that of protrudin- or PDZD8-depleted neurons. Given the molecular functional and structural similarities among spastin, REEP1, and protrudin as well as the contributions of these proteins to the pathogenesis of HSP, we conclude that MCS formation and lipid transfer activity mediated by the protrudin-PDZD8 system are essential for endosomal maturation in the physiological setting.

Mutation of the gene encoding VAP-B (ALS8) gives rise to amyotrophic lateral sclerosis[50], whereas that of the PDZD8 gene is a major risk factor for posttraumatic stress disorder[51]. PDZD8 has also been associated with virus infection, such as that by human immunodeficiency virus or herpes simplex virus, with this association possibly being related to PDZD8 function in endosome sorting of viral particles in human cells[52,53]. Further characterization of the VAP-protrudin-PDZD8 complex should contribute to our understanding not only of the physiological roles of this complex but also of its contribution to the pathogenesis of various neural diseases.

## Methods

**Plasmids.** Construction of vectors encoding human or mouse protrudin was described previously[16]. Vectors for split-GFP analysis—pcDNA3.1_ERj1(1–200)_GFP(1–10), pcDNA3.1_Lamp1_HA_GFP(11), and pcDNA3.1_Tom70(1–70)_GFP(11)—were also described previously[33]. Mouse Complementary DNAs (cDNAs) encoding PDZD8 (GenBank accession number CH466585.1), siRNA-resistant PDZD8 containing synonymous mutations of 936T > A and 951T > A, the C2 domain (nucleotides 811–1281) of Lactadherin (Lact-C2) (GenBank accession number NM_001045489), syntaxin7 (GenBank accession number NM_016797.5), KIF5A (GenBank accession number NM_001039000), the proteasome subunit S4 (GenBank accession number NM_001330212), or Cul1 (GenBank accession number NM_003592) were amplified from cDNA of C57BL/6J mouse brain by the polymerase chain reaction (PCR) with PrimeSTAR HS DNA polymerase (Takara, Shiga, Japan) and specific primers. A human cDNA encoding CD63 (GenBank accession number NM_001780) was amplified from HeLa cell cDNA by PCR. The cDNAs encoding mutant versions of these various proteins were also generated by PCR. Amplified cDNA fragments were subcloned into pcDNA3 or pcDNA3.1-

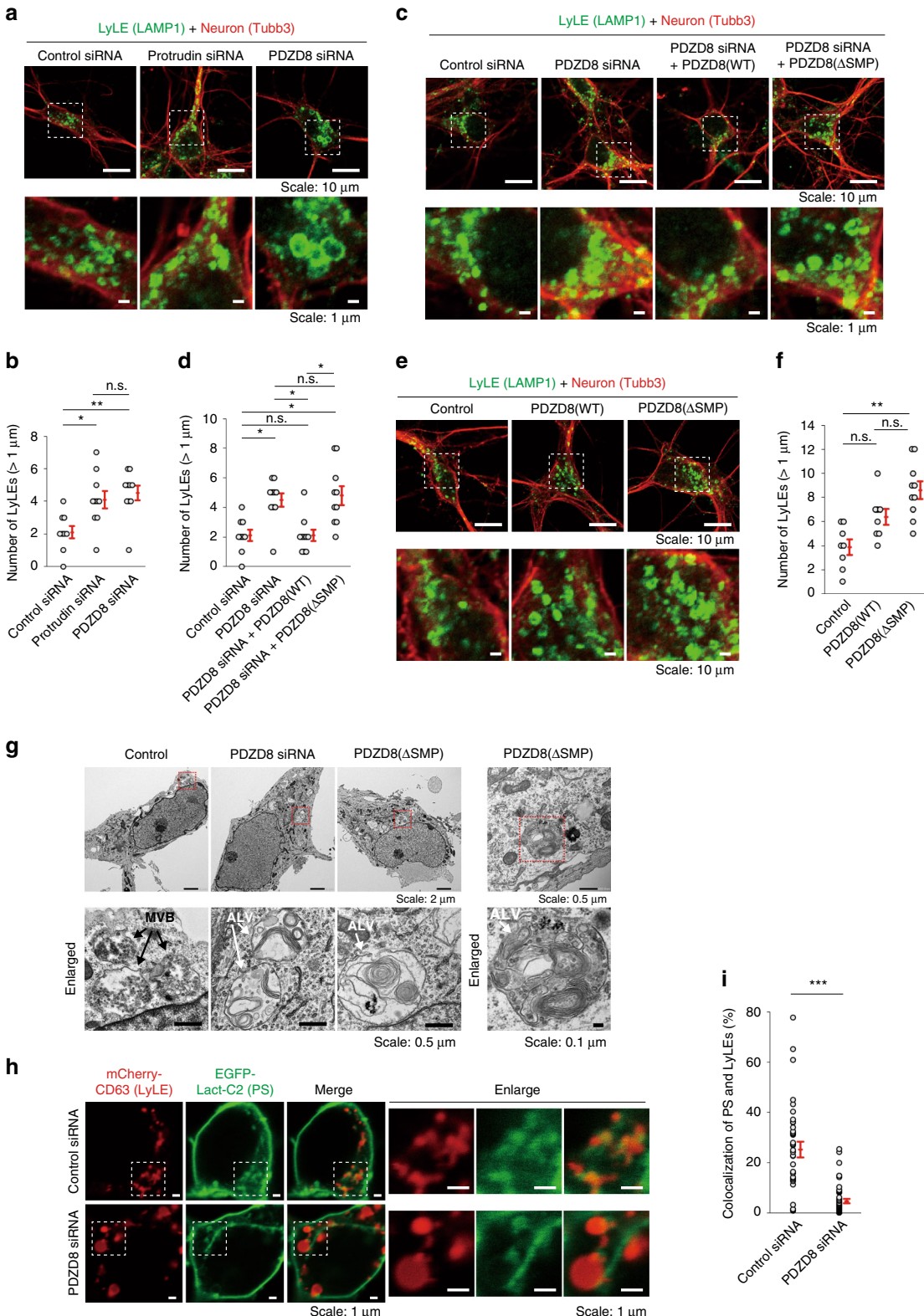

Myc-His (Invitrogen Life Technologies, Carlsbad, CA), p3×FLAG-CMV7.1 (Sigma-Aldrich, St. Louis, MO), pGEX-6P (GE Healthcare, Little Chalfont, UK), pET30 (Novagen, Darmstadt, Germany), pEF-BOS-2×HA (kindly provided by S. Nagata and H. Sumimoto, Kyoto University, Japan), or pMX-puro (kindly provided by T. Kitamura, Tokyo University, Japan). The EGFP and mCherry vectors were from Clontech (Mountain View, CA).

**Antibodies**. For proteomics analysis and other immunoprecipitation, a mouse monoclonal antibody to protrudin was generated by Kurabo (Osaka, Japan) with

the use of a His₆-tagged recombinant protein comprising amino acids 212 to 323 of mouse protrudin-L[22] that had been expressed in and purified from *Escherichia coli*. Antibodies to protrudin (for immunoblot analysis) and to TMEM55B were obtained from ProteinTech (Chicago, IL); those to HSP90 were from BD Biosciences (San Jose, CA); those to FLAG (mouse monoclonal M2 for immunoprecipitation, immunoblot, and immunofluorescence analyses and rabbit polyclonal for immunoblot analysis), to the Myc epitope (9E10), to Tau1, and to Tubb3 were from Sigma-Aldrich; those to HA (HA.11) were from Covance (Princeton, NJ); those to calreticulin were from Stressgen (Victoria, British Columbia, Canada);

**Fig. 7 Effects of PDZD8 on LyLE maturation in neurons. a–f** Mouse primary cortical neurons transfected with control, protrudin, or PDZD8 siRNAs (**a**), with PDZD8 siRNA and expression vectors for siRNA-resistant forms of FLAG-tagged PDZD8(WT) or PDZD8(ΔSMP) (**c**), or with expression vectors for PDZD8(WT) or PDZD8(ΔSMP) (**e**) were cultured for 9 days in vitro, fixed, stained with antibodies to the neuronal marker Tubb3 and to the LyLE marker LAMP1, and observed with a confocal fluorescence microscope. Scale bars, 10 μm. The boxed regions in the upper panels are shown enlarged below. Scale bars, 1 μm. The number of LyLEs with a diameter of >1 μm in images as in (**a**), (**c**), and (**e**) are quantified in (**b**), (**d**), and (**f**), respectively. Data are means + SE for 8 to 10 independent cells. *P < 0.05, **P < 0.01 (Kruskal–Wallis test followed by the Steel-Dwass multiple comparison test). **g** Neurons transfected with PDZD8 siRNA or with an expression vector for PDZD8(ΔSMP) were cultured for 7 days in vitro, fixed, embedded in epoxy resin, and imaged by TEM. The boxed regions in the upper panels are shown at higher magnification in the lower panels. ALVs containing both ILVs and irregular intraluminal structures in cells expressing PDZD8(ΔSMP) are shown in the right panels. Scale bars are as indicated. **h** Neurons transfected with expression vectors for mCherry-CD63 and EGFP-Lact-C2, as well as with control or PDZD8 siRNAs were observed with a confocal fluorescence microscope after 6 days in vitro. (N = 44 to 52 for independent cells). Scale bars, 1 μm. The boxed regions in the left group of panels are shown enlarged on the right. Scale bars, 1 μm. **i** Quantification of the mCherry/EGFP colocalization ratio in images as in **h**. Data are means + SE for 36 to 53 independent regions. ***P < 0.001 (Student's t-test). We repeated all experiments at least three times independently with similar results.

those to PDZD8 were from LSBio (Seattle, WA); those to Rab7, to Rab9, to LC3, and to LAMP1 (for HeLa cells) were from Cell Signaling Technology (Beverly, MA); those to GST were from Frontier Institute (Hokkaido, Japan); those to HA (Y-11), to Tom20, and to LAMP1 (for neurons) were from Santa Cruz Biotechnology (Santa Cruz, CA); those to Map2 were from Abcam (Cambridge, MA); those to calnexin were from Enzo Life Sciences (Farmingdale, NY); those to α-tubulin (TU01) were from Thermo Fisher Scientific (Waltham, MA); and those to EEA1 were from BD Transduction Laboratories (Franklin Lakes, NJ). Alexa Fluor 488- or Alexa Fluor 546-conjugated goat antibodies to mouse or rabbit immunoglobulin G (IgG) were obtained from Molecular Probes (Eugene, OR).

**Cell culture, transfection, and retrovirus infection**. Neuro2A, HEK293T, HeLa, PC12, and Plat-E cells were cultured under a humidified atmosphere of 5% CO$_2$ at 37 °C in Dulbecco's modified Eagle's medium (Invitrogen Life Technologies) supplemented with 10% fetal bovine serum (Invitrogen Life Technologies). The culture medium for Plat-E cells was also supplemented with blasticidin (10 μg/ml). Cells were transfected with the use of the FugeneHD or XtremeGene9 reagents (Roche, Mannheim, Germany) for 24 or 48 h. For retroviral infection, Plat-E cells were transiently transfected with pMX-puro-based vectors and then cultured for 48 h. The retroviruses in the resulting culture supernatants were used to infect Neuro2A cells, and the cells were then subjected to selection with puromycin (1 μg/ml). HeLa cells were transiently transfected with the pCL-Ampho vector (Novus Biologicals, Centennial, CO) and pMX-puro-based vectors and then cultured for 48 h. The retroviruses in the resulting culture supernatants were used to infect HeLa cells, and the cells were then subjected to selection with puromycin (2 μg/ml). Magic red was obtained from Immunochemistry Technologies (Bloomington, MN).

**Mouse primary neurons**. Neurons were isolated from the cerebral cortex of Jcl-ICR mouse embryos at 18 days postcoitum and were dispersed with a Nerve-Cell Culture System (Sumitomo, Tokyo, Japan) and cultured in Nerve-Cell Culture Medium (Sumitomo) on dishes coated with poly-L-lysine (Sigma-Aldrich). Primary neurons were transfected with the use of a Nucleofector system (Lonza, Basel, Switzerland).

**Membrane fraction preparation and protein extraction**. Neuro2A cells or whole mouse brain were homogenized with a Dounce or Potter homogenizer, respectively, in a solution containing 20 mM HEPES-NaOH (pH 7.6), 0.25 M (cells) or 0.32 M (brain) sucrose, 1 mM EDTA, 1 mM Na$_3$VO$_4$, 25 mM NaF, aprotinin (10 μg/ml), leupeptin (10 μg/ml), and 1 mM phenylmethylsulfonyl fluoride. The homogenate was centrifuged twice at 500 × g for 5 min at 4 °C to remove nuclei and nondisrupted cells, and the resulting supernatant was centrifuged at 100,000 × g for 1 h at 4 °C to isolate a membrane fraction (pellet). This pellet was solubilized in a lysis buffer (40 mM HEPES-NaOH [pH 7.5], 150 mM NaCl, 10% glycerol, 0.5% Triton X-100, 1 mM Na$_3$VO$_4$, 25 mM NaF, aprotinin [10 μg/ml], leupeptin [10 μg/ml], 1 mM phenylmethylsulfonyl fluoride), incubated for 1 h at 4 °C, and then centrifuged again at 20,400 × g for 10 min at 4 °C to remove insoluble material. The protein concentration of the extract was determined with a Pierce BCA protein assay kit (Thermo Fisher Scientific).

**Identification of protrudin-associated proteins isolated by dual affinity purification**. The membrane fraction of Neuro2A cells expressing His$_6$-FLAG-tagged mouse protrudin was subjected to affinity purification with anti-FLAG (M2)-agarose affinity gel (Sigma-Aldrich), and the material eluted with FLAG peptide (Sigma-Aldrich) was then subjected to affinity purification with Ni-NTA agarose (ProBond resin, Invitrogen Life Technologies). Proteins eluted with imidazole were concentrated by precipitation with chloroform-methanol, fractionated by SDS-PAGE, and stained with silver. The membrane fraction of mouse brain was subjected to immunoprecipitation with mouse monoclonal antibodies to protrudin, and the resulting immunoprecipitates were fractionated by SDS-PAGE and stained

with silver. The stained gels were sliced into pieces, and the abundant proteins therein were subjected to in-gel digestion with trypsin. The resulting peptides from Neuro2A cells and mouse brain were dried, dissolved in a mixture of 0.1% tri-fluoroacetic acid and 2% acetonitrile, and then applied to a nanoflow LC system (Paradigm MS4; Michrom BioResources, Auburn, CA) equipped with an L-column (C18, 0.15 by 50 mm, particle size of 3 μm; CERI, Tokyo, Japan). Nanoscale LC (nanoLC)–MS/MS analysis was performed with a system consisting of a Q Exactive Plus mass spectrometer (Thermo Fisher Scientific) coupled with a nanoLC instrument (Advance, Michrom BioResources). All MS/MS spectra were compared with protein sequences in the International Protein Index (IPI, European Bioinformatics Institute) mouse version 3.44 with the use of the MASCOT algorithm. Assigned high-scoring peptide sequences (MASCOT score of ≥35) were considered for correct identification. Identified peptides from independent experiments were integrated and regrouped by IPI accession number. For the mouse brain experiments, the peptides identified in protrudin knockout mice were subtracted from those identified in WT mice.

**Immunoprecipitation and immunoblot analysis**. Immunoprecipitation and immunoblot analysis were performed as described previously[54]. In brief, protein extracts were subjected to immunoprecipitation for 1 h at 4 °C with primary antibodies and protein G-Sepharose 4 Fast Flow (Amersham Biosciences, Uppsala, Sweden). The immunoprecipitates were washed three times with cell lysis buffer and then subjected to immunoblot analysis. Images of blots were scanned with a LAS-4000 instrument (GE Healthcare).

**Generation of mutant mice**. For generation of protrudin-deficient mice, genomic DNA corresponding to the *Zfyve27* (protrudin gene) locus was isolated from a 129/Sv mouse genomic DNA library. The targeting vector was constructed by replacing exons 1 to 2 of *Zfyve27* (which contain the start codon) with a cassette containing the mouse phosphoglucokinase (PGK) gene promoter, the neomycin phosphotransferase gene (*neo*), and a poly(A) sequence. Mouse embryonic stem (ES) cells (129/Sv strain) were subjected to electroporation with the linearized targeting vector followed by positive (G418) and negative (ganciclovir) selection. For confirmation of the homologous recombination event, DNA prepared from PCR-positive clones was subjected to Southern blot analysis. ES cells heterozygous for the targeted mutation were microinjected into C57BL/6J mouse blastocysts, which were then implanted into pseudopregnant ICR female mice. The resulting male chimeras were mated with C57BL/6J females, and germ-line transmission of the mutant allele was confirmed by PCR and Southern blot analyses. Heterozygous offspring were intercrossed to produce homozygous mutant animals.

For generation of PDZD8-deficient mice, Cas9 nickase mRNA (IDT 1074181, Alt-R S.p. Cas9 Nuclease 3NLS; IDT, San Jose, CA), Generic tracrRNA (IDT 1072532, Alt-R CRISPR-Cas9 tracrRNA), and target-specific crRNA (IDT, Custom Alt-R CRISPR crRNA) were mixed and injected into C57BL/6J mouse zygotes with the use of a Super Electroporator NEPA21 Type II (Nepagene, Tokyo, Japan). The 5′ (sgRNA-1) and 3′ (sgRNA-2) sgRNAs (single-guide RNAs) with 20-nucleotide sequences targeted to exon 1 (which contains the start codon) of the PDZD8 gene and containing PAM (protospacer-adjacent motif) sequences were 5′-GCAGGC CGAGGGTTGCGGCGGGG-3′ and 5′-GCAGATTCCCAGCACGACCCTGG-3′, respectively. Potential predicted off-target sites were checked for at http://crispr.mit.edu. The resultant mutant mice were backcrossed with C57BL/6J mice before experiments. Phenotypes other than those described in the present study will be published elsewhere. All mouse experiments were approved by the animal ethics committees of Kyushu University and Nagoya City University.

**RNA interference as well as RT and real-time PCR analysis**. Stealth siRNAs targeted to rat protrudin, to rat PDZD8, to human protrudin, to human PDZD8, to mouse protrudin, and to mouse PDZD8 were obtained from Invitrogen Life Technologies. The sequence of each siRNA are shown in Supplementary Table 1. Human siRNAs were introduced into HeLa cells with the use of Lipofectamine

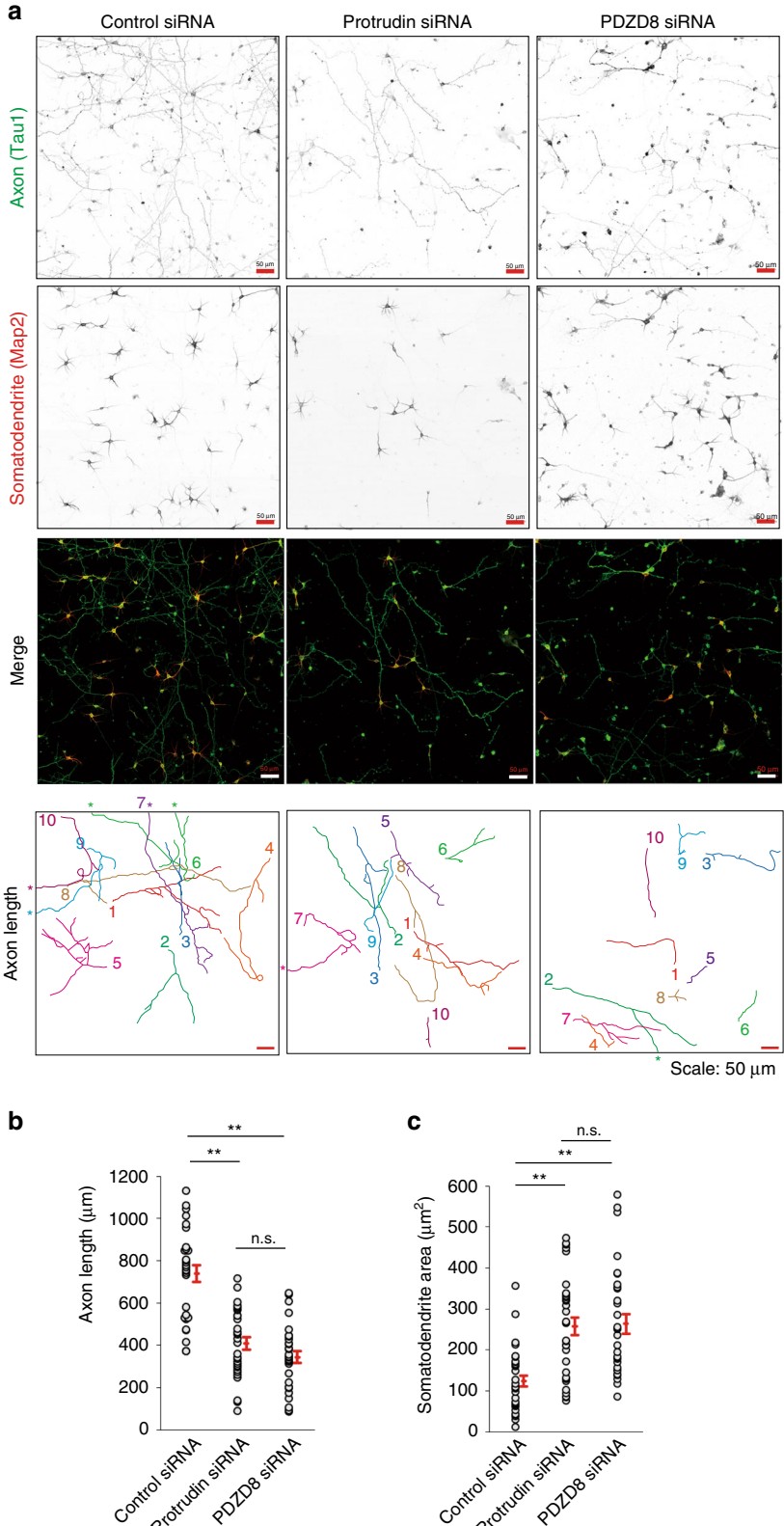

**Fig. 8 Effects of protrudin and PDZD8 on neuronal polarity. a** Mouse primary cortical neurons transfected with control, protrudin, or PDZD8 siRNAs were cultured in vitro for 5 days, fixed, stained with antibodies to the axonal marker Tau1 and to the somatodendrite marker Map2, and observed with a confocal fluorescence microscope. Axons were traced and are distinguished by number and color for measurement of their length (lowest panels). Asterisks indicate axons that extend beyond the area shown. Scale bars, 50 μm. **b, c** Axon length (**b**) and somatodendrite area (**c**) determined from images as in **a** with the use of the ImageJ-based program Fiji. Data are means + SE for 31 to 33 independent cells. *$P < 0.05$, **$P < 0.01$ (Kruskal–Wallis test followed by the Steel-Dwass multiple comparison test). We repeated all experiments at least three times independently with similar results.

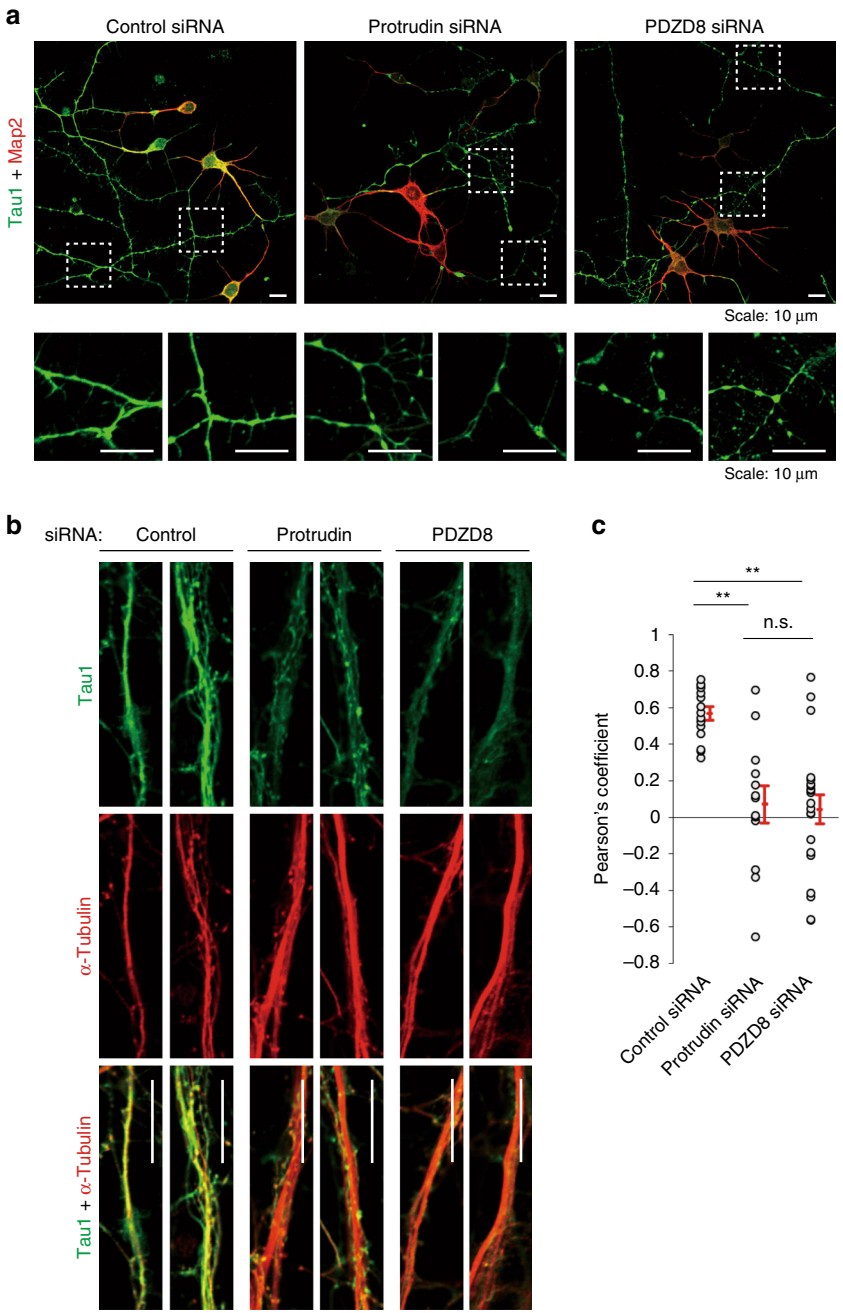

**Fig. 9 Effects of protrudin and PDZD8 on neuronal integrity. a** Mouse primary cortical neurons transfected with control, protrudin, or PDZD8 siRNAs were cultured for 4 days in vitro, fixed, stained with antibodies to Tau1 and to Map2, and observed with a confocal fluorescence microscope. Scale bars, 10 μm. The boxed regions of the upper panels are shown enlarged below. Scale bars, 10 μm. **b** Neurons transfected as in **a** were cultured for 7 days in vitro, fixed, stained with antibodies to Tau1 and to α-tubulin, and observed with a confocal fluorescence microscope. Enlarged images of axons are shown. Scale bars, 10 μm. **c** Pearson's colocalization coefficient for Tau1 and α-tubulin in images as in **b**. Data are means + SE for 13 to 21 independent regions. **$P < 0.01$ (Kruskal–Wallis test followed by the Steel-Dwass multiple comparison test). We repeated all experiments at least three times independently with similar results.

RNAiMAX (Invitrogen Life Technologies), and rat and mouse siRNAs were introduced into PC12 cells and mouse primary neurons, respectively, with the use of a Nucleofector instrument (Lonza). The cells were then cultured for 72 h. Total RNA was isolated from cells with the use of an RNeasy Kit (Qiagen, Hilden, Germany) and was subjected to reverse transcription (RT) with the use of a QuantiTect Rev Transcription Kit (Qiagen). The resulting cDNA was subjected to real-time PCR analysis with Power SYBR Green PCR Master Mix in an ABI-Prism 7000 sequence detection system (Applied Biosystems, Foster City, CA). The amounts of each target mRNA were calculated and normalized by that of GAPDH mRNA as described previously. Primers for PCR are shown in Supplementary Table 2.

**Immunostaining**. Immunostaining was performed as described previously[55]. HeLa cells or primary neurons cultured on glass cover slips were fixed for 10 min at room temperature with 3.7% formaldehyde in phosphate-buffered saline (PBS) and then incubated consecutively with primary antibodies and Alexa Fluor 488- or Alexa Fluor 546-labeled secondary antibodies in PBS containing 0.5% Triton X-100. Nuclei were also stained with Hoechst 33258 (Wako, Tokyo, Japan) as indicated. The cells were finally covered with a drop of GEL/MOUNT (Biomeda, Hayward, CA) and observed with an Olympus IX-81 fluorescence microscope (Olympus, Tokyo, Japan) or LSM800 confocal microscope (Zeiss, Tokyo, Japan).

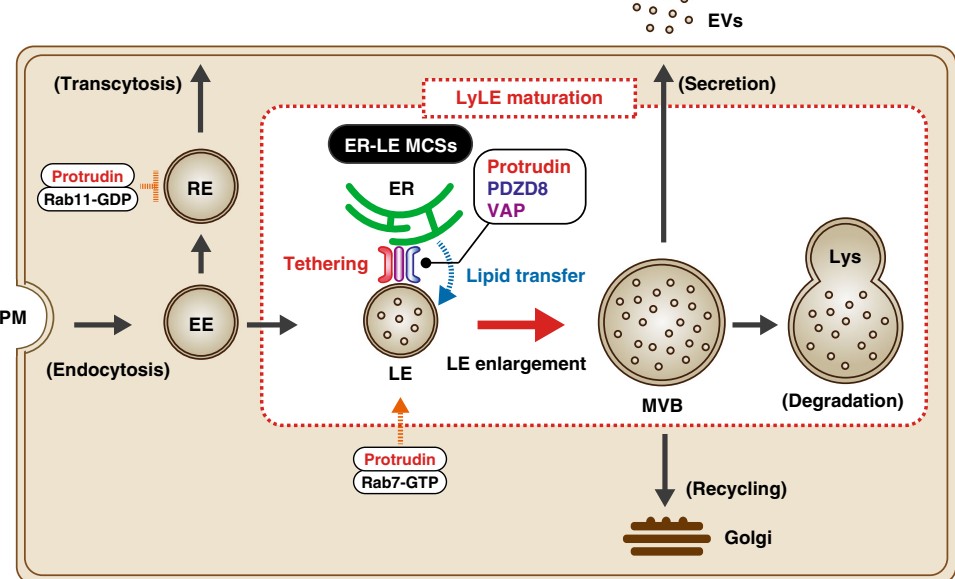

**Fig. 10 Protrudin and PDZD8 promote endosome maturation at ER-LyLE MCSs.** LEs undergo maturation to form MVBs through an increase in size and the formation of ILVs. This process is promoted by tethering of LyLEs to the ER via a VAP-protrudin- PDZD8 complex, with PDZD8 mediating a transfer of lipid from the ER to LyLEs. EVs, extracellular vesicles; Lys lysosome, PM plasma membrane, RE recycling endosome.

**Transmission electron microscopy.** Cells were fixed for 30 min to 1 h with 2% glutaraldehyde in 0.05 M cacodylate buffer containing 5 mM $CaCl_2$. They were washed in cacodylate buffer with $CaCl_2$, exposed to buffered 1% $OsO_4$ plus 0.8% $K_4[Fe(CN)_6] \cdot 3H_2O$ for 1 to 2 h, dehydrated with acetone or ethanol, and embedded in Epon-Araldite or Spurr's medium. Thin sections were stained with uranyl acetate and lead citrate and observed with a JEM-1400 Plus instrument (JEOL, Tokyo, Japan).

**Focused ion beam-scanning electron microscopy.** Cells were fixed at room temperature with Karnovsky fixative (2% paraformaldehyde, 2.5% glutaraldehyde, and 2 mM $CaCl_2$ in 0.1 M cacodylate buffer [pH 7.3]) and then placed on ice for 15 min. After five washes with cacodylate buffer, the specimens were further fixed for 30 min at 4 °C with 1.5% potassium ferrocyanide and 2% $OsO_4$ in 0.1 M cacodylate buffer and then washed five more times with water. The specimens were incubated for 1 h at 60 °C with 1% thiocarbohydrazide, washed five times with water, incubated with 2% $OsO_4$ in water, and washed five times with water. They were then subjected to en bloc staining by incubation overnight with 4% uranyl acetate in distilled water. After three washes with water, the specimens were immersed in Walton's lead aspartate solution, dehydrated with increasing concentrations of ethanol (5 min each in 20%, 50%, 70%, 80%, 90%, 100%, and 100%), and infiltrated with epoxy resin mixture (Epon812; TAAB, Reading, UK). The resin was then polymerized by incubation for 72 h at 65 °C, after which the resin blocks in plastic dishes were immersed in toluene to remove the dishes. The remaining resin-embedded specimens were placed overnight in an oven at 60 °C before analysis with a Quanta3D FEG instrument (Thermo Fisher Scientific).

**Super-resolution imaging.** A z-series of super-resolution images was acquired with a super-resolution spinning-disk confocal microscope (SpinSR10, Olympus) and was subjected to deconvolution and calculation of Manders overlap with imaging software (cellSens, Olympus). The images were also visualized with three-dimensional image construction software (Imaris, Zeiss).

**Expression and purification of recombinant proteins.** Recombinant GST- or $His_6$-tagged proteins were expressed in and purified from *E. coli*. The BL21(DE3) pLysS bacterial cells were transformed with pGEX-6P- or pET30-based vectors, cultured, and then exposed to 0.5 mM isopropyl-β-D-thiogalactopyranoside for 16 h at 10 °C. The cells were then subjected to ultrasonic treatment, and the soluble fraction of cell lysates was purified. The expressed GST-tagged proteins were purified with glutathione-Sepharose 4B beads (Amersham Biosciences) and were eluted from the beads with reduced glutathione. The expressed $His_6$-tagged proteins were purified with Ni-NTA agarose (Wako) and were eluted with imidazole (Wako). Purified proteins were concentrated with an Amicon Ultra device (Merck Millipore, Tokyo, Japan) and dialyzed against PBS with a Mini Dialysis Kit (GE Healthcare).

**Liposome-FRET assay.** All lipids were obtained from Avanti (Sigma-Aldrich): POPC (16:0-18:1), 1-palmitoyl-2-oleoyl-sn-glycero-3-phosphocholine; POPE (16:0-18:1), 1-palmitoyl-2-oleoyl-sn-glycero-3-phosphoethanolamine; rhodamine-

DPPE (18:1 Liss Rhod-PE), 1,2-dioleoyl-sn-glycero-3-phosphoethanolamine-*N*-(lissamine rhodamine B sulfonyl); NBD-PS (16:0-12:0), 1-palmitoyl-2-{12-[(7-nitro-2-1,3-benzoxadiazol-4-yl)amino]dodecanoyl}-sn-glycero-3-phosphoserine (ammonium salt); NBD-PE (18:1-12:0), 1-oleoyl-2-{12-[(7-nitro-2-1,3-benzoxadiazol-4-yl)amino]dodecanoyl}-sn-glycero-3-phosphoethanolamine; NBD-PC (16:0-12:0), 1-oleoyl-2-{12-[(7-nitro-2-1,3-benzoxadiazol-4-yl)amino]dodecanoyl}-sn-glycero-3-phosphocholine; NBD-PA (16:0-12:0), 1-palmitoyl-2-{12-[(7-nitro-2-1,3-benzoxadiazol-4-yl)amino]dodecanoyl}-sn-glycero-3-phosphate (ammonium salt); 25-NBD-cholesterol, 25-{*N*-[(7-nitro-2-1,3-benzoxadiazol-4-yl)methyl]amino}-27-norcholesterol; and C6-NBD-ceramide, *N*-{6-[(7-nitro-2-1,3-benzoxadiazol-4-yl)amino]hexanoyl}-D-erythro-sphingosine. Donor liposomes (final concentration of 6.25 μM, containing 2.5 μM PC, 2.5 μM PE, 0.125 μM rhodamine-PE, and 0.5 μM NBD-labeled lipid) and acceptor liposomes (final concentration of 25 μM, containing 2.5 μM PC and 2.5 μM PE) were mixed, GST-tagged proteins were added to the indicated final concentrations, and the mixture was incubated at 25 °C. All reactions were performed in the presence of acceptor liposomes unless indicated otherwise. The amount of lipid extracted (nM lipid) = (fluorescence value for GST-PDZD8 (ΔTM) – fluorescence value for GST) × 500 nM/fluorescence value after the addition of Triton X-100. The initial velocity of lipid extraction (nM lipid/s) = amount of lipid extracted (nM lipid) during the initial 10 s/10 s. For anchoring of $His_6$-tagged proteins to liposomes, 1.25 μM DGS-NTA(Ni) (Avanti) was included in the donor liposomes. The fluorescence of NBD was measured with a spectrofluorometer (FP8500; JASCO, Tokyo, Japan) for 300 s at 5-s intervals, with excitation at 460 nm and emission at 530 nm. For measurement of the fluorescence of extracted lipids in the aqueous phase, the lipid extraction reaction mixture at 300 s was subjected to ultracentrifugation at $100,000 \times g$ for 15 min at 25 °C to remove the liposomes.

**Lipid-binding analysis.** PIP Strips and Membrane Lipid Strips were obtained from Echelon Biosciences (Santa Clara, CA). Custom strips were prepared by spotting lipids on a Hybond C extra membrane (Amersham). The strips were incubated for 1 h at room temperature with GST-tagged proteins (final concentration of 10 nM) in Tris-buffered saline containing 3% fatty acid-free bovine serum albumin (Sigma-Aldrich). Protein binding to the strips was detected with antibodies to GST and immunoblot reagents.

**Liposome-binding analysis.** Recombinant GST or GST-PDZD8(C1) (final concentration of 4 μM) was incubated for 30 min at 4 °C with liposomes (final concentration of 10 mM) consisting of PC and cholesterol, with or without PS, in HEPES buffer, after which the liposomes were isolated by ultracentrifugation at $200,000 \times g$ for 60 min at 4 °C, and the resulting pellet and supernatant were subjected to SDS-PAGE followed by staining with Coomassie brilliant blue.

**In vivo analysis of lipid transfer.** Mouse cortical neurons that had been transfected with plasmids encoding EGFP-Lact-C2 and mCherry-CD63, as well as with siRNAs with the use of a Nucleofector system (Lonza) were observed with an LSM800 confocal microscope (Zeiss), and the images were processed for calculation of colocalization with imaging software (ZEN, Zeiss).

**Statistical analysis.** Quantitative data are presented as means ± SE (or ±SD only for real-time PCR) and were analyzed by Student's *t*-test or the Kruskal–Wallis test followed by the Steel-Dwass multiple comparison test. A *P*-value of <0.05 was considered statistically significant.

**Reporting summary.** Further information on research design is available in the Nature Research Reporting Summary linked to this article.

## Data availability

The MS data have been deposited in the J-POST Repository Database as follows: PRT complex in mouse brain, WT (JPST000714). With Accession ID: PXD016817. PRT complex in mouse brain, KO (JPST000716). With Accession ID: PXD016818. The source data are provided as a file "Source Data." All other data are available in the Article, Supplementary Information or available from the authors upon reasonable request. Source data are provided with this paper.

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

## Acknowledgements

We thank S. Nagata and H. Sumimoto for providing the pEF-BOS-2×HA vector; T. Kitamura for providing the pMX-puro vector; M. Matsumoto for proteomics analysis; H. Takase for TEM analysis; K. Sawamoto for 3D construction of FIB-SEM images; T. Yasuda and A. Matsumoto for generation of CRISPR mutant mice; T. Ohnishi, Y. Kita, K. Matsuo, A. Kawajiri, R. Takemoto, I. Yamahata, H. Takahashi, Y. Ota, and H. Tashiro for technical assistance; T. Endo for discussion; and the Research Equipment Sharing Center at Nagoya City University for assistance. This work was supported in part by KAKENHI grants from Japan Society for the Promotion of Science (JSPS) and the Ministry of Education, Culture, Sports, Science, and Technology of Japan to M.S. (20H03255 and 20H04907) and to K.I.N. (18H05215).

## Author contributions

M.S. designed experiments, supervised the study, and wrote the manuscript. M.W., K.M., N.H., R.K., Y.M., and H.N. performed experiments. F.M. designed and performed experiments and analyzed data. K.O. performed FIB-SEM analysis. Y.T. provided split-GFP vectors and provided technical advice on lipid transfer analysis. K.I.N. supervised the study and wrote the manuscript.

## Competing interests

The authors declare no competing interests.
