## [Peer Review File · Nature Communications]

Reviewers' comments:

Reviewer #1 (Remarks to the Author):

This paper identified an interaction between protrudin and PDZD8. Protrudin and PDZD8 are known ER-proteins involved in ER-late endosome (LE) and ER-mitochondria membrane contact sites (MCS), respectively. The authors found that protrudin and PDZD8 stabilize each other. The domains required for this interaction were also identified. By using a split GFP assay, they found that overexpression of protrudin or PDZD8 induces MCS between the ER and LE, but also between the ER and mitochondria. By using liposome-FRET assay and analyzing several deletion constructs, they found that PDZD8 has lipid extraction activity; the TM, SMP and PDZ domains of PDZD8 contribute to lipid extraction and the C1 domain specifically binds to PS but is not required for lipid extraction activity. Finally, the authors show that co-expression of protrudin and PDZD8 promotes MVB formation.

The characterization of a new interaction between protrudin and PDZD8, lipid transfer activity of PDZD8 and role of the protrudin-PDZD8 complex on endosome membrane expansion is very valuable for cell biologist. The data presented are generally of good quality. However, the findings are not well interconnected as expected by the title of the paper "The protrudin-PDZD8 complex promotes endosome maturation at MCS with the ER by mediating lipid transfer" The authors did not connect their results to get more mechanistic insights. Separately: protrudin and PDZD8 overexpression induces MCS formation, but the role of the complex in this function was not added; by using liposome-FRET assay, lipid transfer activity of PDZD8 and domains required were identified but it was not analyzed how this lipid transfer activity could contribute to the expansion/maturation of endosomes. There are a few important points to be clarified before publication.

Major points:

- 1) From the experiments using the split GFP assay and overexpression of protrudin or PDZD8, the authors concluded that protrudin-PDZD8 complex plays a role in MCS formation between the ER and LE. Knockdown of these proteins need to be done to analyze whether MCS formation is impaired. It is important to demonstrate that the protrudin-PDZD8 complex play a role in MCS formation by overexpressing deletion constructs, which were shown to affect the interaction between these proteins. CYP1A1-PDZD8 chimera should be tested in the ability to induce MCS, since it localizes at the ER but cannot bind protrudin. Is protrudin-PDZD8 interaction required to induce MCS?
- 2) The authors suggested that C1 domain might function to tether the ER to LE membrane by interaction with PS enriched in the LE membrane. It would be good to test whether PDZD8 requires or not its C1 domain to induce MCS formation by overexpression of deltaC1 versus FL using split GFP assay.
- 3) The authors found that PDZD8 has lipid extraction activity, which they try to connect to the expansion of endosome membrane (maturation). Disruption of interaction between protrudin and PDZD8 need to be done to analyze whether the specific interaction is required to induce MVB formation. Overexpression of WT protrudin plus different deletions constructs for PDZD8 need to be tested. Is the TM domain and region between PDZ and C1 of PDZD8 required for MVB formation? Is the C1 domain that bind PS required for endosome expansion?
- 4) Endosome maturation is important for lysosomal degradation. Is the activity of lysosome affected by overexpression of protrudin-PDZD8 complex? Magic red or Sir-Lyso trackers could be used to analyze cathepsin activity and distribution.

- 5) Expansion of endosomal membrane could be also caused by disruption of endosome fission. ER-endosome contact has been already shown to be important for endosome fission, which has an impact on lysosome function (Allison et al., JCB, 2017). Is the overexpression of protrudin plus PDZD8 causing fission defects? This need to be added to the discussion.
- 6) The authors suggest that protrudin and PDZD8 stabilize each other to form a stable complex. However, there is just one experiment from brain extracts from WT mice (+/+) versus protrudin deficient (-/-) or pdzd8-deficient mice (-/-) and just one experiment from PC12 cells. At least 2 independent experiments and quantification need to be added to Figures 1d, e and f.
- 7) The authors show that protrudin and PDZD8 overexpression also induce MCS between the ER and mitochondria (Fig 2f-h). Then, a FIB-SEM reconstruction image of a MVB is shown in a cell co-expressing protrudin and PDZD8, the MVB is associated to ER and mitochondria (Fig 6g). There is not discussion about the role of protrudin-PDZD8 in endosome versus mitochondria MCS formation. Is this protrudin-PDZD8 complex required for the contact between these three organelles?

Minor points.

- 1) In Fig 5a, red signal is weak. An image with better FLAG staining (red) intensity need to be added.
- 2) In Fig 5d, there is a difference in PDZD8 distribution in cells co-expressing protruding and PDZD8 in the presence of different endosome and mitochondria markers. For example, PDZD8 distribution is different for Rab7 and Tom20 compared to the ones with calnexin and Rab9. Is the overexpression of markers affecting PDZD8 distribution and vacuole formation?

Ginny G. Farias

Reviewer #2 (Remarks to the Author):

ERMES is an ER-mitochondria tethering complex in yeast. Proteins in the complex have SMP domains, which bind and transfer lipids between membranes. Recently, the Polleux group found the mammalian protein PDZD8 is similar to the ERMES protein Mmm1 and tethers the ER to mitochondria. This study shows that PDZD8 binds protrudin at ER-late endosome (LE) contact sites. It argues that PDZD8 transfers lipids from the ER to LEs and plays a role in LE biogenesis and MVB production. If true, this is an important discovery since it would identify a novel ER-LE tether that facilitates LE membrane expansion and MVB production. However, considerable additional work is necessary to make the central claims of the study convincing.

1. There is no evidence that endogenous PDZD8 is at contact sites. Also, split-GFP is irreversible and therefore it will drive the formation of contact sites. It is important to show that PDZD8 is at contact sites in unperturbed cells. The effect of PDZD8 knock down and over expression contacts sites should be quantitatively determined by a method that does not rely on split-GFP
2. It is not possible to gauge the rate of lipid transport in vitro. How many lipid monomers are transferred per minute per protein? Is the transport rate high enough to be physiologically relevant?
3. The study would be significantly stronger if it showed that PDZD8 depletion slows lipid transport

from the ER to the LE in vivo.

4. The results in Figure 4 should be verified with liposome binding assays.

5. The images in Figure 5 and 6 should be quantified.

Reviewer #3 (Remarks to the Author):

This manuscript from Shirane et al, identifies PDZD8 as a protrudin-associated protein and suggests that the ER resident VAP-protrudin-PDZD8 complex promotes ER-endosome contact sites and lipid transfer from ER to endosomes, thereby facilitating endosome maturation.

In proteomics and mass spectrometry analyses of HEK293T cells, Neuro2A cells and brain extracts of mice the authors identify PDZD8, previously shown to function in ER-mitochondria contact sites, as being in a complex with the ER-protein protrudin. They proceed with co-IP experiments to map the regions of protrudin and PDZD8 required for their association. By using KO mice of either Protrudin or PDZD8, in addition to siRNA mediated depletion in PC12 cells, they show that Protrudin and PDZD8 mutually stabilize each other. They further show that PDZD8 promotes ER-endosome contact site formation, similar to protrudin, by using a split-GFP approach. Since PDZD8 contains protein domains that have been implicated in lipid transfer activity in other proteins, they use a FRET-based liposome assay and lipid overlay analyses to address whether recombinantly purified PDZD8 harbors lipid binding and/or lipid transfer activity, and conclude that PDZD8 binds selectively to PS, and has a lipid extraction activity, but is not able to mediate insertion of lipids in an acceptor membrane.

Finally, the authors show by immunofluorescence and electron microscopy that transient co-expression of PDZD8 and protrudin in HeLa cells causes the formation of giant vacuolar compartments of late endocytic origin, containing intraluminal vesicles, and enwrapped in ER. Given the importance of membrane contact sites and the implication of the involved proteins in different neurological diseases, the results presented in this manuscript are of potentially great interest to a broad readership, pending major revision. The manuscript reads well and the data are presented in a clear and logical way. The studies regarding the association of PDZD8 with protrudin are of high quality and provide important information about this protein complex. The lipid extraction activity of PDZD8 is convincingly shown. The functional assays regarding the formation of ER-endosome contact sites and a possible role of the PDZD8-protrudin complex in endosome maturation are, however, very preliminary and overinterpreted, and require substantial additional experiments to be conclusive.

Major points:

1. A major concern with this study is the proposed role for protrudin and PDZD8 in endosome maturation shown in Fig. 5 and 6. I am concerned that the enlarged, convoluted vacuoles seen by Protrudin and PDZD8 co-expression are simply an overexpression artefact due to massive ER-endosome contacts, leading to hyper-recruitment of ER membranes which surround the endosome (Fig 5d, Fig 6 d). This will prevent natural endosome maturation, which is maintained through a balance between fusion and fission events, and thus does not represent a physiological role of the PDZD8-protrudin complex in endosome maturation. The morphology of the enlarged MVBs looks

aberrant. ILVs typically have a round shape with a diameter around 50 nm (Adell and Teis 2011, Teis 2010, Wenzel 2018). The “ILVs” in the enlarged MVBs caused by protrudin and PDZD8 co-expression have irregular shapes and it is difficult to see from the images whether these putative ILVs are abscised or still connected to the limiting membrane of the endosome as membrane invaginations (Fig. 6c).

To address a functional role of the protrudin-PDZD8 synergy in ER-endosome contact sites and a possible role in endosome maturation, the authors need to co-express these proteins close to their endogenous levels, and use deletion mutants of PDZD8, deficient in lipid binding or protrudin binding, (preferably siRNA resistant, in cells depleted for endogenous PDZD8 by siRNA) to address any changes that might occur regarding LE abundance and morphology, the number of ILVs, cargo degradation such as EGFR degradation in lysosomes and retromer dependent recycling of receptors such as the CI-M6PR, as readouts for endosome maturation and functionality.

Protrudin has an established role in the regulation of directional endosome transport (Shirane and Nakayama 2006, Raiborg 2015). The authors have not considered whether PDZD8 can influence the role of protrudin in regulation of LE positioning. Whereas the generation of large vacuoles seen by co-expression of high levels of protrudin and PDZD8 in this study hampers the possibility to study endosome positioning, this could easily be tested under more controlled expression levels as suggested above.

The statement that the numbers of ILVs are increased should be backed up by quantifications. Tomography should be performed to assess whether the proposed ILVs (Fig. 6c) are abscised or if they represent membrane invaginations of convoluted vacuoles.

2. Whereas the split-GFP assay used in Fig.2 is an elegant approach to assay MCSs, these experiments are missing important controls, and should be exploited to study the potential synergy between PDZD8 and protrudin in contact site formation. The author’s conclusion that “the protrudin-PDZD8 complex plays a role in formation of MSCs (sic) between the ER and LEs”, is not supported by the present data. Whereas the authors nicely show that the stable expression of either protrudin or PDZD8 in HeLa cells promote ER-endosome contact site formation, it will be important to know whether PDZD8 is able to influence the ability of protrudin to form contact sites, or vice versa. This can be achieved by utilizing the split-GFP assay in stable cell lines co-expressing protrudin and PDZD8 WT and deletion mutants as suggested for assaying endosomal function above.

The stable cell lines used in Fig. 2 should be characterized by Western Blotting to show the level of expression compared to endogenous PDZD8 and protrudin.

The GFP-signal from the split GFP-probes used to detect ER-endosome contact sites and ER-mitochondria contact sites should be tested for colocalization with endosomal and mitochondrial markers respectively.

The results indicate that protrudin can mediate contact between ER and mitochondria in addition to endosomes. This potentially additional role of protrudin should be commented on and discussed. Do the authors think that this reflects a true function of protrudin? Alternatively it could reflect mitophagocytic events, where damaged mitochondria, induced by the GFP-probe, are in close apposition to the ER. This could be easily tested by using autophagosomal markers like LC3.

The area of cells (%) covered by GFP-dots is around 10% for the endosome probe (Fig2. d,e) and 0.15% for the mitochondria probe (Fig. 2h). These numbers do not fit their relative immune fluorescence images (Fig. 2 c,g). Why is the area measured for mitochondria contacts so much lower, although it looks more from the IF?

3. Fig. 4c suggests a model where the PDZD8-C1 domain promotes ER-endosome contact by binding to endosomal PS. Whereas this is a nice hypothesis, this model has to be supported by experimental data. The authors should use their split-GFP assay and compare PDZD8 WT and deltaC1 to test this. In the model PDZD8 forms contact with endosomes through binding to PS. If PDZD8 extracts or transfers PS between membranes, how could the binding to PS be utilized to form the contact? The more logical explanation would perhaps be that since PDZD8 is in a complex with protrudin, protrudin is providing the contact, whereas PDZD8 mediates PS extraction. This could be tested by investigating whether the PDZD8-mediated contact site formation observed in the split-GFP assay is sensitive to the VPS34 inhibitor SAR405, which should abolish protrudin-mediated contact site formation.

The model in Fig. 4c also puts forward the possible need of Ca²⁺ for the PDZD8 mediated ER-endosome contact site formation. Given that PDZD8 has an established role in Ca²⁺ homeostasis and transfer of Ca²⁺ from ER to mitochondria, and that other ER-endosome contact sites depend on Ca²⁺ (Eden 2016), this is a relevant hypothesis which should be tested using a Ca²⁺ chelator in the same assay.

Minor points:

1. In Fig. 2 the number of GFP dots and the area of dots per cell should be quantified from three independent experiments, and data shown as mean between independent experiments.

2. In Fig. 3, the authors conclude that PDZD8 has lipid extraction activity. This is especially tested in the experiments in Fig. 3 h-k. In the title of this manuscript as well as in several statements throughout the text and in the model in fig. 7, however, the authors claim that PDZD8 and its role in membrane contact sites and endosome maturation is explained by its lipid transfer activity and that PDZD8 transfer lipids from the ER to the endosome. The authors should modify their statements according to their data. If PDZD8 extracts PS and other lipids, where do these lipids end up?

3. In Fig. 4, it would also be informative to compare the lipid binding specificities of PDZD8 WT, SMP and PDZ with C1 in the lipid overlay assay. The authors state that the PDZD8-C1 binds specifically to PS. There is, however, binding to several phosphoinositides, especially PI4P. This should be discussed. How many times was the lipid overlay assay performed? Since lipid overlay assays can be rather unspecific, the binding of PDZD8 to PS should be backed up in a liposome-assay.

4. Whereas it is evident from their data in Figure. 5 and 6, that co-expression of protrudin and PDZD8 promote the formation of enlarged late endosomal structures, there is no data to support that this is caused by the proposed lipid transfer activity of PDZD8 as claimed in the title. This should be tested by co-expressing protrudin with deletion mutants of PDZD8 which reduce lipid extraction activity, such as the deltaSMP and the deltaPDZ. Also the number of early and late endosomes should be quantified following these more targeted perturbations to assess the functions of protrudin and PDZD8 in endosome maturation.

5. In the introduction, the authors write the following statement: "Such endosome maturation is associated with expansion of the LE membrane, with the lipids required for this expansion being

thought to be supplied by the ER at MCSs". Clearly ER-endosome contact sites are important in the regulation of endosome maturation and function, and transfer of cholesterol between the two membranes. Loss of ER-endosome contact sites lead to impaired ESCRT-mediated ILV formation and enlarged MVEs (Eden 2016, Kobuna 2010). Endosomal membrane dynamics are, however, mainly maintained by a combination of endosome fusion and fission of recycling tubules. The authors should provide specific references for their statement regarding the requirement for ER-endosome contact sites in LE membrane expansion.

6. To the best of my knowledge, the phenotypes of the protrudin- or PDZD8 KO mice have never been described. It would be informative to provide some information about this in the manuscript or to referees, if the authors want to save this information for a separate publication.

7. The model in fig. 7 states that ILVs "increase". Unless the data on ILVs is severely strengthened (see major point 1), no statement on ILV morphology, size or number can be given in the model.

8. In all figures showing insets/magnifications, the corresponding area should be highlighted by a box in the original image.

Response to Reviewer #1

We thank the reviewer for the careful evaluation of our manuscript and for the statements that “The characterization of a new interaction between protrudin and PDZD8, lipid transfer activity of PDZD8 and role of the protrudin-PDZD8 complex on endosome membrane expansion is very valuable for cell biologist,” and that “The data presented are generally of good quality.” We also thank the reviewer for suggestions that we feel have helped us to improve our manuscript.

In the revised manuscript, we have shown that the protrudin-PDZD8 system mediates membrane tethering, MCS formation, and lipid transfer between the ER and endolysosomes as well as demonstrated the physiological relevance of this system in neurons and the pathological relevance of its dysfunction, with the physiological and pathological aspects in particular being newly added. On the basis of our additional findings, we have largely modified the original manuscript, including a change to the title.

Our specific responses to the points raised are as follows:
(This response has low resolution thumbnails for clarity, but please refer to the illustration in the text for high resolution data.)

Major points

1-1. From the experiments using the split GFP assay and overexpression of protrudin or PDZD8, the authors concluded that protrudin-PDZD8 complex plays a role in MCS formation between the ER and LE. Knockdown of these proteins need to be done to analyze whether MCS formation is impaired.

[Response] As suggested by the reviewer, we depleted HeLa cells of protrudin or PDZD8 with the use of specific siRNAs and then performed the split-GFP assay. The efficiency of siRNA-mediated knockdown of protrudin and PDZD8 is shown in **new Supplementary Figure 9a and 9b**, respectively. Depletion of protrudin or PDZD8 significantly attenuated MCS formation between the ER and endolysosomes (late endosomes and lysosomes, or LyLEs) (**new Fig. 2d**). Such depletion had a smaller effect on MCS formation between the ER and mitochondria (**new Fig. 2h**), with a modest reduction being observed in cells depleted of protrudin (not significant, likely as a result of the relatively low efficiency of depletion) or of PDZD8 ($P < 0.05$). These results suggest that endogenous protrudin and PDZD8 indeed contribute to MCS formation, with the formation of MCSs between the ER and LyLEs being more dependent on the protrudin-PDZD8 system than that of those between the ER and mitochondria. We have now clarified these points in the revised manuscript (pages 7-8, lines 183-185, 197-201).

(Supplementary Fig. 9a and 9b) Efficiency of siRNA-mediated depletion of protrudin (a) and PDZD8 (b) in HeLa cells.

(Fig. 2d and 2h) MCS formation between the ER and either LyLEs (d) or mitochondria (h) in HeLa cells depleted of protrudin or PDZD8.

1-2. *It is important to demonstrate that the protrudin-PDZD8 complex play a role in MCS formation by overexpressing deletion constructs, which were shown to affect the interaction between these proteins. CYP1A1-PDZD8 chimera should be tested in the ability to induce MCS, since it localizes at the ER but cannot bind protrudin. Is protrudin-PDZD8 interaction required to induce MCS?*

[Response] As suggested, we forcibly expressed PDZD8(WT) or the CYP-PDZD8 chimera in HeLa cells and examined the formation of MCSs with the split-GFP assay. In addition, we examined the effect of the PDZD8(Δ C1) mutant, given that the C1 domain functions to tether the ER and LyLE membranes by interaction with PS and PtdIns(4)P in the LyLE membrane. As expected, the PDZD8(Δ C1) mutant had lost the ability to promote formation of MCSs between the ER and LyLEs (**new Fig. 2e**). Unexpectedly, however, the CYP-PDZD8 chimera fully retained this ability. These results suggest that, although protrudin and PDZD8 are expressed in proximity to each other and cooperate to promote the formation of

MCSs between the ER and LyLEs, direct molecular interaction between protrudin and PDZD8 might not be necessary for this effect. We have addressed these points in the revised manuscript (page 7, lines 185-194).

(Fig. 2e) MCS formation between the ER and LyLEs in HeLa cells expressing WT or mutant forms of PDZD8.

2. The authors suggested that C1 domain might function to tether the ER to LE membrane by interaction with PS enriched in the LE membrane. It would be good to test whether PDZD8 requires or not its C1 domain to induce MCS formation by overexpression of deltaC1 versus FL using split GFP assay.

[Response] As mentioned above, we performed the experiment suggested by the reviewer, and we found that the C1 domain of PDZD8 is indispensable for the promotion of MCS formation (**new Fig. 2e, shown above**). We therefore conclude that the interaction between the C1 domain of PDZD8 and lipids such as PS and PtdIns(4)P is essential for the ability of PDZD8 to tether the ER and LyLE membranes. We have clarified this point in the revised manuscript (page 7, lines 185-194).

3. The authors found that PDZD8 has lipid extraction activity, which they try to connect to the expansion of endosome membrane (maturation). Disruption of interaction between protrudin and PDZD8 need to be done to analyze whether the specific interaction is required to induce MVB formation. Overexpression of WT protrudin plus different deletions constructs for PDZD8 need to be tested. Is the TM domain and region between PDZ and C1 of PDZD8 required for MVB formation? Is the C1 domain that bind PS required for endosome expansion?

[Response] To address these questions, we expressed a series of PDZD8 mutants together with HA-protrudin in HeLa cells and examined the formation of MVB-like structures, which we now refer to as abnormal large vacuoles (ALVs) in the revised manuscript, given that such structures differ substantially from normal MVBs. The PDZD8(Δ SMP) mutant, which manifests reduced lipid extraction activity, had largely lost the ability to induce the

formation of ALVs (**new Fig. 5e and 5f**), suggesting that lipid extraction mediated by PDZD8 is definitely required for this ability. The PDZD8(Δ C1) mutant was partially defective in the ability to induce ALV formation, suggestive of the importance of the interaction between the C1 domain of PDZD8 and lipids such as PS and PtdIns(4)P. In contrast, the CYP-PDZD8 chimeric mutant induced ALV formation to an extent similar to that observed with PDZD8(WT), consistent with the results of the split-GFP assay (**new Fig. 2e**). We have now addressed these points in the revised manuscript (page 10, lines 285-293).

(Fig. 5e and 5f) Formation of ALVs in HeLa cells expressing WT or mutant forms of PDZD8.

4. Endosome maturation is important for lysosomal degradation. Is the activity of lysosome affected by overexpression of protrudin-PDZD8 complex? Magic red or Sir-Lyso trackers could be used to analyze cathepsin activity and distribution.

[Response] As suggested by the reviewer, HeLa cells expressing exogenous protrudin and PDZD8 were examined for Magic red fluorescence (**new Supplementary Fig. 15a**), with quantitative data also being provided (**new Supplementary Fig. 15b**). Lysosomal activity was indeed increased in association with LyLE enlargement in cells overexpressing protrudin and PDZD8 compared with control cells. We have now clarified this point in the revised manuscript (page 11, lines 325-328).

In response to the reviewer's comment regarding the distribution of lysosomes, we examined the distribution of the LyLE marker LAMP1 in HeLa cells expressing HA-protrudin and PDZD8(WT)-FLAG by immunofluorescence analysis with antibodies to LAMP1 and to FLAG. The LAMP1 signal showed marked aggregation around the large MVB-like structures that formed in the transfected cells (**new Supplementary Fig. 13e**). In the case of cells expressing HA-protrudin and PDZD8(Δ SMP)-FLAG, no MVB-like structures were observed, and the LAMP1 signal was dispersed around the nucleus. We have now described these results in the revised manuscript (pages 11-12, lines 329-339).

(Supplementary Fig. 15) The intensity of Magic red representing lysosomal activity in HeLa cells expressing protrudin and PDZD8.

(Supplementary Fig. 13e) Distribution of LyLEs in HeLa cells coexpressing protrudin and either WT or Δ SMP mutant forms of PDZD8.

5. Expansion of endosomal membrane could be also caused by disruption of endosome fission. ER-endosome contact has been already shown to be important for endosome fission, which has an impact on lysosome function (Allison *et al.*, JCB, 2017). Is the overexpression of protrudin plus PDZD8 causing fission defects? This need to be added to the discussion.

[Response] The paper mentioned by the reviewer (Allison *et al.*, *J. Cell Biol.*, 2017) showed that defects in ER-LyLE contacts induced by mutations of spastin or REEP1 that cause hereditary spastic paraplegia (HSP) result in failure of lysosomal fission in mouse primary neurons. It is of particular interest that mutations of the human protrudin gene (*ZFYVE27*) also cause HSP, suggesting that the protrudin-PDZD8 system might also contribute to endosomal maturation at the level of organelle fusion or fission. We therefore examined the physiological function of the protrudin-PDZD8 system in mouse primary neurons with the approach adopted by Allison *et al.*

Surprisingly, depletion of PDZD8 (which also reduces the abundance of protrudin, as shown in **Figure 1g and 1i**) in mouse primary neurons induced frequent gross enlargement of LAMP1⁺ LyLEs (**new Fig. 7a and 7b**). These vesicles showed abnormal ultrastructures characterized by many endolysosomal membrane coils and almost no sign of normal intraluminal vesicles (**new Fig. 7c**), and they were highly reminiscent of those seen in neurons of spastin or REEP 1 mutant mice. A similar phenotype was also observed in mouse primary neurons expressing the PDZD8(Δ SMP) mutant, likely as a result of a dominant negative effect of this mutant on the normal fission process of LyLEs in neurons. These results suggest that the lipid extraction activity of PDZD8 contributes to the normal endosomal maturation process in these cells. Allison *et al.* showed that endosomal maturation is regulated at the level of the balance between fusion and fission, the latter of which takes place at ER-LyLE MCSs, whose formation is regulated by spastin and REEP1. Given the similar molecular functions shared by spastin, REEP 1, and protrudin as well as the roles of these proteins in the pathogenesis of HSP, we conclude that MCS formation dependent on the protrudin-PDZD8 system is essential for endosomal maturation mediated by lipid transfer activity in the physiological setting. We have now addressed these points in the revised manuscript (page 12, lines 341-360).

The abnormal LyLEs found in PDZD8-depleted mouse primary neurons are also highly similar to those seen in HeLa cells overexpressing both protrudin and PDZD8. As pointed out by another reviewer, this phenotype of HeLa cells is likely attributable to an artifact of overexpression that results in endosomal fission defects. An abnormal increase in MCS formation between the ER and LyLEs might thus result in an excess supply of lipid that impairs endosome fission and maturation. We have now added this point to the Discussion section of the revised manuscript (pages 13-14, lines 396-406).

In the revised manuscript, we have thus provided physiological evidence that the protrudin-PDZD8 system indeed contributes to endosomal maturation by mediating lipid transfer in mouse primary neurons.

(Fig. 7a and 7b) Gross enlargement of LyLEs induced by either depletion of PDZD8 or expression of PDZD8(Δ SMP) in mouse primary neurons.

(Fig. 7c) Ultrastructure of abnormal LyLEs induced by depletion of PDZD8 or expression of PDZD8(Δ SMP) in mouse primary neurons.

6. The authors suggest that protrudin and PDZD8 stabilize each other to form a stable complex. However, there is just one experiment from brain extracts from WT mice (+/+) versus protrudin deficient (-/-) or pdzd8-deficient mice (-/-) and just one experiment from PC12 cells. At least 2 independent experiments and quantification need to be added to Figures 1d, e and f.

[Response] We have now provided another independent result for the WT and mutant mice in the revised manuscript (new Supplementary Fig. 4).

(Supplementary Fig. 4a and 4b) Results of another independent analysis of protrudin and PDZD8 expression in the brain of PDZD8- or protrudin-deficient mice, related to Figure 1g and 1h.

7. The authors show that protrudin and PDZD8 overexpression also induce MCS between the ER and mitochondria (Fig 2f-h). Then, a FIB-SEM reconstruction image of a MVB is shown in a cell co-expressing protrudin and PDZD8, the MVB is associated to ER and mitochondria (Fig 6g). There is not discussion about the role of protrudin-PDZD8 in endosome versus mitochondria MCS formation. Is this protrudin-PDZD8 complex required for the contact between these three organelles?

[Response] Given that both protrudin and PDZD8 are localized to the ER membrane, we believe that it is unlikely that these proteins affect MCS formation between endosomes and mitochondria. However, given that Rab7 was shown to contribute to anchoring between endosomes and mitochondria (Wong *et al.*, *Nature*, 2018) and that protrudin binds to Rab7, it is possible that the protrudin-PDZD8 system is indirectly involved in such anchoring. We have now mentioned this point in the Discussion section of the revised manuscript (page 13, lines 390-395).

Minor points

1. In Fig 5a, red signal is weak. An image with better FLAG staining (red) intensity need to be added.

[Response] As requested, we replaced the original FLAG image with an improved one (**new Fig. 5a**) in the revised manuscript.

(Fig. 5a) HeLa cells transfected with expression vectors for FLAG-protrudin, PDZD8-Myc, or both proteins were stained with antibodies to FLAG (red) or to Myc (green) as indicated.

2. In Fig 5d, there is a difference in PDZD8 distribution in cells co-expressing protrudin and PDZD8 in the presence of different endosome and mitochondria markers. For example, PDZD8 distribution is different for Rab7 and Tom20 compared to the ones with calnexin and Rab9. Is the overexpression of markers affecting PDZD8 distribution and vacuole formation?

[Response] We did not overexpress the markers for endosomes and mitochondria, but rather used antibodies to detect endogenous marker proteins. To avoid misunderstanding, we replaced the original images with improved ones (new Fig. 5e) in the revised manuscript.

(Fig. 5d) HeLa cells transfected with expression vectors for FLAG-protrudin and PDZD8-Myc were stained with antibodies to Myc (green) as well as with those (red) to the organelle marker proteins calreticulin (ER), Rab7 or Rab9 (LEs), LAMP1 (LyLEs), or Tom20 (mitochondria).

Response to Reviewer #2

We thank the reviewer for the careful evaluation of our manuscript and for the statement that “If true, this is an important discovery since it would identify a novel ER-LE tether that facilitates LE membrane expansion and MVB production.” We also thank the reviewer for suggestions that we feel have helped us to improve our manuscript.

In the revised manuscript, we have shown that the protrudin-PDZD8 system mediates membrane tethering, MCS formation, and lipid transfer between the ER and endolysosomes as well as demonstrated the physiological relevance of this system in neurons and the pathological relevance of its dysfunction, with the physiological and pathological aspects in particular being newly added. On the basis of our additional findings, we have largely modified the original manuscript, including a change to the title.

Our specific responses to the points raised are as follows:

(This response has low resolution thumbnails for clarity, but please refer to the illustration in the text for high resolution data.)

1. There is no evidence that endogenous PDZD8 is at contact sites. Also, split-GFP is irreversible and therefore it will drive the formation of contact sites. It is important to show that PDZD8 is at contact sites in unperturbed cells. The effect of PDZD8 knock down and over expression contacts sites should be quantitatively determined by a method that does not rely on split-GFP.

[Response] Localization of endogenous PDZD8 at ER-mitochondria contact sites was previously shown by super-resolution microscopy (3D SIM) as well as by FIB-SEM (Hirabayashi *et al.*, *Science* 358: 623–630, 2017).

As suggested by the reviewer, we have now shown that depletion of protrudin or PDZD8 in HeLa cells indeed attenuated the split-GFP signal for contacts between the ER and endolysosomes (late endosomes and lysosomes, or LyLEs) (**new Fig. 2d**). Conversely, MCS formation was promoted by overexpression of protrudin and PDZD8 (**new Fig. 2c**). These lines of evidence indicate that assembly of the split-GFP fragments depends on preexisting interorganelle contact sites whose formation is facilitated by the protrudin-PDZD8 system, with the signals not being simply generated as a result of irreversible interaction of the split-GFP fragments. Although we understand the reviewer’s concern about potential artifacts of the split-GFP assay, a recent study showed that split-GFP signals at overlapping regions of the ER and mitochondria in yeast largely colocalized with ERMES, an authentic ER-mitochondria tethering structure that contains a yeast ortholog of PDZD8 (Kakimoto *et al.*, *Sci. Rep.* 8: 6175, 2018). However, we now refer to the limitations of this assay by stating that “it is difficult to formally exclude the possibility that these organelle contact sites are artificially formed as a result of the irreversible association of the split-GFP probes” in the revised manuscript (page 7, lines 173-177).

It is of note that imaging of MCSs between the ER and mitochondria by super-resolution microscopy or electron microscopy (Hirabayashi *et al.*, *Science* 358: 623–630, 2017) is relatively easy, but identification of ER-LyLE contact sites with such technologies is difficult, mainly because of the small size of endosomes and their similar morphology to the ER (**Fig. R1 for reviewing purposes only**). Furthermore, such imaging is less reliable with regard to quantitative analysis and less suitable for objective evaluation of multiple samples

than is the split-GFP method. Given that we needed to examine the effects of WT and various mutant forms of protrudin and PDZD8 in the present study, we adopted the split-GFP assay for quantitative analysis. We hope that the reviewer agrees with the technical limitations of other methods and our rationale for selection of the split-GFP assay.

(Fig. 2c–2e) MCS formation between the ER and LyLEs in HeLa cells overexpressing protrudin, PDZD8, or both proteins (c), in HeLa cells depleted of protrudin or PDZD8 (d), or in HeLa cells expressing WT or mutant forms of PDZD8 (e).

(Fig. R1) Representative electron microscopic images illustrating the easy identification of MCSs between the ER and mitochondria (red arrows) but not of those between the ER and LyLEs.

2. It is not possible to gauge the rate of lipid transport in vitro. How many lipid monomers are transferred per minute per protein? Is the transport rate high enough to be physiologically relevant?

[Response] In response to the reviewer's comment, we calculated how many lipid monomers are transferred per minute per protein. In brief, we measured the fluorescence signals for NBD-PS in the presence of 10 nM GST (**new Supplementary Fig. 10a**) or GST-PDZD8(Δ TM) (**new Supplementary Fig. 10b**). The extracted NBD-PS was determined

from the difference in the fluorescence signals between the two reactions (**new Supplementary Fig. 10c**), which had essentially reached a plateau at 300 s. The total signal was determined by the addition of Triton X-100 after the plateau had been reached.

(Supplementary Fig. 10a–10c) The liposome–FRET (fluorescence resonance energy transfer) assay for GST (a) and GST-PDZD8(Δ TM) (b). The difference between the two curves is also shown (c).

Lipid extraction amount and velocity were calculated as follows:

Lipid extraction amount (nM lipid/nM protein) = $\{[(\text{fluorescence value of GST-PDZD8}(\Delta\text{TM}) \text{ at } 300 \text{ s}) - (\text{fluorescence value of GST at } 300 \text{ s})] / [(\text{fluorescence value after addition of Triton X-100}) / 500 \text{ nM lipid}]\} / 10 \text{ nM protein}$.

Lipid extraction velocity (nM lipid/nM protein per min) = (lipid extraction amount at the time corresponding to half the maximal signal) / (time in minutes corresponding to half the maximal signal).

The values for lipid extraction amount and velocity determined for PA, PS, PE, PC, ceramide, and cholesterol are now presented (**new Fig. 3d, 3e, 3l, and 3m and Supplementary Fig. 10d**), and we have now described these results in the revised manuscript (page 8, lines 220-226 and page 9, lines 242-246).

(Fig. 3d, 3e, 3l, and 3m) Extraction amount and velocity for PA, PS, PE, PC, ceramide, and cholesterol mediated by PDZD8(Δ TM).

d

PDZD8(Δ TM) (10 nM), NBD-lipid (500 nM)

Extracted lipid	PA	PS	PE	PC	Cer	Chol
Amount (nM lipid/ nM protein)	138.9	310.9	77.9	95.6	388.7	14.3
Velocity (nM lipid/ nM protein per min)	16.7	44.4	12.6	14.0	68.6	2.9

(Supplementary Fig. 10d) Summary of extraction amount and velocity for PA, PS, PE, PC, ceramide, and cholesterol mediated by PDZD8(Δ TM).

Given that the amount and rate of extraction of phospholipids differ among tissues, cells, organelles, internal and external lipid bilayers, and environments surrounding the membrane, it is not clear whether the results obtained for PDZD8 in the present study are physiologically relevant. However, the values are similar to those previously determined for the ERMES complex, which contains a yeast ortholog of PDZD8 (Kawano *et al*, *J. Cell Biol.*, 2017), suggesting that our results are reasonable and that the experimental system is valid. In addition, the amount of lipid extracted by PDZD8(Δ TM) increased in a protein concentration-dependent manner (**new Fig. 3h and 3i**), providing further support for the validity of the experimental system. We have now mentioned these points in the revised manuscript (page 9, lines 242-246).

(Fig. 3h and 3i) Extraction of PS by PDZD8(Δ TM) in a protein concentration–dependent manner.

3. The study would be significantly stronger if it showed that PDZD8 depletion slows lipid transport from the ER to the LE in vivo.

[Response] Whereas it is technically difficult to directly measure lipid transport velocity in vivo, we now present evidence showing that lipid [PS or PtdIns(4)P] binding activity of PDZD8 is essential for MCS formation as well as normal endosome maturation by siRNA-mediated depletion or mutant expression of PDZD8 in HeLa cells or mouse primary neurons.

As mentioned above, depletion of protrudin or PDZD8 in HeLa cells attenuated MCS formation between the ER and LyLEs (**new Fig. 2d**), whereas this process was promoted by overexpression of protrudin and PDZD8 (**new Fig. 2c**). In contrast, the PDZD8(Δ C1) mutant, which lacks lipid [PS or PtdIns(4)P] binding activity, had no effect (**new Fig. 2e**), suggesting that lipid binding is essential for MCS formation.

We have also now shown that the protrudin-PDZD8 system contributes to endosomal maturation in mouse primary neurons. Depletion of PDZD8 (which also reduces the abundance of protrudin as shown in **Figure 1g and 1i**) in mouse primary neurons induced frequent gross enlargement of LAMP1⁺ LyLEs (**new Fig. 7a and 7b**). These vesicles showed abnormal ultrastructures with many endolysosomal membrane coils and almost no sign of normal intraluminal vesicles (**new Fig. 7c**). A similar phenotype was observed in mouse primary neurons expressing the PDZD8(Δ SMP) mutant, likely as a result of a dominant negative effect on the normal fission process of LyLEs. These results indicate that the lipid extraction activity of PDZD8 contributes to the normal endosomal maturation process in these cells. They are also highly reminiscent of those of a previous study (Allison *et al.*, *J. Cell Biol.*, 2017) showing that defects in ER-LyLE contacts induced by mutations of spastin or REEP 1 that cause hereditary spastic paraplegia (HSP) result in failure of lysosomal fission and the appearance of grossly enlarged endosomes in mouse primary neurons. It is of particular interest that mutations of the human protrudin gene (*ZFYVE27*) also cause HSP, suggesting that the protrudin-PDZD8 system might also contribute to the endosomal maturation process at the level of fusion-fission. Given the similar molecular functions and structures of spastin, REEP1, and protrudin as well as the roles of these proteins in the pathogenesis of HSP, we conclude that MCS formation and lipid transfer mediated by the protrudin-PDZD8 system are essential for endosome maturation in the

physiological setting. We have now addressed these points in the revised manuscript (page 12, lines 341-360).

(Fig. 7a and 7b) Gross enlargement of LyLEs induced by either depletion of PDZD8 or expression of PDZD8(Δ SMP) in mouse primary neurons.

(Fig. 7c) Ultrastructure of abnormal LyLEs induced by depletion of PDZD8 or expression of PDZD8(Δ SMP) in mouse primary neurons.

4. The results in Figure 4 should be verified with liposome binding assays.

[Response] As requested by the reviewer, we verified the interaction between GST-PDZD8(C1) and PS with a liposome binding assay (new Fig. 4d and 4e). The assay revealed that GST-PDZD8(C1) was more efficiently moved to the pellet fraction (P) from the supernatant fraction (S) by binding to liposomes containing PS than by that to control

liposomes without PS. We have now described these results in the revised manuscript (page 9, lines 256-260).

(Fig. 4d and 4e) Liposome binding assay for GST-PDZD8(C1) in the presence or absence of PS.

5. The images in Figure 5 and 6 should be quantified.

[Response] As requested, we quantitatively evaluated the extent of abnormal large vacuole (ALV) formation induced by protrudin and PDZD8 in HeLa cells (**new Fig. 5c**). Overexpression of PDZD8 alone did not significantly increase the number of such vacuoles, whereas that of protrudin induced a moderate increase and the combination of the two proteins had a synergistic effect, suggesting that ALV formation is primarily a protrudin-driven phenomenon that is significantly enhanced by PDZD8. We have now modified the text according to these results in the revised manuscript (page 10, lines 274-279).

(Fig. 5c) Quantitative analysis of ALV formation induced by overexpression of protrudin and PDZD8.

Response to Reviewer #3

We thank the reviewer for the careful evaluation of our manuscript and for the statements that “the results presented in this manuscript are of potentially great interest to a broad readership,” and that “The manuscript reads well and the data are presented in a clear and logical way.” We also thank the reviewer for suggestions that we feel have helped us to improve our manuscript.

In the revised manuscript, we have shown that the protrudin-PDZD8 system mediates membrane tethering, MCS formation, and lipid transfer between the ER and endolysosomes as well as demonstrated the physiological relevance of this system in neurons and the pathological relevance of its dysfunction, with the physiological and pathological aspects in particular being newly added. On the basis of our additional findings, we have largely modified the original manuscript, including a change to the title.

Our specific responses to the points raised are as follows:

(This response has low resolution thumbnails for clarity, but please refer to the illustration in the text for high resolution data.)

Major points

1-1. A major concern with this study is the proposed role for protrudin and PDZD8 in endosome maturation shown in Fig. 5 and 6. I am concerned that the enlarged, convoluted vacuoles seen by Protrudin and PDZD8 co-expression are simply an overexpression artefact due to massive ER-endosome contacts, leading to hyper-recruitment of ER membranes which surround the endosome (Fig 5d, Fig 6 d). This will prevent natural endosome maturation, which is maintained through a balance between fusion and fission events, and thus does not represent a physiological role of the PDZD8-protrudin complex in endosome maturation. The morphology of the enlarged MVBs looks aberrant. ILVs typically have a round shape with a diameter around 50 nm (Adell and Teis 2011, Teis 2010, Wenzel 2018). The “ILVs” in the enlarged MVBs caused by protrudin and PDZD8 co-expression have irregular shapes and it is difficult to see from the images whether these putative ILVs are abscised or still connected to the limiting membrane of the endosome as membrane invaginations (Fig. 6c).

[Response] On the basis of evidence obtained during revision (see below), we now agree with the reviewer that the enlarged, convoluted vacuoles seen in HeLa cells coexpressing protrudin and PDZD8 are indeed an artifact of overexpression resulting from excessive formation of ER-LyLE contacts and consequent hyperrecruitment of ER membrane surrounding LyLEs. We also agree that these abnormal vacuoles are likely produced as a result of the prevention of normal endosome maturation achieved through a balance between fusion and fission events. We have thus largely changed the conclusion of our manuscript as outlined below.

The best and simplest approach to investigation of the precise role of protrudin and PDZD8 is to deplete these proteins, rather than overexpress them, in a more physiological setting. To this end, we adopted the same experimental system as that described previously by Allison *et al.* (*J. Cell Biol.*, 2017) and which allows evaluation of endolysosome (LyLE) maturation mediated through a balance between fusion and fission events in mouse primary cortical neurons. We confirmed that the morphology of MVBs appeared normal, with ILVs typically

having a round shape and a diameter of ~50 nm, in such neurons (**new Fig. 7c**). Allison *et al.* showed that defects in ER-LyLE contacts induced by mutations of spastin or REEP1 that cause hereditary spastic paraplegia (HSP) result in failure of endolysosomal fission in mouse primary neurons. It is of particular interest that mutations of the human protrudin gene (*ZFYVE27*) also cause HSP, suggesting that the protrudin-PDZD8 system might also contribute to the endolysosomal maturation process mediated by fusion-fission.

We found that depletion of PDZD8 (which also reduces the abundance of protrudin as shown in **Figure 1g and 1i**) in mouse primary neurons induced frequent gross enlargement of LAMP1⁺ LyLEs (**new Fig. 7a and 7b**). These vesicles showed abnormal ultrastructures with many endolysosomal membrane coils and almost no signs of normal ILVs (**new Fig. 7c**), and they were highly reminiscent of those seen in neurons of spastin or REEP1 mutant mice. Similar abnormalities were observed in mouse primary neurons expressing the PDZD8(Δ SMP) mutant, suggesting that lipid extraction activity of PDZD8 contributes to the normal endosomal maturation process in these cells. Allison *et al.* showed that endosomal maturation is regulated by the balance of fusion and fission, the latter of which takes place at MCSs formed between the ER and LyLEs in a manner regulated by spastin or REEP1. Given the similar molecular functions and structures shared by spastin, REEP1, and protrudin as well as the roles of these proteins in the pathogenesis of HSP, we conclude that MCS formation and lipid transfer activity dependent on the protrudin-PDZD8 system are essential for endolysosomal maturation in the physiological setting. We have now addressed these points in the revised manuscript (page 12, lines 341-360).

(Fig. 7a and 7b) Gross enlargement of LyLEs induced by either depletion of PDZD8 or expression of PDZD8(Δ SMP) in mouse primary neurons.

(Fig. 7c) Ultrastructure of abnormal LyLEs induced by depletion of PDZD8 or expression of PDZD8(Δ SMP) in mouse primary neurons.

The abnormal LyLEs found in PDZD8-depleted mouse primary neurons are also highly similar to those seen in HeLa cells overexpressing both protrudin and PDZD8. As pointed out by the reviewer, this phenotype of HeLa cells is likely attributable to an artifact of overexpression that results in endosomal fission defects and an impaired balance of endosomal homeostasis. Indeed, we observed many abnormal large vacuoles (ALVs) linked together, likely as a result of such a fission defect, in these cells (**Fig. 6f**). We also detected many abnormal MVBs containing typical ILVs and irregular multilammellar structures prior to apparent ALV formation in HeLa cells expressing protrudin and PDZD8 (**Fig. 6g-i**). An abnormal increase in MCS formation between the ER and LyLEs thus likely results in an excess transfer of lipid at these sites that impairs endosome fission and maturation. We have now added these comments to the Discussion section of the revised manuscript (page 11, lines 307-317).

(Fig. 6f) Representative images of linkage between ALVs likely resulting from a fission defect in HeLa cells overexpressing protrudin and PDZD8.

(Fig. 6g–6i) Many abnormal MVBs were detected in HeLa cells overexpressing protrudin and PDZD8.

Collectively, our new findings obtained in a physiological setting and presented in the revised manuscript support the notion that the protrudin-PDZD8 system indeed contributes to endolysosomal maturation by mediating lipid transfer in mouse primary neurons.

1-2. *To address a functional role of the protrudin-PDZD8 synergy in ER-endosome contact sites and a possible role in endosome maturation, the authors need to co-express these proteins close to their endogenous levels, and use deletion mutants of PDZD8, deficient in lipid binding or protrudin binding, (preferably siRNA resistant, in cells depleted for endogenous PDZD8 by siRNA) to address any changes that might occur regarding LE abundance and morphology, the number of ILVs, cargo degradation such as EGFR degradation in lysosomes and retromer dependent recycling of receptors such as the CI-M6PR, as readouts for endosome maturation and functionality.*

[Response] In the original manuscript, we overexpressed protrudin and PDZD8 in HeLa cells by transient transfection. In response to the reviewer's comment, we also examined the effect of stable overexpression of these proteins by retrovirus infection. We measured mRNA abundance for the endogenous and exogenous proteins in both transient and stable expression models. The mRNA levels for exogenous protrudin and PDZD8 in the stably infected cells were ~1% to 10% of those in the transiently transfected cells (**new Supplementary Fig. 9c–9f**). HeLa cells stably expressing exogenous protrudin and PDZD8 at these relatively low levels did not manifest the grossly enlarged vacuoles detected in the transiently transfected cells (**new Supplementary Fig. 12**), consistent with the notion that

the high levels of protrudin and PDZD8 expression in the latter cells result in excessive formation of ER-LyLE MCSs and lipid transfer, leading to impaired endolysosomal homeostasis as a result of an imbalance in fusion and fission events. We have now addressed these points in the revised manuscript (page 10, lines 281-284, and page 11, lines 307-317, page 14, lines 405-406).

(Supplementary Fig. 9c–9f) Quantification of mRNA abundance for the endogenous and exogenous proteins in HeLa cells stably (c, d) or transiently (e, f) expressing exogenous protrudin and PDZD8.

(Supplementary Fig. 12) The relatively low levels of protrudin and PDZD8 expression achieved by retrovirus infection do not induce the formation of grossly expanded vacuoles in HeLa cells.

1-3. *Protrudin has an established role in the regulation of directional endosome transport (Shirane and Nakayama 2006, Raiborg 2015). The authors have not considered whether PDZD8 can influence the role of protrudin in regulation of LE positioning. Whereas the generation of large vacuoles seen by co-expression of high levels of protrudin and PDZD8 in this study hampers the possibility to study endosome positioning, this could easily be tested under more controlled expression levels as suggested above.*

[Response] A previous study (Hong *et al.*, *J. Cell Biol.*, 2017) showed that depletion of protrudin resulted in abnormal LyLE positioning around the nucleus on refeeding of cells after serum deprivation. As suggested by the reviewer, we examined the effect of coexpression of protrudin and PDZD8 in HeLa cells at the relatively low levels achieved by retrovirus infection as described above (**new Supplementary Fig. 9c–9f**). However, we found that such low expression levels of protrudin and PDZD8 did not affect LyLE positioning (**data not shown**). We then depleted HeLa cells of protrudin, PDZD8, or both proteins by siRNA transfection and found that the abnormality of LyLE positioning described by Hong *et al.* was reproduced by protrudin depletion. In contrast, PDZD8 depletion had no such effect, and the phenotype of cells depleted of both proteins was similar to that of those depleted of protrudin alone (**new Supplementary Fig. 16**). We therefore conclude that PDZD8, unlike protrudin, is not involved in LyLE positioning. We have now clarified these points in the revised manuscript (pages 11-12, line 329-339).

(Supplementary Fig. 16) Localization of LyLEs in HeLa cells depleted of protrudin, PDZD8, or both proteins.

1-4. *The statement that the numbers of ILVs are increased should be backed up by quantifications. Tomography should be performed to assess whether the proposed ILVs (Fig. 6c) are abscised or if they represent membrane invaginations of convoluted vacuoles.*

[Response] As pointed out by the reviewer, the abnormal intraluminal structures formed in HeLa cells overexpressing protrudin and PDZD8 do not appear to be typical ILVs. The

grossly enlarged endolysosomes showed abnormal ultrastructures containing many endolysosomal membrane coils and almost no signs of normal ILVs. This phenotype is highly reminiscent of that of mouse primary neurons depleted of PDZD8. We also found many abnormal MVB containing typical ILVs and irregular multilammellar structures in prior to apparent ALV formation in HeLa cells expressing protrudin and PDGD8, both of which likely facilitate endosome maturation process in a cooperative manner (**new Fig. 6g–6i**). These findings thus suggested that excessive amounts of protrudin and PDZD8 cooperatively promote artifactual formation of ALVs as a result of their excess tethering of the ER and LyLEs and subsequent excess lipid transfer from the ER to LyLEs. This lipid transfer likely supplies the membrane components that allow the enlargement of LyLEs, resulting in the development of ALVs instead of normal MVBs. As mentioned above, we therefore conclude that the enlarged, convoluted vacuoles formed in response to protrudin and PDZD8 overexpression in HeLa cells represent an artifact of overexpression due to excessive ER-LyLE contacts, as is now described in the revised manuscript (pages 10-11, lines 294-317).

(Fig. 6g–6i) HeLa cells transiently transfected with control vector (**g**) or expression vectors for GFP-protrudin and PDZD8-GFP (**h**). Quantification of the number of normal and abnormal MVB is shown (**i**).

Further in-depth analysis of these artifactual structures was thus not undertaken, given that it would be unlikely to provide substantial physiological insight. Instead, we focused on the analysis of mouse primary neurons in the revised manuscript.

2-1. Whereas the split-GFP assay used in Fig.2 is an elegant approach to assay MCSs, these

experiments are missing important controls, and should be exploited to study the potential synergy between PDZD8 and protrudin in contact site formation. The author's conclusion that "the protrudin-PDZD8 complex plays a role in formation of MSCs (sic) between the ER and LEs", is not supported by the present data. Whereas the authors nicely show that the stable expression of either protrudin or PDZD8 in HeLa cells promote ER-endosome contact site formation, it will be important to know whether PDZD8 is able to influence the ability of protrudin to form contact sites, or vice versa. This can be achieved by utilizing the split-GFP assay in stable cell lines co-expressing protrudin and PDZD8 WT and deletion mutants as suggested for assaying endosomal function above.

[Response] As suggested by the reviewer, we performed a series of experiments based on the split-GFP assay. First, we examined the effect of expression of both protrudin and PDZD8 in HeLa cells on MCS formation between the ER and LyLEs (**new Fig. 2c**) and between the ER and mitochondria (**new Fig. 2g**). Coexpression of the two proteins showed a synergistic effect on the formation of ER-LyLE contacts but not on that of ER-mitochondria contacts.

Second, we depleted HeLa cells of either protrudin or PDZD8 with specific siRNAs, with the knockdown efficiency being shown in **new Supplementary Fig. 9a and 9b**. Depletion of protrudin or PDZD8 resulted in a significant attenuation of MCS formation between the ER and LyLEs (**new Fig. 2d**). The effects of such depletion on MCS formation between the ER and mitochondria were less pronounced (**new Fig. 2h**), with a modest reduction being observed for cells depleted of protrudin (not significant, likely as a result of the relatively low knockdown efficiency) or of PDZD8 ($P < 0.05$). These results suggest that endogenous protrudin and PDZD8 indeed contribute to MCS formation, with MCSs formed between the ER and LyLEs being more dependent on protrudin and PDZD8 than are those formed between the ER and mitochondria.

Third, we examined the effects of expression of either the PDZD8(Δ C1) mutant, which lacks the C1 domain required for the initial step of membrane tethering, or the CYP-PDZD8 chimeric mutant, which has lost the ability to interact with protrudin (**new Fig. 2e**). As expected, the PDZD8(Δ C1) mutant did not promote the formation of MCSs between the ER and LyLEs. Unexpectedly, however, the CYP-PDZD8 mutant had an effect similar to that of PDZD8(WT), suggesting that, although protrudin and PDZD8 are expressed in proximity to each other and cooperate to promote MCS formation between the ER and LyLEs, direct molecular interaction between the two proteins might not be necessary for this effect.

Collectively, these results suggest that the protrudin-PDZD8 system contributes mostly to the formation of ER-LyLE MCSs. We have now addressed these points in the revised manuscript (pages 7-8, lines 180-201).

(Fig. 2c–2e, 2g, and 2h) Formation of MCSs between the ER and either LyLEs (c–e) or mitochondria (g, h) is modulated by depletion or expression of WT or mutant forms of protrudin and PDZD8 in HeLa cells.

2-2. The stable cell lines used in Fig. 2 should be characterized by Western Blotting to show the level of expression compared to endogenous PDZD8 and protrudin.

[Response] Given that the expression levels of endogenous protrudin and PDZD8 in HeLa cells are low compared with those in the brain cortex or midbrain (**new Supplementary Fig. 9g and 9h**), it is technically difficult to estimate expression levels for the endogenous and exogenous molecules by immunoblot analysis. Instead, as mentioned above, we measured the abundance of endogenous and exogenous mRNAs for these proteins in HeLa cells subjected to transient transfection or retrovirus infection. The exogenous mRNA levels in cells stably expressing protrudin and PDZD8 were ~1% to 10% of those in the transiently transfected cells (**new Supplementary Fig. 9c–9f**). We have addressed these points in the revised manuscript (page 10, lines 281-284; page 12, lines 343-345).

(Supplementary Fig. 9g and 9h) Quantification of protrudin and PDZD8 mRNA abundance in HeLa cells as well as mouse brain cortex and midbrain.

2-3. The GFP-signal from the split GFP-probes used to detect ER-endosome contact sites and ER-mitochondria contact sites should be tested for colocalization with endosomal and mitochondrial markers respectively.

[Response] In response to the reviewer's comment, we performed immunofluorescence analysis with antibodies to LAMP1 or to Tom20 in HeLa cells expressing split-GFP constructs. Split-GFP signals for ER-LyLE contacts overlapped with LAMP1 signals but not with Tom20 signals (**new Supplementary Fig. 8a**), whereas those for ER-mitochondria contacts overlapped with Tom20 signals but not with LAMP1 signals (**new Supplementary Fig. 8b**). These results thus confirmed that the GFP signals used to detect ER-LyLE and ER-mitochondria MCSs localize to the expected organelles, as is now described in the revised manuscript (page 7, lines 173-175).

(Supplementary Fig. 8a and 8b) Colocalization of split-GFP signals with organelle markers.

2-4. *The results indicate that protrudin can mediate contact between ER and mitochondria in addition to endosomes. This potentially additional role of protrudin should be commented on and discussed. Do the authors think that this reflects a true function of protrudin? Alternatively it could reflect mitophagocytic events, where damaged mitochondria, induced by the GFP-probe, are in close apposition to the ER. This could be easily tested by using autophagosomal markers like LC3.*

[Response] As mentioned above, protrudin appears to have much less impact on the formation of ER-mitochondria contacts than on that of ER-LyLE contacts, on the basis of the effects of its overexpression (**new Fig. 2c and 2g**) or depletion (**new Fig. 2d and 2h**). As suggested by the reviewer, HeLa cells coexpressing protrudin and PDZD8 were subjected to immunofluorescence (**new Supplementary Fig. 13**) and immunoblot (**new Supplementary Fig. 14**) analyses. The former revealed that coexpression of protrudin and PDZD8 slightly increased the LC3 signal (**new Supplementary Fig. 13a, b**), suggestive of an increased extent of autophagy. However, signals for the EE marker EEA1 (**new Supplementary Fig. 13c**), the LyLE marker LAMP1 (**new Supplementary Fig. 13d, e**), and the lysosomal tracer Magic red (**new Fig. 6l**) were also all increased, suggesting that the increase in LC3 immunoreactivity might reflect disruption of endosomal homeostasis by overexpression of protrudin and PDZD8. Similar results were obtained by immunoblot analysis (**new Supplementary Fig. 14**). We have now mentioned these points in the revised manuscript (pages 11-12, lines 318-339).

(Supplementary Fig. 13) Immunofluorescence analysis of LC3, EEA1, and LAMP1 in HeLa cells coexpressing protrudin and PDZD8.

(Supplementary Fig. 14) Immunoblot analysis of LC3, EEA1, and LAMP1 in HeLa cells coexpressing or depleted of protrudin and PDZD8.

(Supplementary Fig. 15) The intensity of Magic red representing lysosomal activity in HeLa cells expressing protrudin and PDZD8.

2-5. The area of cells (%) covered by GFP-dots is around 10% for the endosome probe (Fig2. d,e) and 0.15% for the mitochondria probe (Fig. 2h). These numbers do not fit their relative immune fluorescence images (Fig. 2 c,g). Why is the area measured for mitochondria contacts so much lower, although it looks more from the IF?

[Response] The mitochondrial data pointed out by the reviewer were incorrectly documented in the original manuscript, with the actual values being 100 times those reported (15%). All experiments were repeated to allow for more accurate quantification based on normalization by mCherry fluorescence, as is now described in the revised manuscript (page 7, lines 173-180).

3-1. Fig. 4c suggests a model where the PDZD8-C1 domain promotes ER-endosome contact by binding to endosomal PS. Whereas this is a nice hypothesis, this model has to be supported by experimental data. The authors should use their split-GFP assay and compare PDZD8 WT and deltaC1 to test this.

[Response] We examined ER-LyLE MCS formation by the split-GFP assay in HeLa cells expressing the PDZD8(Δ C1) mutant and found that the mutant protein had lost the ability to promote such contact compared with PDZD8(WT) (**new Fig. 2e**). We thus conclude that the C1 domain is indispensable for the ability of PDZD8 to promote the formation of MCSs between the ER and LyLEs. We have now clarified this point in the revised manuscript (page 7, lines 185-188).

(Fig. 2e) Formation of MCSs between the ER and LyLEs in HeLa cells expressing PDZD8 mutants.

3-2. In the model PDZD8 forms contact with endosomes through binding to PS. If PDZD8 extracts or transfers PS between membranes, how could the binding to PS be utilized to form the contact? The more logical explanation would perhaps be that since PDZD8 is in a complex with protrudin, protrudin is providing the contact, whereas PDZD8 mediates PS

extraction. This could be tested by investigating whether the PDZD8-mediated contact site formation observed in the split-GFP assay is sensitive to the VPS34 inhibitor SAR405, which should abolish protrudin-mediated contact site formation.

[Response] In response to the reviewer's comment, we explored the physiological relevance of PDZD8 binding to PS. As mentioned above, whereas PDZD8(WT) promoted ER-LyLE MCS formation, the PDZD8(Δ C1) mutant had lost this ability (**new Fig. 2e**). Furthermore, we now show that the C1 domain of PDZD8 binds not only to PS but also to PtdIns(4)P (**new Fig. 4a and 4b**), which is enriched in the LyLE membrane. These results suggest that the C1 domain of PDZD8 plays a key role in MCS formation as an initial step of membrane tethering between the ER and LyLEs. Consistent with these observations, the PDZD8(Δ C1) mutant was found to be less effective at inducing the formation of large vacuoles in HeLa cells compared with the wild-type protein (**new Fig. 5e and 5f**). We have now addressed this issue in the revised manuscript (page 7, lines 185-191; page 10, lines 285-293).

(Fig. 4a and 4b) Lipid strip assays for GST-PDZD8(Δ C1).

(Fig. 5e and 5f) Large vacuole formation induced by coexpression of protrudin and either WT or mutant forms of PDZD8 in HeLa cells.

3-3. The model in Fig. 4c also puts forward the possible need of Ca²⁺ for the PDZD8 mediated ER-endosome contact site formation. Given that PDZD8 has an established role in Ca²⁺ homeostasis and transfer of Ca²⁺ from ER to mitochondria, and that other ER-endosome contact sites depend on Ca²⁺ (Eden 2016), this is a relevant hypothesis which should be tested using a Ca²⁺ chelator in the same assay.

[Response] No Ca²⁺ was included in our in vitro assays of lipid binding, so we did not examine the effect of a Ca²⁺ chelator. Our results thus suggest that Ca²⁺ is not required for membrane tethering, at least in vitro, and we have now modified the model by replacing “Ca²⁺” with “Signal” to avoid any misunderstanding (**new Fig. 4i**). It is of note that the lipid extraction activity of PDZD8 was increased by loss of the CC domain (**new Fig. 4f–4h**), suggesting that the CC domain inhibits this activity. A conformational change induced by an unknown signal may therefore increase lipid extraction activity mediated by the SMP domain by allowing the binding of the C1 domain to endolysosomal lipids. As with other SMP proteins, the efficiency of lipid extraction by PDZD8 may vary physiologically in a signal-dependent manner (**new Fig. 4i**). We have now addressed these issues in the revised manuscript (pages 9-10, lines 265-270).

(Fig. 4f–4i) The CC domain of PDZD8 inhibits ER-LyLE tethering. Loss of the CC domain facilitates both the amount and velocity of PS extraction by PDZD8 (f–h). Our hypothetical model is also shown (i).

Minor points

1. In Fig. 2 the number of GFP dots and the area of dots per cell should be quantified from three independent experiments, and data shown as mean between independent experiments.

[Response] As suggested, we quantified the number of GFP dots and the area of dots per cell from three independent experiments (**new Fig. 2c–2e, 2g, 2h**). We also improved the accuracy of quantification by normalizing transfection efficiency for each cell on the basis of the fluorescence of mCherry derived from a cotransfected vector (**new Fig. 2a**). We have included this information in the revised manuscript (pages 7-8, lines 177-201).

a**(Fig. 2a) Schematic representation of the split-GFP assay.****(Fig. 2c–2e, 2g, 2h) Formation of MCSs between the ER and either LyLEs (c–e) or mitochondria (g, h) in HeLa cells is modulated by depletion or expression of WT or mutant forms of protrudin and PDZD8.**

2-1. In Fig. 3, the authors conclude that PDZD8 has lipid extraction activity. This is especially tested in the experiments in Fig. 3 h-k. In the title of this manuscript as well as in several statements throughout the text and in the model in fig. 7, however, the authors claim that PDZD8 and its role in membrane contact sites and endosome maturation is explained by its lipid transfer activity and that PDZD8 transfer lipids from the ER to the endosome. The authors should modify their statements according to their data.

[Response] As suggested by the reviewer, we modified the text by changing “lipid transfer

activity” to “lipid extraction activity” when referring to the molecular function of PDZD8 alone. However, many lines of evidence in our study suggest that the protrudin-PDZD8 system as a whole serves to transfer lipid mainly through ER-LyLE contacts. We now use this term appropriately depending on the context throughout the revised manuscript.

2-2. If PDZD8 extracts PS and other lipids, where do these lipids end up?

[Response] PDZD8 is an ER protein, and the extracted lipid is expected to be transferred from the ER, as a donor membrane, to LyLEs and mitochondria, as acceptor membranes.

3. In Fig. 4, it would also be informative to compare the lipid binding specificities of PDZD8 WT, SMP and PDZ with C1 in the lipid overlay assay. The authors state that the PDZD8-C1 binds specifically to PS. There is, however, binding to several phosphoinositides, especially PI4P. This should be discussed. How many times was the lipid overlay assay performed? Since lipid overlay assays can be rather unspecific, the binding of PDZD8 to PS should be backed up in a liposome-assay.

[Response] We examined the lipid binding specificity of PDZD8 with lipid overlay assays and found that the C1 domain of PDZD8 associated strongly with PS and PtdIns(4)P, both of which are concentrated in the endolysosomal membrane (**Fig. 4a, b**). We also now show the binding of the C1 domain to PS with a custom lipid strip (**new Fig. 4c**). Such experiments were repeated more than five times, and representative results with different lots of lipid strips are also now shown (**new Supplementary Fig. 11**). Given that, as the reviewer points out, lipid overlay assays can be relatively unspecific, we backed up the data obtained with these assays with a liposome binding assay. The latter confirmed that GST-PDZD8(C1) substantially bound to PS in vitro (**new Fig. 4d and 4e**). We have addressed these points in the revised manuscript (page 9, lines 253-260).

(Fig. 4c) Lipid overlay assay for GST-PDZD8(C1) with PA, PC, PE, and PS.

(Supplementary Fig. 11a) Lipid overlay assay for GST-PDZD8(C1) with various membrane lipids.

(Supplementary Fig. 11b) Quantitative lipid overlay assay for GST-PDZD8(C1) and PA, PC, PE, and PS.

(Fig. 4d and 4e) Liposome binding assay for GST-PDZD8(C1) with or without PS.

4-1. Whereas it is evident from their data in Figure. 5 and 6, that co-expression of protrudin and PDZD8 promote the formation of enlarged late endosomal structures, there is no data to support that this is caused by the proposed lipid transfer activity of PDZD8 as claimed in the title. This should be tested by co-expressing protrudin with deletion mutants of PDZD8 which reduce lipid extraction activity, such as the *deltaSMP* and the *deltaPDZ*.

[Response] As suggested, we coexpressed protrudin and PDZD8 mutants in HeLa cells and examined the number and size of large LyLEs, which are now referred to as abnormal large vacuoles (ALVs), with the use of immunofluorescence analysis (**new Fig. 5e and 5f**). Coexpression of PDZD8(Δ SMP) with protrudin resulted in the formation of significantly fewer ALVs compared with cells coexpressing PDZD8(WT) and protrudin, suggesting that SMP-mediated lipid extraction by PDZD8 is essential for the formation of the giant vacuoles in HeLa cells.

We also performed similar experiments with mouse primary neurons. The number of enlarged LyLEs was increased in neurons in which PDZD8 was depleted or PDZD8(Δ SMP) was expressed, suggestive of a defect in LyLE fission (**new Fig. 7a and 7b**). In this case, PDZD8(Δ SMP) likely exerts a dominant negative effect on the normal fission process. In addition, TEM analysis revealed the presence of normal MVBs in control neurons, but few normal MVBs were observed in cells in which PDZD8 was depleted or PDZD8(Δ SMP) was expressed, with many LyLEs with enlarged ILVs and irregular multilamellar-like structures being detected instead (**new Fig. 7c**). Some of these enlarged LyLEs contained both normal-size ILVs ($< 0.1 \mu\text{m}$) and abnormal multilamellar structures, suggesting that they arose as a result of defective LyLE maturation. The large LyLEs observed in mouse primary neurons closely resembled the ALVs observed in HeLa cells overexpressing protrudin and PDZD8.

Collectively, our results with both HeLa cells and mouse primary neurons indicate that PDZD8(Δ SMP), which manifests reduced lipid extraction activity, is defective in the ability to mediate lipid transfer at ER-LyLE contacts. They also suggest that PDZD8 is essential for

the normal maturation process of LyLEs in neurons, which involves lipid transport from the ER to LyLEs. We have now addressed these issues in the revised manuscript (page 10, lines 285-293; page 12, lines 351-360).

4-2. Also the number of early and late endosomes should be quantified following these more targeted perturbations to assess the functions of protrudin and PDZD8 in endosome maturation.

[Response] As mentioned above, HeLa cells coexpressing protrudin and PDZD8 were subjected to immunofluorescence analysis of LC3 as an autophagy marker (**new Supplementary Fig. 13a, b**), of EEA1 as an EE marker (**new Supplementary Fig. 13c**), and of LAMP1 as a LyLE marker (**new Supplementary Fig. 13d, e**), as well as to fluorescence microscopic analysis of Magic red as a tracer of lysosomal activity (**new Supplementary Fig. 15**). The signals for LC3, EEA1, LAMP1, and lysosomal activity were all increased by expression of both protrudin and PDZD8, suggesting that the increase in the LC3 signal may reflect disruption of endosomal homeostasis. Similar results were obtained by immunoblot analysis (**new Supplementary Fig. 14**). These results are now described in the revised manuscript (page 11, lines 318-328).

5. In the introduction, the authors write the following statement: “Such endosome maturation is associated with expansion of the LE membrane, with the lipids required for this expansion being thought to be supplied by the ER at MCSs”. Clearly ER-endosome contact sites are important in the regulation of endosome maturation and function, and transfer of cholesterol between the two membranes. Loss of ER-endosome contact sites lead to impaired ESCRT-mediated ILV formation and enlarged MVEs (Eden 2016, Kobuna 2010). Endosomal membrane dynamics are, however, mainly maintained by a combination of endosome fusion and fission of recycling tubules. The authors should provide specific references for their statement regarding the requirement for ER-endosome contact sites in LE membrane expansion.

[Response] We apologize for the inappropriate statement in the Introduction of the original manuscript. We have now rewritten the Introduction, stating that endosomal membrane dynamics are mainly attributable to a combination of endosome fusion and fission, with appropriate reference citation (page 3, lines 51-57).

6. To the best of my knowledge, the phenotypes of the protrudin- or PDZD8 KO mice have never been described. It would be informative to provide some information about this in the manuscript or to referees, if the authors want to save this information for a separate publication.

[Response] We have already described the neuronal phenotype of protrudin knockout mice (Ohnishi *et al.*, *Genes Cells*, 2014). In brief, neurons deficient in protrudin manifest a defect in cell polarity, with the balance between axons and dendrites being impaired and the size of the soma increased. Investigation of the phenotypes of PDZD8 knockout mice is now under way and will be reported in the future. Instead, we now show that depletion of PDZD8 or expression of PDZD8(Δ SMP) in mouse primary neurons results in an increase in the size of the soma (**new Fig. 8**), recapitulating the defect of neurons derived from protrudin knockout mice. However, it is difficult in practice to estimate the effect of either specific deficiency,

given that knockdown of PDZD8 inevitably leads to destabilization of protrudin, and vice versa, resulting in a concomitant decrease in the amounts of both proteins.

(Fig. 8) Polarity defect in mouse primary neurons depleted of PDZD8 or expressing PDZD8(ΔSMP).

7. The model in fig. 7 states that ILVs “increase”. Unless the data on ILVs is severely strengthened (see major point 1), no statement on ILV morphology, size or number can be given in the model.

[Response] TEM revealed that the abnormal large vacuoles (ALVs) that appeared in HeLa cells overexpressing both protrudin and PDZD8 were highly similar to the grossly expanded endolysosomes that formed in neurons depleted of PDZD8. In particular, in the case of the neurons, normal MVBs found in control cells were essentially not detected in PDZD8-depleted cells, which instead showed a marked increase in large multilamellar-like structures. We have now removed the label “ILVs: Increase” from the model (**new Fig. 9**).

8. In all figures showing insets/magnifications, the corresponding area should be highlighted by a box in the original image.

[Response] As suggested, we have highlighted the magnified area in the original images with a box.

Reviewers' comments:

Reviewer #1 (Remarks to the Author):

In this revised manuscript, Shirane et al., have added new evidences strengthening a novel role of Protrudin and PDZD8 in promoting lipid transfer that is required for endosome maturation. The authors have made an extensive revision of the manuscript and addressed my initial concerns. The authors have also added new data performed in primary cultures of mouse neurons, in which knockdown of PDZ8 or expression of PDZ8 lacking the ability to transfer lipids, cause endosome maturation defects in neurons. Part of the work performed in neurons is novel and of great relevance for the neurobiology field. However, the conclusions about PDZ8-protrudin system and its role in preventing neurodegeneration and its essential role for the establishment of neuronal polarity are incorrect. Some statements and data should be removed from the manuscript.

The authors have made two strong statements regarding their work in neurons:

1. "The protrudin-PDZD8 axis is essential for endosome maturation and prevention of neurodegeneration in neurons" The authors analyzed the effect of KD PDZD8, PDZD8-WT and PDZD8 Δ SMP in endosome maturation in primary cultures of mouse neurons. Although shRNA against mouse protrudin is listed in Methods, protrudin-KD was not analyzed in this experiment. Does protrudin-KD or protruding plus PZD8 co-expression cause same phenotype of enlarged LyLEs than PDZ8-KD? Is the phenotype caused by PDZD8-KD rescued by PDZD8-WT and PDZD8- Δ SMP? My major concern is that there is not any data suggesting a role of protrudin-PDZ8 system in prevention of neurodegeneration. Authors should remove "prevention of neurodegeneration" from the sentence.
2. "The protrudin-PDZD8 system is required for establishment of neuronal polarity" The data presented here is of poor quality. Also, it is not analyzed the role of the protrudin-PDZD8 system. The images selected do not show the neurons including their entire axon (Fig. 8a), so it is difficult to say something about neuronal polarity. Moreover, neurons at DIV3-4 have lot of variation in the length of axon and dendrites per neuron. total length of branched axon and dendrites need to be quantified of at least 30 neurons per condition to then select a representative image. Also, I do not understand why the authors consider the soma size as an indicator of neuronal polarity (Fig. 8b-f). Quantification of axon and dendrite length is typically used to analyze neuronal polarity. In addition, for the experiments in Fig 8 is indicated in Methods/Legend that nucleofection was followed by staining for Tau/MAP2 or Tubb3. It is difficult to ensure that the imaged neurons were transfected without using a marker for transfected cells. Even using nucleofector system, the percentage of neurons transfected is around 60-70%.

In the title and abstract are highlighted these findings in neurons that are not properly added in this study, as I mentioned above. I strongly recommend the authors remove the statements of the role of protrudin-PDZ8 in prevention of neurodegeneration and the data for neuronal polarity (Figure 8).

Reviewer #2 (Remarks to the Author):

More work is required to address my concerns about this story. The idea that PDZD8 transfers lipids

from the ER to endosomes and is required for endosome membrane expansion is intriguing, but the evidence remains unpersuasive.

1. As I said in my previous review, it is important to show that PDZD8 is at ER-endosome contact sites in unperturbed cells, by a method that does not rely on split GFP. The rebuttal letter points out, correctly, that this is very challenging for these contacts. Nonetheless, some evidence that endogenous PDZD8 is at these contacts is necessary, in my opinion, for this study to be appropriate for Nature Communications or a similar journal; because the authors are proposing that PDZD8 is required for endosomal membrane expansion, it is important to know whether the endogenous protein is enriched at ER-endosome contacts, which seems necessary for it to provide significant amounts of lipid to endosomes. Alternatively, it is possible that only a small amount of PDZD8 at ER-endosome contacts is sufficient for it to facilitate endosome membrane expansion, but evidence would need to be provided. The authors also argue that PDZD8 plays a role in maintaining contacts, but this still needs to be verified by a method that does not use split GFP reporters. Even the split-GFP data lacks controls to show that over expression or knock down of PDZD8 does not change expression levels of the reporters.

2. The evidence that PDZD8 can extract lipids and transport them between membranes in vitro is still not convincing. There are many issues that need to be addressed.

a. The addition of quantitative transport rate data is helpful (rather than just showing changes in fluorescence), but initial transport velocity should be calculated. Does the initial velocity change as expected when the concentrations of the donor membranes, transported lipids, and protein changes? It is not possible to tell from the data provided.

b. For all the lipids tested, transport rates appear to be biphasic. There is an initial rate that is very high and then a much slower rate. This should be addressed. Could the seemingly fast initial rate actually be caused by liposome aggregation, fusion, or something unrelated to transport?

c. Since the acceptor liposomes are 4-fold more abundant than donor liposomes, most of the NBD lipid should be transferred to the acceptors, but this was not found and, in fact, only a small fraction of the NBD lipid is moved to acceptor liposomes. What is the explanation?

d. The extraction data (Fig. 3g and others) is hard to understand. What causes the initial increase in fluorescence? What is the explanation for the slower subsequent rate? Why are acceptor vesicles included in some cases?

e. All of the transport data is with NBD lipids. It is well established that NBD lipids do not behave exactly like endogenous lipids and are probably easier for transport proteins to extract and move between membranes because they are more hydrophilic than endogenous lipids. At a minimum, this should be discussed but it would be better to measure transport rates for lipids without NBD (or BODIPY) groups. This has been done with lipid-binding reporters. For example, with a PS-binding C2 domain (PMID: 26206936).

3. This study argues that PDZD8 plays a role in endosome membrane expansion by facilitating lipid transfer to endosomes. There is still no direct evidence that this occurs in cells. As the authors say in the rebuttal letter, estimating lipid transport in cells is challenging, but it has been done. There are various methods that could be used but a number of studies in recent years have used fluorescent lipid-binding reporters and similar methods could be employed here.

Reviewer #3 (Remarks to the Author):

The authors have addressed all my concerns, and I recommend publication of the revised manuscript.

The authors have made an impressive amount of new experiments, which have significantly strengthened the conclusions of this manuscript.

Important controls have been performed and the introduction of the more physiological measurements of endosome integrity and polarity of neurons, have added important value to the biological role of the protrudin-PDZD8 system.

Although it is convincing from the data the protrudin-PDZD8 system is maintaining endosome integrity, important for neuronal polarity, the precise mechanism how protrudin and PDZD8 cooperate to perform this function is still missing, since, surprisingly, the CYP-PDZD8 chimera, unable to bind protrudin, did not show any effects in the assays tested. Hopefully, this will be resolved in future studies.

Response to Reviewer #1

We thank the reviewer for the careful evaluation of our manuscript and for the statement that “The authors have made an extensive revision of the manuscript and addressed my initial concerns.” We also thank the reviewer for suggestions that we feel have helped us to improve our manuscript. Our specific responses to the points raised are as follows (low-resolution thumbnails of new figures are included here, with the corresponding high-resolution data being submitted with the manuscript):

Major points

1-1. *“The protrudin-PDZD8 axis is essential for endosome maturation and prevention of neurodegeneration in neurons”* The authors analyzed the effect of KD PDZD8, PDZD8-WT and PDZD8 Δ SMP in endosome maturation in primary cultures of mouse neurons. Although shRNA against mouse protrudin is listed in Methods, protrudin-KD was not analyzed in this experiment. Does protrudin-KD or protruding plus PZD8 co-expression cause same phenotype of enlarged LyLEs than PDZ8-KD? Is the phenotype caused by PDZD8-KD rescued by PDZD8-WT and PDZD8- Δ SMP?

[Response] In response to this comment, we depleted mouse primary neurons of protrudin with the use of specific siRNAs, and we found that depletion of protrudin induced frequent gross enlargement of LAMP1⁺ LyLEs (**new Fig. 6a, b**) similar to that seen in neurons depleted of PDZD8. The phenotype induced by PDZD8 depletion was indeed rescued by expression of an siRNA-resistant form of PDZD8(WT), but not by that of an siRNA-resistant form of PDZD8(Δ SMP) (**new Fig. 6c, d**). Coexpression of protrudin and PDZD8 in mouse primary neurons did not induce the formation of enlarged LyLEs (**new Supplementary Fig. 13a, b**), likely because neurons express the endogenous proteins at a much higher level than do HeLa cells and are resistant to the effects of a further increase in expression levels. These additional results reinforce our conclusion that the protrudin-PDZD8 system regulates LyLE dynamics through lipid transfer activity. We have now addressed these points in the revised manuscript (pages 11-12, lines 329-342).

(Fig. 6a and 6b) Depletion of protrudin or PDZD8 in mouse primary neurons induced frequent gross enlargement of LAMP1⁺ LyLEs.

(Fig. 6c and 6d) The phenotype induced by PDZD8 depletion was rescued by expression of PDZD8(WT) but not by that of PDZD8(Δ SMP).

(Supplementary Fig. 13a, b) Coexpression of protrudin and PDZD8 did not induce abnormal enlargement of LyLEs in mouse primary neurons.

1-2. *My major concern is that there is not any data suggesting a role of protrudin-PDZ8 system in prevention of neurodegeneration. Authors should remove “prevention of neurodegeneration” from the sentence.*

[Response] The facts that mutations of the human protrudin gene (*ZFYVE27*) cause HSP and that protrudin, spastin, and REEP1 are all ER-resident proteins that contain a hairpin domain suggest that the protrudin-PDZD8 system might contribute to maintenance of neuronal integrity, and that dysfunction of this system might result in neurodegeneration. We found that mouse primary neurons depleted of protrudin or PDZD8 indeed manifested a morphology that is consistent with axonal degeneration and characterized by thinning of axons as well as a reduced level and disrupted pattern of Tau1 staining (**new Fig. 8a**). Furthermore, we examined the colocalization of Tau1 and α -tubulin (microtubules) within axons, given that Tau1 has been shown to dissociate from microtubules during axonal

degeneration (ref. 40). Colocalization of Tau1 and α -tubulin was indeed reduced by depletion of protrudin or PDZD8 in mouse primary neurons (**new Fig. 8b, c**), suggesting that the protrudin-PDZD8 complex contributes to suppression of neuronal degeneration. We have now addressed this issue in the revised manuscript (pages 12-13, lines 359-373). Although we believe that our new results demonstrate a role for the protrudin-PDZD8 complex in prevention of neuronal degeneration, we agree with the reviewer that further in-depth studies are necessary to support this conclusion. We have therefore removed the words “preventing neurodegeneration” from the Abstract and “prevention of neurodegeneration” from the Introduction and the subheading of the Results section.

(Fig. 8a–c) Depletion of protrudin or PDZD8 in mouse primary neurons promoted axonal degeneration.

2. “The protrudin-PDZD8 system is required for establishment of neuronal polarity” The data presented here is of poor quality. Also, it is not analyzed the role of the protrudin-PDZD8 system. The images selected do not show the neurons including their

entire axon (Fig. 8a), so it is difficult to say something about neuronal polarity. Moreover, neurons at DIV3-4 have lot of variation in the length of axon and dendrites per neuron. total length of branched axon and dendrites need to be quantified of at least 30 neurons per condition to then select a representative image. Also, I do not understand why the authors consider the soma size as an indicator of neuronal polarity (Fig. 8b-f). Quantification of axon and dendrite length is typically used to analyze neuronal polarity. In addition, for the experiments in Fig 8 is indicated in Methods/Legend that nucleofection was followed by staining for Tau/MAP2 or Tubb3. It is difficult to ensure that the imaged neurons were transfected without using a marker for transfected cells. Even using nucleofector system, the percentage of neurons transfected is around 60-70%. In the title and abstract are highlighted these findings in neurons that are not properly added in this study, as I mentioned above. I strongly recommend the authors remove the statements of the role of protrudin-PDZ8 in prevention of neurodegeneration and the data for neuronal polarity (Figure 8).

[Response] We apologize for the poor quality and quantification of the original data. Polarized neurons are divided into two regions, the axon and somatodendrite. A reduction in the extent of polarity is associated with shortening of the axon and enlargement of the somatodendrite area. We performed immunofluorescence analysis of mouse primary neurons at DIV5 ($n = 30$ per condition) with antibodies to Tau1 (axon marker) and to Map2 (somatodendrite marker) in order to quantify axon length and somatodendrite area, respectively. Depletion of protrudin or PDZD8 was indeed associated with attenuation of neuronal polarity (**new Fig. 7a–c**). We have now addressed this issue in the revised manuscript (page 12, lines 351-354).

(Fig. 7a–7c) Depletion of protrudin or PDZD8 in mouse primary neurons reduced the extent of neuronal polarity.

Response to Reviewer #2

We thank the reviewer for the careful evaluation of our manuscript. We also thank the reviewer for suggestions that we feel have helped us to improve our manuscript. Our specific responses to the points raised are as follows (low-resolution thumbnails of new figures are included here, with the corresponding high-resolution data being submitted with the manuscript):

1. *As I said in my previous review, it is important to show that PDZD8 is at ER-endosome contact sites in unperturbed cells, by a method that does not rely on split GFP. The rebuttal letter points out, correctly, that this is very challenging for these contacts. Nonetheless, some evidence that endogenous PDZD8 is at these contacts is necessary, in my opinion, for this study to be appropriate for Nature Communications or a similar journal; because the authors are proposing that PDZD8 is required for endosomal membrane expansion, it is important to know whether the endogenous protein is enriched at ER-endosome contacts, which seems necessary for it to provide significant amounts of lipid to endosomes. Alternatively, it is possible that only a small amount of PDZD8 at ER-endosome contacts is sufficient for it to facilitate endosome membrane expansion, but evidence would need to be provided. The authors also argue that PDZD8 plays a role in maintaining contacts, but this still needs to be verified by a method that does not use split GFP reporters. Even the split-GFP data lacks controls to show that over expression or knock down of PDZD8 does not change expression levels of the reporters.*

[Response] As suggested by the reviewer, we attempted to examine whether PDZD8 is located at ER-endosome contact sites in unperturbed cells by a method that does not rely on split-GFP. We therefore performed immunofluorescence analysis with antibodies to PDZD8, to PDI (a marker for the ER), and to LAMP1 (a marker for LyLEs) with the use of super-resolution microscopy (**new Fig. 2a–c**). The images revealed that endogenous PDZD8 is indeed located at MCSs between the ER and LyLEs in mouse primary neurons. These results thus reinforce our conclusion that PDZD8 plays a key role in tethering between the ER and LyLEs. We have now addressed this issue in the revised manuscript (pages 6-7, lines 163-172).

(Fig. 2a–2c) Localization of PDZD8 at MCSs between the ER and LyLEs in mouse primary neurons.

2-a. *The addition of quantitative transport rate data is helpful (rather than just showing changes in fluorescence), but initial transport velocity should be calculated. Does the initial velocity change as expected when the concentrations of the donor membranes, transported lipids, and protein changes? It is not possible to tell from the data provided.*

[Response] In response to the reviewer's comment, we calculated the initial transport velocity (first 10 s of the reaction) (**new Supplementary Fig. 10e**), which changed in response to changes in the lipids and proteins tested. The observed changes were roughly proportional to those in extraction rate at half-max and to those in the total amount of lipid transported. It was necessary that the concentration of the donor membrane be identical throughout all the experiments, and so it was not changed. We have now described these new results in the revised manuscript (page 9, lines 251-253).

e Initial (10 s) velocity (nM lipid/min) [NBD-lipid: 500 nM]

Fig. 3c	Lipid	PA	PS	PE	PC
	Protein	Δ TM, 10 nM	Δ TM, 10 nM	Δ TM, 10 nM	Δ TM, 10 nM
	nM lipid/min	353.3	972.9	393.0	355.9

Fig. 3h	Lipid	PS (+ Acc)	PS (+ Acc)	PS (+ Acc)	PS (- Acc)	PS (- Acc)	PS (- Acc)
	Protein	Δ TM, 5 nM	Δ TM, 10 nM	Δ TM, 20 nM	Δ TM, 5 nM	Δ TM, 10 nM	Δ TM, 20 nM
	nM lipid/min	190.2	1120.0	1565.1	164.5	972.9	1378.8

Fig. 3j, k	Lipid	Cer	Cer	Chol	Chol
	Protein	Δ TM, 2 nM	Δ TM, 10 nM	Δ TM, 2 nM	Δ TM, 10 nM
	nM lipid/min	188.6	804.8	6.9	20.0

Fig. 3f (PS)	Lipid	PS	PS	PS	PS	PS	PS
	Protein (5 nM)	Δ TM	Δ TM Δ SMP	Δ TM Δ PDZ	Δ TM Δ PRT	Δ TM Δ C1	SMP
	nM lipid/min	184.4	72.4	121.6	153.1	199.5	21.8
							SMP-PDZ
							105.4

(Supplementary Fig. 10e) The initial transport velocity of lipid transfer or extraction by PDZD8.

2-b. For all the lipids tested, transport rates appear to be biphasic. There is an initial rate that is very high and then a much slower rate. This should be addressed. Could the seemingly fast initial rate actually be caused by liposome aggregation, fusion, or something unrelated to transport?

[Response] We believe that the presence or absence of insertion activity (that is, both extraction and insertion or only extraction) of the lipid transport protein determines whether the reaction rate is monophasic or biphasic (**Figure R1, for reviewing purposes only**). If a lipid transport protein possesses both extraction and insertion activities (**left**), it is likely that it will be recycled after lipid has been inserted into the acceptor membrane and that the reaction will proceed in an almost monophasic manner until the labeled lipid achieves equilibrium between the donor and acceptor liposomes. On the other hand, if the lipid transport protein possesses only extraction activity (**right**), the reaction will slow down and become biphasic because the transport protein becomes saturated with lipid. We have shown that PDZD8 possesses only extraction activity (**Fig. 3g**), which is consistent with a biphasic lipid transport pattern. Furthermore, increasing the amount of PDZD8 in this reaction system increased the amount of lipid transported until the reaction was saturated, supporting the above notion. We have now addressed this point in the revised manuscript (page 9, lines 249-251).

(Figure R1) Difference in reaction progression between lipid transport proteins that possess both extraction and insertion activities (left) or only extraction activity (right).

2-c. *Since the acceptor liposome are 4-fold more abundant than donor liposomes, most of the NBD lipid should be transferred to the acceptors, but this was not found and, in fact, only a small fraction of the NBD lipid is moved to acceptor liposomes. What is the explanation?*

[Response] The observation that only a portion of the NBD-labeled lipid is transported into acceptor liposomes may be attributable to the low concentration of the transport protein. As mentioned above, an increase in the concentration of the transport protein results in an increase in activity (**Figure R2, for reviewing purposes only**). We have thus shown that the lipid transport activity of PDZD8 depends on the concentration of the protein (**Fig. 3h, i**). We have addressed this point in the revised manuscript (page 9, lines 251-253).

(Figure R2) Lipid transport activity depends on the concentration of the transport protein.

2-d. *The extraction data (Fig. 3g and others) is hard to understand. What causes the initial increase in fluorescence? What is the explanation for the slower subsequent rate? Why are acceptor vesicle included in some cases?*

[Response] See the above responses.

2-e. *All of the transport data is with NBD lipids. It is well established that NBD lipids do not behave exactly like endogenous lipids and are probably easier for transport proteins to extract and move between membranes because they are more hydrophilic than endogenous lipids. At a minimum, this should be discussed but it would be better to measure transport rates for lipids without NBD (or BODIPY) groups. This has been done with lipid-binding reporters. For example, with a PS-binding C2 domain (PMID: 26206936).*

[Response] PDZD8 is an SMP protein, and almost all previous studies of lipid transport by SMP proteins have used NBD-lipids (AhYoung *et al.*, *Proc. Natl. Acad. Sci. USA*, 2015; Watanabe *et al.*, *Nat. Commun.*, 2015; Bian *et al.*, *EMBO J.*, 2017; Kawano *et al.*, *J. Cell Biol.*, 2017). As far as we are aware, the experiment suggested by the reviewer has not been adopted in any such previous paper. Lipid transfer by SMP proteins is thus generally measured precisely with the use of NBD-lipids. Although we appreciate the constructive suggestion of the reviewer, we believe that such an experiment is beyond the standard normally required.

Our lipid transport experiments used many lipids, including PA, PS, PE, PC, ceramide, and cholesterol. The system proposed by the reviewer for monitoring lipid transport with a lipid-binding protein would not be realistic for measuring the transport kinetics for many types of lipids, given that reporters for all such lipids have not been established.

3. *This study argues that PDZD8 plays a role in endosome membrane expansion by*

facilitating lipid transfer to endosomes. There is still no direct evidence that this occurs in cells. As the authors say in the rebuttal letter, estimating lipid transport in cells is challenging, but it has been done. There are various methods that could be used but a number of studies in recent years have used fluorescent lipid-binding reporters and similar methods could be employed here.

[Response] As suggested by the reviewer, we examined whether PDZD8 mediates lipid transport to endosomes in vivo. Mouse primary neurons expressing a PS reporter protein (EGFP–Lact-C2) and a LyLE marker protein (mCherry-CD63) were transfected with control or PDZD8 siRNAs, and the colocalization rate of PS (EGFP) and LyLEs (mCherry) was determined. The colocalization rate was lower in the PDZD8-depleted neurons than in the control neurons (**new Fig. 6h, i**), suggesting that PDZD8 indeed transfers lipid from the ER to endosomes in vivo. We have now addressed this point in the revised manuscript (page 12, lines 342-348).

(Fig. 6h and 6i) Depletion of PDZD8 attenuated the colocalization of PS and LyLEs in mouse primary neurons.

Response to Reviewer #3

We thank the reviewer for the careful evaluation of our manuscript and for the statement that “The authors have made an impressive amount of new experiments, which have significantly strengthened the conclusions of this manuscript.” This reviewer has no further critiques.

REVIEWER COMMENTS

Reviewer #1 (Remarks to the Author):

In this revised manuscript, Shirane et al., have made an impressive amount of new experiments in both first and second revision. They have satisfactorily addressed all my concerns in this revised version of their manuscript. These new evidences strengthen the conclusions of the authors.

Reviewer #2 (Remarks to the Author):

My concerns have not been fully addressed. Overall, there are many interesting findings in this study but it still does not make a strong case that PDZ8 mediates lipid transport at physiologically relevant rates in vitro or in cells. Here are my responses to the authors, using the numbering from the previous review.

1. The new demonstration that endogenous PDZ8 is at contact sites between the ER and endosomes (Fig. 2A-C) is a very nice addition. However, the quantification (Figs 2C) is not terribly useful. The important question is what percent of ER-endosome contacts have PDZ8. The images in 2A suggest that most do not. The percentage should be determined.

2a. I do not understand what was done or the results. Lipid transport and lipid extraction reactions can be modeled in the same way as chemical reactions. A curve fitting the data should be calculated and used to derive the initial velocity. The result should yield a rate that can be expressed as number of molecules of lipid transferred (or extracted) per protein per unit time. I do not understand how the results in Supplementary Fig. 10e were calculated. They are given in concentration (nM) per minute, which is meaningless. Perhaps the authors mean change in concentration per minute or nmol per min.

2b. The response is not adequate because addressing my concerns requires additional experimentation. Lipid transport and extraction reactions are usually first-order reactions and not biphasic. The authors suggest that the rates at which PDZ8 extracts and delivers lipid from membranes are not the same and this difference explains their findings. This could be true, though it seems highly implausible to me, but it is not enough to just suggest this idea, it must be experimentally tested. It is also possible that the liposomes are not all unilamellar or aggregate or fuse. I had assumed that the authors would experimentally address these possibilities.

2c. In my previous review, I asked why in the lipid transport reaction, most of the NBD-labeled lipid does not end up in the acceptor liposomes, since they are 4-fold more abundant than donor liposomes. The author responded that this is explained by the "low concentration of the transport protein." This cannot be correct. No matter what the protein concentration, ~80% of the NBD-lipid should end up in the acceptor liposome as the reaction nears equilibrium. It is possible, as the authors suggest, that most of the NBD lipid is bound by the protein and therefore is not available to be transferred to the acceptor liposomes. If this were correct, then the protein transports much less than one lipid per reaction, which does not suggest it transports lipids at a physiological rate and, in

any case, experimental evidence is required to support this idea. The more likely explanation is that the donor and acceptor liposomes are not unilamellar or are highly aggregated.

2d. There was no response since the authors felt their response to 2b and 2c was adequate. Even if I agreed with these responses, I do not see how they address my concerns.

2e. I have concerns about any lipid transport study that relies only on NBD (or BODIPY)-labeled lipids. However, I certainly understand that using unlabeled lipids is very challenging. This study would be stronger if it presented evidence that unlabeled lipids are transported at rates similar to NBD lipids. At a minimum, the caveats of using only labeled lipids should be discussed.

3. The added experiments do not measure lipid transport rates *in vivo*, only changes in PS composition, which might not be caused by changes in nonvesicular lipid transport.

Reviewer #4 (Remarks to the Author):

Overall this is an interesting manuscript that details identification of protrudin interaction with the putative ERMES subunit homologue PDZD8 as a potential mechanism to tether and transfer lipids during Ly/LE maturation. I will specifically address the comments made by reviewer 2 and the rebuttal by the authors.

1. The authors have adequately address the co-localization of PDZD8 and protrudin at contact sites. Their explanation and suggested modifications seem appropriate.

2a. Some papers show molarity in their extraction assays since it is easy to compare to the concentration of added lipid binding protein (ie. one can get a sense of the efficiency/stoichiometry of extraction). In this case I agree with reviewer 2 that it is ambiguous since it is expressed relative to time, in this case 300 sec or the entire assay. As suggested by reviewer 2, they could determine extraction rates from the initial velocities (which is feasible). The other issue is the apparent amount of PS extracted in Fig. 3c and d. Only 10 nM of GST-PDZD8-dTM is added to the assay yet panel d reports 300 nM of PS extracted after 300 sec (the entire assay). If this assay reports only extraction (see below), how can 30x more PS be removed? There are either some details missing in the methods and data presentation or the liposome assays is subject to non-specific dequenching of NBD.

2b. Like reviewer 2, I to was initially confused by the supposed lipid transfer activity of PDZD8. The authors have created this situation by initially indicating that PDZD8 has lipid transfer activity. The Figure 3 legend title is "Lipid transfer activity of PDXD8" when clearly not the case. By removing the TM, PDZD8 is untethered and soluble, capable of extraction of phospholipids but unable to deliver to a donor membrane; thus, it does not have transfer activity. Even more ambiguity is added by the model in Fig. 3B that shows transfer at a MCS between two liposomes and unquenching as the lipid is transferred. Later results in this figure (panel g and h) indicate that it is a lipid extraction reaction

(Fig. 3g and h) but this should have been made initially obvious to readers. The in vitro extraction results in Fig. 3 and 4 do not support the model in Fig. 4I because they do not use the full length or a membrane tethered version of PDZD8. On this last point, proteins can be tethered to liposomes by His-tagging and binding to nickel chelating lipids (ie. Ni-NTA-DOGS).

In the new models in the rebuttal letter, they continue to confuse the lipid transfer and lipid extraction. In R4, in the extraction only panel (right), the graph at the bottom still refers to “lipid transfer”. Similarly, in 5R explaining the effect of increased lipid binding protein, the authors continues to refer to ‘extraction’ activity as ‘lipid transfer’ activity.

The new Figure 3b they propose (in the rebuttal letter) is not an improvement. I shows the unquenched, extracted NBD-lipid in the soluble phase when it has to be bound to the PDZD8. Are they authors proposing that PDZD8 extracts the NBD-lipid and then releases it into the aqueous phase?

2c. Again this is confusion created by the authors insistent use of ‘lipid transfer’ to describe what it only a partial extraction reaction. As mentioned above, they have addressed this partially but not to any satisfaction.

2e. Yes, there are concerns with using NBD-lipids since they have bulky hydrophilic groups that interfere with binding. The study would have been strengthen by the use of isotopically labelled lipids in extraction assays. I am not sure excluding this approach would preclude acceptance. However, as outlined above there are outstanding issues regarding the in vitro lipid extraction assays themselves.

3. The authors have addressed this satisfactory given the current methods that are available to measure lipid movement in intact cells.

Response to Reviewer #4

We thank the reviewer for the careful evaluation of our manuscript and for the statement that “Overall this is an interesting manuscript that details identification of protrudin interaction with the putative ERMES subunit homologue PDZD8 as a potential mechanism to tether and transfer lipids during Ly/LE maturation.” We also thank the reviewer for suggestions that we feel have helped us to improve our manuscript. Our specific responses to the points raised are as follows (low-resolution thumbnails of new figures are included here, with the corresponding high-resolution data being submitted with the manuscript):

1. The authors have adequately address the co-localization of PDZD8 and protrudin at contact sites. Their explanation and suggested modifications seem appropriate.

[Response] We are pleased that the reviewer agrees that we have addressed this issue adequately.

2a. Some papers show molarity in their extraction assays since it is easy to compare to the concentration of added lipid binding protein (ie. one can get a sense of the efficiency/stoichiometry of extraction). In this case I agree with reviewer 2 that it is ambiguous since it is expressed relative to time, in this case 300 s or the entire assay. As suggested by reviewer 2, they could determine extraction rates from the initial velocities (which is feasible). The other issue is the apparent amount of PS extracted in Fig. 3c and d. Only 10 nM of GST-PDZD8-ΔTM is added to the assay yet panel d reports 300 nM of PS extracted after 300 s (the entire assay). If this assay reports only extraction (see below), how can 30x more PS be removed? There are either some details missing in the methods and data presentation or the liposome assays is subject to non-specific dequenching of NBD.

[Response] We apologize for any confusion caused by inappropriate presentation of results for the time point of 300 s. We have therefore removed these results and now show all the assay data as initial velocities (nM lipid/s) measured over the first 10 s of the reaction (**new Figs. 3e, 3h, 4c, 4f, 5g and Supplementary Fig. 10d**). The data show that the moles of lipid extracted per second roughly match the moles of PDZD8(ΔTM) protein. We have now clarified this point in the revised manuscript (page 8-9, lines 230-236; page 9, lines 244-246; page 9, lines 251-257; page 9-10, lines 263-266; page 10, lines 287-290).

e
[New Fig. 3e] Velocity of extraction of a variety of lipids by PDZD8(Δ TM).

h
[New Fig. 3h] Velocity of concentration-dependent lipid extraction by PDZD8(Δ TM).

c
[New Fig. 4c] Velocity of lipid extraction by PDZD8 mutants.

[New Fig. 4f] Velocity of lipid extraction by His₆-PDZD8(ΔTM) with liposomes containing DGS-NTA(Ni).

[New Fig. 5g] Velocity of lipid extraction by PDZD8 mutants.

d

Fig.3e	NBD-Lipid	PA	PS	PE	PC	Cer	Chol
	GST-PDZD8	ΔTM, 10 nM	ΔTM, 10 nM	ΔTM, 10 nM	ΔTM, 10 nM	ΔTM, 10 nM	ΔTM, 10 nM
	Velocity (nM lipid/s)	5.89	16.21	6.55	5.93	13.41	0.33
Fig.3h	NBD-Lipid	PS (-Acc)	PS (-Acc)	PS (-Acc)	PS (+Acc)	PS (+Acc)	PS (+Acc)
	GST-PDZD8	ΔTM, 5 nM	ΔTM, 10 nM	ΔTM, 20 nM	ΔTM, 5 nM	ΔTM, 10 nM	ΔTM, 20 nM
	Velocity (nM lipid/s)	3.17	16.21	26.08	2.74	18.67	22.98
Fig.4c	NBD-Lipid	PS	PS	PS	PS	PS	PS
	GST-PDZD8 (5 nM)	ΔTM	ΔTMΔSMP	ΔTMΔPDZ	ΔTMΔPRT	ΔTMΔC1	SMP
	Velocity (nM lipid/s)	3.07	1.21	2.03	2.55	3.32	0.36
Fig.5g	NBD-Lipid	PS	PS	PS			
	GST-PDZD8 (10 nM)	ΔTM	ΔTMΔCC	ΔTMΔCCΔSMP			
	Velocity (nM lipid/s)	15.28	21.74	3.79			
Fig.4f	NBD-Lipid	PS (+DGS, -Acc)	PS (+DGS,+Acc)	PS (+DGS,-Acc)	PS (+DGS,+Acc)		
	His-protein (10 nM)	PDZD8(ΔTM)	PDZD8(ΔTM)	GST	GST		
	Velocity (nM lipid/s)	0.27	0.25	0.00	0.06		

[New Fig. S10d] Summary of velocity of lipid extraction by PDZD8 and its mutants.

As shown in the previous version of our manuscript, the amount of lipid extracted by PDZD8(Δ TM) is essentially the same in the presence or absence of acceptor liposomes (**Fig. 3f–h**), suggesting that PDZD8 is responsible for lipid extraction from donor membranes but not for subsequent lipid insertion into acceptor membranes. We therefore expected that the extracted lipids would be released into the aqueous phase of the assay mixture, and we have now verified this by an experiment in which fluorescently labeled lipid was detected in the aqueous phase after removal of liposomes by ultracentrifugation, only in the presence of PDZD8(Δ TM) (**new Fig. 3i**). The fact that about 30 times as much PS was removed by PDZD8(Δ TM) over 300 s as protein present might thus be due to recycling of PDZD8(Δ TM). Although we previously speculated that PDZD8(Δ TM) is not recycled on the basis of the biphasic pattern of the reaction, this claim essentially had no experimental basis. As mentioned in more detail later, membrane-bound PDZD8 shows a monophasic course of lipid extraction activity in the liposome-FRET assay (**new Fig. 4d–f**). The reason why the pattern of the reaction differs between soluble and membrane-bound forms of PDZD8 is unclear at present and warrants further investigation in the future. We have now clarified this point in the revised manuscript (page 9, lines 246-248; page 9-10, lines 259-268).

[New Fig. 3i] Detection of NBD-lipid extracted by PDZD8(Δ TM) in the aqueous phase after removal of liposomes by ultracentrifugation.

[New Fig. 4d–f] Liposome-FRET assay as performed with His₆-PDZD8(Δ TM) and liposomes containing DGS-NTA(Ni).

The possibility of nonspecific dequenching of NBD is unlikely, given that lipid extraction activity was seen only in the presence of PDZD8(Δ TM). If the data were due to nonspecific dequenching of NBD, they should have been similar in the presence or absence of PDZD8(Δ TM). Indeed, the reaction clearly depends on the presence of functional PDZD8, with neither GST alone nor a PDZD8 mutant lacking the SMP domain, which is thought to be key for lipid extraction activity of this type of protein, giving rise to a substantial signal (**Fig. 4a–c**). We also observed lipid extraction activity for a variety of lipids and found that the results varied depending on the lipid or protein used (**Fig. 3c–e**). These lines of evidence essentially exclude the possibility that the reaction is simply due to nonspecific aggregation or fusion of liposomes in a manner independent of PDZD8 function. We thus conclude that the lipid extraction reaction is indeed mediated by functional PDZD8. We have now clarified these points in the revised manuscript (page 8-9, lines 230-236; page 9, lines 251-258).

2b. Like reviewer 2, I to was initially confused by the supposed lipid transfer activity of PDZD8. The authors have created this situation by initially indicating that PDZD8 has lipid transfer activity. The Figure 3 legend title is “Lipid transfer activity of PDXD8” when clearly not the case. By removing the TM, PDZD8 is untethered and soluble, capable of extraction of phospholipids but unable to deliver to a donor membrane; thus, it does not have transfer activity. Even more ambiguity is added by the model in Fig. 3B that shows transfer at a MCS between two liposomes and unquenching as the lipid is transferred. Later results in this figure (panel g and h) indicate that it is a lipid extraction reaction (Fig. 3g and h) but this should have been made initially obvious to readers. The in vitro extraction results in Fig. 3 and 4 do not support the model in Fig. 4I because they do not use the full length or a membrane tethered version of PDZD8. On this last point, proteins can be tethered to liposomes by His-tagging and binding to nickel chelating lipids (ie. Ni-NTA-DOGS). In the new models in the rebuttal letter, they continue to confuse the lipid transfer and lipid extraction. In R4, in the extraction only panel (right), the graph at the bottom still refers to “lipid transfer”. Similarly, in 5R explaining the effect of increased lipid binding protein, the authors continues to refer to ‘extraction’ activity as ‘lipid transfer’ activity. The new Figure 3b they propose (in the rebuttal letter) is not an improvement. I shows the unquenched, extracted NBD-lipid in the soluble phase when it has to be bound to the PDZD8. Are they authors proposing that PDZD8 extracts the NBD-lipid and then releases it into the aqueous phase?

[Response] We again apologize for any confusion caused by inappropriate use of the term “lipid transfer.” We have now replaced this term with “lipid extraction” throughout the text and figures as well as changed the order of figures in the revised manuscript. In addition, the model figures have been altered to avoid misunderstanding (**new Figs. 3b, 4d, and 5h**).

b
[New Fig. 3b] Schematic representation of the liposome-FRET assay as performed with GST-PDZD8(Δ TM) in the absence or presence of acceptor liposomes.

d
[New Fig. 4d] Schematic representation of the liposome-FRET assay as performed with His₆-PDZD8(Δ TM) and DGS-NTA(Ni)-containing liposomes.

h
[New Fig. 5h] Mode of action for lipid extraction by PDZD8.

As suggested by the reviewer, we tethered PDZD8(Δ TM) to liposomes containing DGS-NTA(Ni) with the use of a His₆ tag and performed the liposome-FRET assay. The membrane-bound form of PDZD8 showed substantial lipid extraction activity in the absence or presence of acceptor liposomes, whereas His₆-GST did not manifest such activity (**new Fig. 4d–f**), excluding the possibility that this reaction reflects nonspecific dequenching of NBD.

As mentioned above, we examined whether PDZD8(Δ TM) extracts NBD-lipid and then releases it into the aqueous phase by removing the liposomes by ultracentrifugation after the reaction. NBD-labeled lipid was indeed detected in the aqueous phase only in the presence of PDZD8(Δ TM) (**new Fig. 3i**).

We have now addressed these issues in the revised manuscript (page 9, lines 246-248; page 9-10, lines 259-268).

2c. Again this is confusion created by the authors insistent use of 'lipid transfer' to describe what it only a partial extraction reaction. As mentioned above, they have addressed this partially but not to any satisfaction.

[Response] As mentioned above, we have now replaced the term “lipid transfer” with “lipid extraction” throughout the text and figures of the revised manuscript.

2e. Yes, there are concerns with using NBD-lipids since they have bulky hydrophilic groups that interfere with binding. The study would have been strengthen by the use of isotopically labelled lipids in extraction assays. I am not sure excluding this approach would preclude acceptance. However, as outlined above there are outstanding issues regarding the in vitro lipid extraction assays themselves.

[Response] We agree that our study would have been strengthened by the use of isotopically labeled lipids in the extraction assays. In the process of sorting out priorities for revision, however, the editor suggested that this experiment would not be needed for further consideration of our manuscript. We have now referred to the fact that NBD-lipids have bulky hydrophilic groups that might possibly interfere with the binding of PDZD8(Δ TM) in the Discussion section of the revised manuscript (page 15, lines 446-453).

3. The authors have addressed this satisfactory given the current methods that are available to measure lipid movement in intact cells.

[Response] We thank the reviewer for agreement that we have addressed this point raised by Reviewer #2.

REVIEWERS' COMMENTS

Reviewer #4 (Remarks to the Author):

All my concerns and queries have been addressed. Just as an aside, it is possible, as you mention in the discussion, that PDZD8 may required another protein to complete the transfer to donor membranes. However, also possible that the NBD-lipids could be released into the soluble phase due to their structure and increased solubility when compared to the native lipids.

Response to Reviewer #4

We thank the reviewer for the careful evaluation of our manuscript and for the statement that “All my concerns and queries have been addressed.” We also thank the reviewer for suggestions that we feel have helped us to improve our manuscript. Our specific responses to the points raised are as follows:

1. Just as an aside, it is possible, as you mention in the discussion, that PDZD8 may require another protein to complete the transfer to donor membranes.

[Response] We have already stated “It is of note that PDZD8 was found to possess only lipid extraction activity; it did not show the ability to insert lipids into acceptor liposomes by itself. Other proteins associated with PDZD8 might thus contribute to insertion activity, with such a complex containing PDZD8 possessing the ability both to extract lipid from the donor membrane and to insert it into the acceptor membrane.” in the Discussion section of the revised manuscript (page 15, lines 446-450).

2. However, also possible that the NBD-lipids could be released into the soluble phase due to their structure and increased solubility when compared to the native lipids.

[Response] We agree with the reviewer that the NBD-lipids might be released into the soluble phase due to their structure and increased solubility when compared to the native lipids. We have now modified the text in the Discussion section of the revised manuscript (page 15, line 452).

It is of note that lipid extraction activity was dependent on the presence of functional PDZD8, with neither GST alone nor a PDZD8 mutant lacking the SMP domain, which is thought to be key for lipid extraction activity of this type of protein, giving rise to a substantial signal (**Fig. 4a–c**). These lines of evidence essentially exclude the possibility that the reaction is simply due to increased solubility of NBD-lipids in a manner independent of PDZD8 function. We thus conclude that the lipid extraction reaction is indeed mediated by functional PDZD8.